# Exploring the composition and volatility of secondary organic aerosols in mixed anthropogenic and biogenic precursor systems

Aristeidis Voliotis[1], Yu Wang[1], Yunqi Shao[1], Mao Du[1], Thomas J. Bannan[1], Carl J. Percival[2], Spyros N. Pandis[3], M. Rami Alfarra[1,4], and Gordon McFiggans[1]

[1]Centre for atmospheric science, Department of Earth and Environmental Science, School of Natural Sciences, The University of Manchester, Oxford Road, M13 9PL, Manchester, United Kingdom

[2]NASA Jet Propulsion Laboratory, California Institute of Technology, 4800 Oak Grove Drive, Pasadena, CA 91109, USA.

[3]Department of Chemical Engineering, University of Patras, Patras, Greece

[4]National Centre for Atmospheric Science, Department of Earth and Environmental Science, School of Natural Sciences, The University of Manchester, Oxford Road, M13 9PL, Manchester, United Kingdom

*Correspondence to*: g.mcfiggans.manchester.ac.uk; aristeidis.voliotis@manchester.ac.uk

**Abstract.** Secondary organic aerosol (SOA) formation from mixtures of volatile precursors may be influenced by the molecular interactions of the components of the mixture. Here, we report measurements of the volatility distribution of SOA formed from the photo-oxidation of *o*-cresol, *α*-pinene and their mixtures, representative anthropogenic and biogenic precursors, in an atmospheric simulation chamber. The combination of two independent thermal techniques (thermal denuder; TD, and the Filter Inlet for Gases and Aerosols coupled to a high resolution time of flight chemical ionisation mass spectrometer; FIGAERO-CIMS) to measure the particle volatility, along with detailed gas and particle phase composition measurements provides links between the chemical composition of the mixture and the resultant SOA particle volatility. The SOA particle volatility obtained by the two independent techniques showed substantial discrepancies. The particle volatility obtained by the TD was wider, spanning across the LVOC and SVOC range, while the respective derived from the FIGAERO-CIMS using two different methods (i.e., calibrated $T_{max}$ and partitioning calculations) was substantially higher (mainly in the SVOC and IVOC, respectively) and narrow. Although the quantification of the SOA particle volatility was challenging, both techniques and methods showed similar trends, with the volatility of the SOA formed from the photo-oxidation of *α*-pinene being higher that measured in the *o*-cresol system, while the volatility of the SOA particles of the mixture was between those measured at the single precursor systems. This behaviour could be explained by two opposite effects, the scavenging of the larger molecules with lower volatility produced in the single precursor experiments that led to an increase of the average volatility and the formation of unique-to-the-mixture products that had higher O:C, MW, $\overline{OSc}$ and consequently, lower volatility compared to those derived

from the individual precursors. We further discuss the potential limitations of FIGAERO-CIMS to report quantitative volatilities and their implications to the reported results and we show that the particle volatility changes can be qualitatively assessed, while caution should be held when linking the chemical composition to the particle volatility. These results present the first detailed observations of SOA particle volatility and composition in mixed anthropogenic and biogenic systems and provide an analytical context that can be used to explore particle volatility in chamber experiments.

## 1. Introduction

Aerosol particles are ubiquitous in the atmosphere with substantial impacts on climate (Ramanathan et al., 2001) and human health (Lelieveld et al., 2015;Brunekreef and Holgate, 2002). These particles may contain a wide variety of compounds, with the organic fraction contributing up to 90% of the mass in the submicron size range (Jimenez et al., 2009). The majority of this fraction is comprised by secondary organic aerosol (SOA) (Kanakidou et al., 2005;Hallquist et al., 2009;Shrivastava et al., 2017). SOA is formed by the oxidation of volatile organic compounds (VOC) in the atmosphere. These reactions result in a variety of products that can be more or less volatile than their precursors depending on the underlying chemical processes in the gas and/or condensed phase (Donahue et al., 2012). The less volatile products of these reactions tend to partition to pre-existing particles and/or nucleate to form new particles. SOA containing particles can substantially affect climate due to their optical properties (Moise et al., 2015) and their ability to act as cloud condensation nuclei (CCN) (McFiggans et al., 2006), while their potential adverse effects on human health have recently drawn the attention (Chowdhury et al., 2018;Kramer et al., 2016). Our current mechanistic understanding and the resulting SOA representation in predictive models remains inadequate (Shrivastava et al., 2017;McFiggans et al., 2019), leading to a significant uncertainty in the assessment of their impacts.

The analytical limitations in the characterization and detection of all oxidation products as well as the complexity of the ever-evolving atmosphere (Isaacman-VanWertz et al., 2018), makes the understanding of the SOA behaviour challenging. Atmospheric simulation chambers have been employed as a tool to study the formation and aging of SOA under simplified conditions (Burkholder et al., 2017). The oxidation of biogenic VOC (bVOC), and particularly terpenes, has been studied extensively in chamber experiments mainly as a result of their large contribution to global emissions (Spracklen et al., 2011) and their strong SOA formation potential (Lee et al., 2006).

Although anthropogenic VOC (aVOC) emissions contribute less to the global emissions (Goldstein and Galbally,

2007) than the respective biogenic (Aiken et al., 2009), at a regional scale their relative emissions may be substantially higher, therefore they have been widely studied as SOA precursors in chamber experiments (Baltensperger et al., 2005;Wyche et al., 2009). These experiments have shown that both biogenic and anthropogenic VOC oxidation can yield hundreds or thousands of products that are highly reactive and have a wide range of volatilities (Hallquist et al., 2009).

The SOA volatility is strongly dependent on the molecular weight of the compounds (Li et al., 2016) and their functionality (and to a lesser extent, the activity coefficients) (McFiggans et al., 2010;Barley et al., 2009;Topping et al., 2011). Condensed phase reactions may further alter the SOA volatility (Kroll and Seinfeld, 2008). Therefore, the chemical reactions in both the gas and condensed phases determine the aerosol volatility, and thereby the partitioning of the species between the two phases and their subsequent transformation pathways.

Recently, the formation of highly oxygenated organic molecules (HOM), defined as organic compounds with at least 6 oxygen atoms through the autoxidation of peroxy radicals ($RO_2$) (Bianchi et al., 2019), has been shown to be a critical process affecting the growth of newly formed particles (Tröstl et al., 2016;Stolzenburg et al., 2018). The fate of $RO_2$, which can participate in autoxidation or other explicit termination reactions such as $RO_2+HO_2$, $RO_2+NO$ and $RO_2$ cross-reactions, has a determinant role for the SOA formation and volatility (Schervish and

Donahue, 2020).

Chamber studies have mainly focused on studying the SOA formation and their resultant properties from a single VOC precursor. Recently however, McFiggans et al. (2019) showed that upon mixing bVOC precursors, the SOA formation is governed by the molecular interactions of the products. Specifically, it was shown that upon mixing a high and a low yield bVOC ($\alpha$-pinene and isoprene, respectively), the overall SOA formation was suppressed due

to i) the hydroxyl radical (OH) scavenging by the lower yield isoprene and ii) due to product scavenging. In the second process, it was surmised that the peroxy radical ($RO_2$) oxidation products of isoprene were scavenging the highly oxygenated $RO_2$ oxidation products (HOM-$RO_2$) of the $\alpha$-pinene, leading to higher volatility products. This demonstrates that product interactions in mixed precursor systems may alter the product distribution, with consequent implications to the SOA formation and properties.

In this work, we extend the system studied by McFiggans et al. (2019; i.e., $\alpha$-pinene and isoprene) with the investigation of the interactions of an aVOC with the two bVOC. We selected *ortho*-cresol (hereafter *o*-cresol), a

product of toluene oxidation and also a directly emitted aromatic species, as a representative aVOC. In each system, the initial concentrations of the reactants were adjusted to have equal reactivity towards OH at the beginning of each experiment such that the mixtures were initially iso-reactive towards the dominant oxidant in the system. *o*-cresol is a convenient choice, enabling similar concentrations of VOC to be used in initially iso-reactive mixtures, owing to its relatively high reactivity with OH and moderate SOA yield (Atkinson, 2000;Henry et al., 2008). This choice of precursors and contributory reactivities enabled investigation of SOA formation with mixed systems containing a low, a moderate and a high yield precursor (i.e., isoprene; 0-4%; Carlton et al. (2009), *o*-cresol; 7-12%; Smith et al. (1998) and *α*-pinene; 20-30%; Eddingsaas et al. (2012a), respectively) with equal initial chances of reacting with the dominant oxidant and equal contributions of first-generation oxidation products. This manuscript focuses in exploring the ability of the available tools to investigate the SOA particle composition and volatility using the *α*-pinene and *o*-cresol subset of the system as a proof of concept. Subsequent manuscripts will focus on the SOA formation dynamics and other aspects of the binary and ternary mixtures.

The SOA particle volatility in single precursor systems has been investigated based on the aerosol evaporation after heating (Lopez-Hilfiker et al., 2015;Donahue et al., 2006), by isothermal dilution (Yli-Juuti et al., 2017) or a combination of the two (Louvaris et al., 2017;Cain et al., 2020). Previous works have showed that the oxidation of *α*-pinene results in low volatility and semi-volatile components (Zhang et al., 2015), while extremely low volatility material could be formed from oligomerisation reactions (Lopez-Hilfiker et al., 2015). On the other hand, the oxidation of aromatic compounds have recently shown to proceed through rapid, multiple generations of OH oxidation resulting in more oxygenated and less volatile material compared to that derived from biogenic precursors (Garmash et al., 2020;Wang et al., 2020). The effect of mixing anthropogenic and biogenic compounds have been investigated in much lesser extent with their outcomes being inconclusive. Some studies suggest that the SOA formation could be enhanced (e.g., Shilling et al., 2013) or supressed (e.g., Ahlberg et al., 2017) in such mixtures. Further, to our knowledge, there is much less information regarding the composition and volatility changes in mixtures of anthropogenic and biogenic precursors.

In this study, we employ two independent thermal techniques (thermal denuder and FIGAERO coupled to a high resolution time of flight chemical ionisation mass spectrometer) and we aim to provide the first observational quantification of the volatility of SOA formed from the mixing of an aVOC and a bVOC precursor with modest and high yields, respectively. Additionally, we use detailed gas and particle phase composition measurements to contextualise our volatility estimations and identify potential chemical drivers that may influence the SOA particle

volatility in such mixtures. We further aim to investigate the ability of the instrumentation used to explore the SOA particle volatility and discuss any potential implications from the interpretation of their results.

## 2.    Materials and methods

 ### 2.1.    Manchester Aerosol Chamber

The experiments were conducted at the University of Manchester Aerosol Chamber (MAC) facility (Alfarra et al., 2012;Shao et al., 2021). MAC is an 18 $m^3$ fluorinated ethylene propylene (FEP) bag mounted on three rectangular frames, enclosed in a temperature and relative humidity (RH) controlled housing, operating as a batch reactor. The central frame is fixed whereas the lower and upper frames are free to move, allowing the bag to collapse and expand as chamber air volume changes, maintaining atmospheric pressure without dilution. Light is generated by a series of halogen lamps (Solux 50 W/4700 K, Solux MR16, USA) and two 6 kW Xenon arc lamps (XBO 6000 W/HSLA OFR, Osram) over the range of 290-800 nm. Quartz glass filters were used in front of each arc lamp to reduce the radiation flux below 300 nm and mimic the atmospheric radiation spectrum as closely as possible. The photolysis rate of $NO_2$ ($J_{NO2}$), as computed from steady-state actinometry experiments, was 0.11-0.18 $min^{-1}$. The $O_3$ formed by $NO_2$ photolysis was itself photolysed in the moist chamber atmosphere (RH =~50%) to yield a OH concentration from primary production, as calculated by the decay rate of solely OH reacting VOCs (e.g., toluene, $o$-cresol), of around $1 \times 10^6$ molecules $cm^{-3}$. Liquid VOC precursors ($\alpha$-pinene and $o$-cresol; Sigma Aldrich, GC grade ≥99.99% purity) were added to the chamber by injecting the desired amount into a heated glass bulb and transferred into the chamber using gentle electronic capture device- grade N4.8 (purity 99.998%) nitrogen (hereafter ECD $N_2$) stream. Seed particles were nebulised using an aerosol generator (Topaz model ATM 230) from aqueous solutions of ammonium sulfate (Puratonic, 99.999% purity). $NO_x$ concentration was controlled by adding the desired amount of $NO_2$ from a cylinder using ECD $N_2$ as carrier gas. All the reactants (VOC, $NO_x$ and seed) were introduced to the chamber at high flow rate (~3 $m^3$ $min^{-1}$) during the final fill cycle, resulting in rapid mixing (see section 2.2.). The mixing and temperature control throughout the experiment was effected through the use of conditioned air circulated at high flowrate around the chamber housing, continually agitating the chamber walls without physical contact.

## 2.2.    Experimental procedure

The experiments conducted in this study were representative of "daytime" photo-oxidation in the presence of $O_3$ and OH. This study, as also described above, is part of a broader project investigating SOA derived from mixtures of precursors with distinct yields and properties. Here, we focus on the $\alpha$-pinene and $o$-cresol system, as representative biogenic and anthropogenic VOC precursor sources (Hallquist et al., 2009;Schwantes et al., 2017), both with appreciable SOA yields (7-30%; e.g., Henry et al., 2008;Eddingsaas et al., 2012a). To study the effect of mixing anthropogenic and biogenic precursors on SOA particle volatility, two types of experiments were conducted: (a) single precursor experiments, where the photo-oxidation of each precursor was studied individually and (b) mixture experiments, where both precursors were added to the chamber simultaneously. Since $o$-cresol is practically unreactive towards $O_3$ within our experimental timescales and because $\alpha$-pinene's lifetime with respect to OH is shorter than towards $O_3$ at the expected oxidant concentrations in our experiment (5.2 vs 6.3 h; Atkinson, 2000), the concentrations of the reactants were selected based on the reactivity of each precursor with OH. As such, each VOC has equal initial chances of reacting with the dominant oxidant in the mixture experiments, while the overall initial reactivity (defined as the product of the VOC concentration and the reaction rate coefficient with OH; eq.1) in both single-precursor and mixture experiments was the same. The ozone concentration at the beginning of each experiment was zero. Ozone is however formed rapidly from the photolysis of $NO_2$ after the lights are turned on to establish photostationary state and then continuously produced through the oxidation of VOC in the presence of $NO_x$ (Atkinson, 2000). Therefore, our systems are "iso-reactive" at the beginning of each experiment, though not necessarily thereafter. All the experiments were conducted under modest $NO_x$ conditions (VOC/$NO_x$ 6±2 ppb/ppb) by adding $NO_2$, and moderate RH and temperature conditions (50±5% and 24±2 °C, respectively). Ammonium sulfate seed particles (53±12 µg m$^{-3}$) were added to provide a medium for condensation of the condensable VOC oxidation products to successfully compete with the walls, to reduce the influence of vapour wall-losses and also to suppress nucleation (Zhang et al., 2014). The initial experimental conditions are summarised in Table 1.

$$Initial\ reactvity\ (s^{-1}) = \sum_i C_{VOC,i} \times k_{OH,i}\ (1)$$

Where, $C_{VOC,i}$ is the concentration of each VOC (molecules cm$^{-3}$) in the system and $k_{OH,i}$, is the reaction rate coefficient of each VOC towards OH (cm$^3$ molecules$^{-1}$ s$^{-1}$).

Each experiment was comprised of a common set of procedures shown in Fig. S5a. Briefly:

i)     The "pre-experiment" procedure entailed flushing of the chamber with clean air at a high flow rate ($\sim 3$ m$^3$ min$^{-1}$) for $\sim 1.5$ h.

    ii)     The "chamber stabilisation" was established subsequently by leaving the chamber with clean air without reactants. This phase lasted for $\sim 1$ h and the established baselines that were subtracted from the measurements.

iii)     The "dark unreactive" phase commenced after the addition of the VOC precursor(s), $NO_2$ and seed aerosol to the chamber in the dark. In order to ensure rapid and effective mixing, MAC was partly flushed to an approximate 1/3 of its volume and all the reactants were subsequently added to the chamber while filling with clean air to its maximum capacity (i.e., final fill cycle). During this period (also $\sim 1$ h), the initial conditions of the chamber were established.

iv)     The "experiment" phase commenced when the chamber lights were turned on, initiating the photo-oxidation and consequent SOA formation. Each "experiment" phase lasted for $\sim 6$ h.

    v)     A "post-experiment" cleaning procedure normally comprised the high flowrate flushing of the chamber with clean air for $\sim 1.5$ h, subsequently filling with high concentration of $O_3$ ($\geq 1$ ppm) and soaking overnight to oxidise any remaining $O_3$-reacting gas-phase organic species. It should be further
noted that within the experimental campaign an intensive cleaning procedure was conducted periodically (at least once per week). In this procedure, the chamber was filled with high concentrations of $O_3$ ($\geq 1$ ppm) and water vapour (RH$\geq 70\%$) and the chamber was illuminated for at least 4 h, after the quartz filters have been removed in front of the arc lamps to maximise the jO($^1$D) and thereby the OH levels. The chamber was then flushed for 1.5h with high flow rate clean air (3 m$^3$
190     min$^{-1}$). All the results given below correspond to the "experiment" phase only.

As shown in Table 1, a number of repeat experiments were conducted for each system in order to increase confidence in the validity of our results as well as to overcome technical difficulties due to instrument failures over certain experiments (see Table S1). The repeat experiments showed good agreement, as it is depicted in the maximum SOA masses (Table 1), as well as from the relatively low standard deviations of the average
values within each system for a number of parameters such as the carbon and oxygen number distribution, the sum thermograms and the mass fraction remaining (see Fig. S1-4). In the sections below, three experiments were randomly chosen as characteristic; Experiments 2, 6 and 10 (Table 1) to represent the single $\alpha$-pinene, single $o$-cresol and mixed VOC, respectively.

**Table 1: Summary of the initial conditions for each of the experiments conducted in this study.**

| Experiment No. | Experiment type | $NO_x$ (ppb) | VOC (ppb) | $VOC/NO_x$ (ppb/ppb) | Seed Concentration ($\mu g\ m^{-3}$) | Maximum SOA mass ($\mu g\ m^{-3}$) |
|---|---|---|---|---|---|---|
| | **Single VOC** | | | | | |
| 1 | $\alpha$-pinene | 40 | 309 | 7.7 | 72.6 | 273.2 |
| 2 | $\alpha$-pinene | 43 | 309 | 7.2 | 67.6 | 283.1 |
| 3 | $\alpha$-pinene | 37 | 309 | 8.4 | 62.0 | n.a.[*] |
| 4 | $\alpha$-pinene | 50 | 309 | 6.2 | 39.4 | n.a.[*] |
| 5 | $o$-cresol | 56 | 400 | 7.1 | 67.6 | 23.5 |
| 6 | $o$-cresol | 71 | 400 | 5.6 | 36.0 | n.a.[*] |
| 7 | $o$-cresol | 98 | 400 | 4.1 | 50.8 | 26.7 |
| | **Mixture** | | | | | |
| 8 | $\alpha$-pinene+$o$-cresol | n.a.[*] | 355 (155+200)[a] | n.a.[*] | 42.5 | 130.1 |
| 9 | $\alpha$-pinene+$o$-cresol | 30 | 355 (155+200)[a] | 11.8 | 57.0 | 131.4 |
| 10 | $\alpha$-pinene+$o$-cresol | 52 | 355 (155+200)[a] | 6.8 | 48.3 | 122.3 |

[*]**n.a. values correspond to corresponding instrument failures at the day of the experiment.**

[a]**Sum of the VOC added in the mixture the individual concentrations of α-pinene and o-cresol added are shown in the brackets, respectively.**

## 2.3. Instrumentation

### 2.3.1. TD- AMS/SMPS

A scanning mobility particle sizer (SMPS; DMA 3081 and CPC 3776, TSI Inc., USA) was used to measure aerosol size distributions in the range of 10-670 nm, operating at 0.3 L min$^{-1}$ (1:10 sample/sheath flow). The non-refractory
aerosol chemical composition and mass were measured by a high-resolution time of flight aerosol mass spectrometer (HR-ToF-AMS, ARI Inc., USA; hereafter HR-AMS) (DeCarlo et al., 2006), sampling at 0.1 L min$^{-1}$. The HR-AMS was calibrated regularly using previously published procedures (Jimenez et al., 2003;Allan et al.,

2004). Briefly, a series of calibrations using monodisperse ammonium nitrate and sulfate particles (350 nm) were conducted within the experimental campaign. An average ionisation efficiency of ammonium nitrate was estimated at $9.38 \times 10^{-8}$ ions molecule$^{-1}$. Based on the ion balance of ammonium nitrate and ammonium sulfate in these calibrations, specific relative ionisation efficiencies (RIE) were determined for $NH_4^+$ and $SO_4^{2-}$ as $3.57 \pm 0.02$ and $1.28 \pm 0.01$, respectively. The RIE for all organics was set to the default value (i.e., 1.4; Alfarra et al., 2004). A real-time collection efficiency (CE) was applied to the data of each experiment by comparing the HR-AMS total mass with the total mass from the SMPS multiplied by an effective density, based on the organic/inorganic ratio from the HR-AMS, assuming densities of 1.77 and 1.4 g cm$^{-3}$ for the inorganic and organic fraction, respectively. The CE for ammonium sulfate, determined in the "dark unreactive" period was found to be ~0.3-0.5, in line with previous findings (Alfarra et al., 2004), and the more SOA condensed the CE increased to reach almost unity (i.e., ~0.9-1.1), also in line with previous of observations (Matthew et al., 2008).

The TD unit employed here was designed based on the recommendations of Fuentes and McFiggans (2012). Briefly, sample flow enters and exits the TD unit via a cylindrical 0.12 m long and 36.6 mm inner diameter (ID) stainless steel compartment. The heating section which is placed in a housing with insulating material, has a length of 0.51 m and an ID of 151 mm, and its temperature is controlled by four PID controllers (Watlow EZ-ZONE). A cooling section with a length of 0.2 m and an ID of 20 mm, allows the sample to cool down to ambient temperature with minimal re-condensation (Fuentes and McFiggans, 2012). In our configuration, the TD was operating at temperatures ranging from 30 to 90 °C in 12 steps and the HR-AMS and SMPS sampling alternated between the bypass (BP) and TD every 4 and 6 min, respectively, using a 3-way electronic valve. The alternating sampling through the TD unit was initialised at the final 2 h of each experiment where minimal changes in the aerosol composition over time are expected (i.e., SOA formation rate<wall loss rate). The resulting mass fraction remaining (MFR) in each temperature step was calculated as the ratio of the organics mass (measured by the HR-AMS; hereafter SOA mass) passing through the TD to the average bypass concentration obtained before and after each step. The sample flow rate through the TD and BP was adjusted to 1 L min$^{-1}$, resulting in an average residence time of the aerosol in the heating section of 31 s, using a vacuum line and a mass flow controller. All the TD data were corrected for particle losses, based on characterisation experiments conducted using sodium chloride particles and the SMPS unit to measure the total mass losses for each temperature step.

## 2.3.2. FIGAERO-ToF-CIMS

Near-real time gas and particle composition was measured using the Filter Inlet for Gas and Aerosols (FIGAERO; Aerodyne Research Inc.) (Lopez-Hilfiker et al., 2014) coupled to an Iodine high-resolution time of flight chemical ionisation mass spectrometer (Lee et al., 2014) (hereafter FIGAERO-CIMS). Briefly, the FIGAERO manifold allows the determination of both particle and gas phases by collecting particles on membrane filter while sampling the gas phase. After a period of time, the deposited aerosol is thermally desorbed using ultra high purity (UHP) $N_2$ as a carrier gas. In our setting, particles were sampled from the middle of the chamber through a 2 m long, ¼-inch stainless steel tube at 1 sL min$^{-1}$ and deposited on a PTFE filter (Zefluor, 2.0 µm pore size). The flow over the filter was continually monitored by an MKS mass flow meter (MFM) after the sample has passed over the filter to ensure a known volume is sampled in each cycle. The gas phase was sampled through a separate line, using a 2 m long, ¼-inch PTFE tubing (Sulyok et al., 2002;Mittal et al., 2013) at 1 sL min$^{-1}$ that results in a sample residence time of <~2s. This gas phase sample flow was again constantly measured a MFM at the exhaust of the IMR pump which included the reagent ion flows. The instrument was run in negative-ion mode with I$^-$ reagent ion in all the experiments. The reagent ions were produced by passing a mix of $CH_3I$ and UHP $N_2$ over a $^{210}$Po radioactive source and passed directly into the IMR.

The FIGAERO-CIMS was operated in cyclic mode with 30 min of gas sampling and simultaneous particle collection, followed by 15 min of thermal desorption from ambient to 200 ºC (temperature ramp period) and then kept constant at the maximum temperature for another 10 min (i.e., soak period) to ensure that all the remaining organics mass was evaporated. Last, the FIGAERO was cooled down to ambient temperature within 10 min (i.e., cooling period). An example of the FIGAERO cycle is shown in Fig. S5b and 5c. In order to account for any potential instrument contamination during the gas-phase sampling, the instrument was flushed with UHP $N_2$ for a period of 0.2 min every 2 min, thus a gas phase instrument background value was regularly established. The signal over each background cycle was averaged and was subtracted from the nearest in time measurements.

As a result of the low organic particles concentration in the first hours of the *o*-cresol experiment and for comparison with the concurrent TD measurements, all the results shown by the FIGAERO-CIMS here correspond to the data obtained during the last cycle of the FIGAERO in each experiment.

It should be noted that quantification from the FIGAERO-CIMS in both the particle and gas phase remains challenging as a result of a lack of available standards and the experimental limitations (Riva et al., 2019). Despite

a number of studies attempting to constrain those limitations (Lopez-Hilfiker et al., 2016a;Aljawhary et al., 2013), significant challenges remain. Therefore, here, we assumed uniform instrument sensitivity across the whole range of the detected products and an assessment of the impact of increased sensitivity to particular functional group additions on the reported results is discussed in Section 4.3.

## 2.4. Data analysis

### 2.4.1. FIGAERO-ToF-CIMS

The FIGAERO-CIMS data were analysed using the Tofware package in Igor Pro (v. 3.2.1., Wavemetrics©) (Stark et al., 2015). The peak fitting was conducted in the region of 190-550 m/z, where the vast majority of the total signal occurred (>70%; excluding the reagent ions; $I^-$ $I_2^-$ $I_3^-$ and $I.H_2O^-$). The mass to charge calibration was performed in the range of 127-380 m/z using four known masses, $I^-$, $CH_2O_2I^-$, $I_2^-$ and $I_3^-$ and the resulted calibration error was within 3 ppm. Initially we sorted all the remaining unit mass resolution (UMR) peaks based on their contribution to the total signal and we began the HR peak identification in descending order, until ≥80% of the total signal was assigned. All the elemental formulae were assigned to the HR spectrum according to their mass defect, within ~6 ppm error. The remaining unidentified individual UMR peaks made minor contributions to total signal (≤0.3%) and no possible formulae were available within a reasonable fitting error (i.e., ≤ 6 ppm), while 30-50% of it had low signal to noise ratio (S/N≤2). In this work, only the species with assigned formulas were considered. Further, note that only the particle phase data collected during the temperature ramp were considered here as the integration of the data requires a linear increase in the desorption temperature (e.g., Buchholz et al., 2020).

The gas-phase data obtained during "instrument background" mode (see Section 2.3 and Fig. S5b) were averaged in background cycle and subtracted from the nearest in time measurements. To account for any potential background gas phase species in the chamber, the gas phase sampling data obtained during the "chamber stabilisation" phase were also subtracted from the data obtained during the "experiment" phase (see Section 2.2 and Fig. S5). For the particle-phase data, a similar procedure was followed. In certain cases, a high initial particle phase signal was observed as it was carried out from the gas phase sampling, with a desorption profile not being evident (see Fig. S5c; examples #1 and 4). This is likely the result of the instrument being contaminated by the high levels of "sticky" compounds measured during the gas phase sampling. In other cases, the high initial particle

phase signal was depleted within several seconds and a desorption profile was evident (see Fig. S5c; examples #2, 3 and 5). To correct for these interferences a corresponding particle-phase "instrument background" period was assumed. We selected the 60-90th second of each desorption cycle as a particle-phase "instrument background" period (Fig. S5c) because (i) the desorption temperature was still at ambient levels (<22°C; Fig. S6b), implying that the thermal desorption process had not yet started and (ii) this period avoided the "noisy" signal that appeared during the first 10-15 s of each desorption cycle produced by the IMR pressure change due to the actuator shifting position (Fig. S6a). The average signal of the particle-phase "instrument background" period was then subtracted from the raw desorption data and the resultant thermograms (signal vs. desorption temperature) were integrated using the trapezoid rule. Subsequently, the integrated area of each peak measured during the FIGAERO-CIMS desorption cycle during "dark unreactive" phase (see Section 2.2; Fig. S5a) was subtracted from the data to remove any seed and/or filter effect. All the subsequent analysis presented in this work contains the background-corrected data.

To identify the products that were uniquely formed in the mixed VOC systems, as well as those that uniquely correspond to either precursor in the mixed system, we compared the products identified in single precursor experiments to those of the mixture as follows. First, the compounds with the same assigned formulae that were found in all systems (i.e., both single precursor and mixture) were classified as "common". Subsequently, the remaining products that were found in each single precursor system and in the mixed system were classified as unique products of either precursor in the mixture. All the remaining products, that were not found in either single precursor system nor were classified as "common", were classified as "initially unique to the mixture". Finally, in order to take into account any potential products which were not identified in single experiments but still uniquely correspond to either of the two precursors and were potentially miscounted as "unique to the mixture" products, we compared the products categorised as "unique to the mixture" with the Master Chemical Mechanism (MCM v3.2;Saunders et al., 2003;Jenkin et al., 2003). More specifically, the products initially classified as "unique to the mixture" were compared with the product lists of α-pinene and o-cresol found in MCM and any products that were found to uniquely match any of the records or were found common between all lists were subsequently moved to the corresponding individual precursor's or common lists. The above procedure is shown schematically in Fig. S7.

### 2.4.2.   Volatility retrieval from TD-AMS

The Mass Fraction Remaining (MFR) data obtained from the HR-AMS were used to calculate the volatility distributions of the bulk SOA. The volatility distributions here were expressed as effective saturation concentration at 298 K ($C^*$; µg m$^{-3}$) bins in the Volatility Basis Set (VBS) framework (Donahue et al., 2006) and were calculated by the algorithm of Karnezi et al. (2014). Briefly, the approach uses the thermodynamic mass transfer model of Riipinen et al. (2010) to fit the MFR measurements of the thermodenuder selecting appropriate volatility distributions and effective enthalpies of vapourisation. The model neglects any temperature gradients in the TD and assumes monodisperse aerosol population, equal to the mean volumetric diameter. The inputs of the model, in addition to the MFR measurements, were the average residence time in the heating section (31 s), the length of the heating section (0.51 m), the average volumetric diameter of the particles entering the TD (calculated from SMPS data for the by-pass line), the average total mass concentration bypassing the TD and the SOA density (assumed as 1.4 g cm$^{-3}$). The volatility distribution was expressed in six volatility bins spanning from $\log_{10}C^*$=-3 to 2 µg m$^{-3}$.

### 2.4.3.   Volatility retrieval from FIGAERO-CIMS

In this work, we have estimated the SOA particle volatility from the products identified in the FIGAERO-CIMS using two different approaches: i) using explicit instrument calibrations similar to those described in Bannan et al. (2019) and ii) based on the gas to particle ratio of each individual product and the absorptive partitioning theory.

Briefly, in the former approach, the compound's maximum desorption temperature ($T_{max}$) in the FIGERO-CIMS can be related to their enthalpy of vapourisation (Lopez-Hilfiker et al., 2014). After a series of calibrations with compounds with known vapour pressures ($V_p$), such as the homologous series of polyethyle glycols (PEG) (Krieger et al., 2018), a $T_{max}-V_p$ relation can be established (Fig. S8). In such way, the vapour pressure of each individual compound could be estimated based on their $T_{max}$, and subsequently that can be converted to effective saturation concentration, expressed in logarithmically spaced bins ($\log_{10}C^*$) as follows (Pankow, 1994):

$$log_{10}C^* = log_{10}\left(\frac{MW_i \times V_{p,i} \times \alpha_i}{R \times T} \times 10^6\right) \text{ (2)}$$

where, $MW_i$ is the molecular weight of each individual, $i$, compound (g mol$^{-1}$), $V_{p,i}$ is the estimated vapour pressure of each compound from the $T_{max}$ vs $V_p$ calibrations (Pa), $\alpha_i$ is the activity coefficient (here assumed as 1), $R$ is the universal gas constant (m$^3$ Pa mol$^{-1}$ K$^{-1}$) and $T$ is the temperature (K).

It should be noted that recent studies have showed that the filter loading as well as the delivery of the calibrant in the FIGAERO (e.g., syringe injection vs atomising solutions) may affect the $T_{max}$ and thereby the volatility estimations (Ylisirniö et al., 2021). Our instrument was rebuilt after the campaign and consequently calibrations at only one filter loading with the syringe method were available. The implications of this to our results is discussed in the Section 4.1.

In the latter approach, the fraction of each identified product in the particle phase (i.e., partitioning coefficient, $f_{p,i}$) was calculated based on:

$$f_{p,i} = \frac{Particle_i}{Particle_i + Gas_i} \quad (3)$$

where, $Particle_i$ and $Gas_i$ are the integrated signals of each compound in the particle and gas phase, normalised to their sampling volume (i.e., ions m$^{-3}$), respectively. The effective saturation concentration of each product ($\log_{10}C^*_i$) 360 was calculated based on the absorptive partitioning theory using the total absorptive mass concentration ($C_{tot}$) obtained by the HR-AMS and the $f_{p,i}$ as follows (Donahue et al., 2006):

$$\log_{10}C^*_i = \log_{10}\left(\left(\frac{1}{f_{p,i}} - 1\right) \times C_{tot}\right) \quad (4)$$

Owing to the considerable variability in the sum thermograms (i.e., sum of the thermograms of each individual 365 elemental formula) in the $o$-cresol experiments (Fig. S3), the volatilities derived from the FIGAERO-CIMS are shown here as averages from all the available experiments in each system (see Sections 3 and 4).

# 3. Results

## 3.1. Gas and particle phase composition

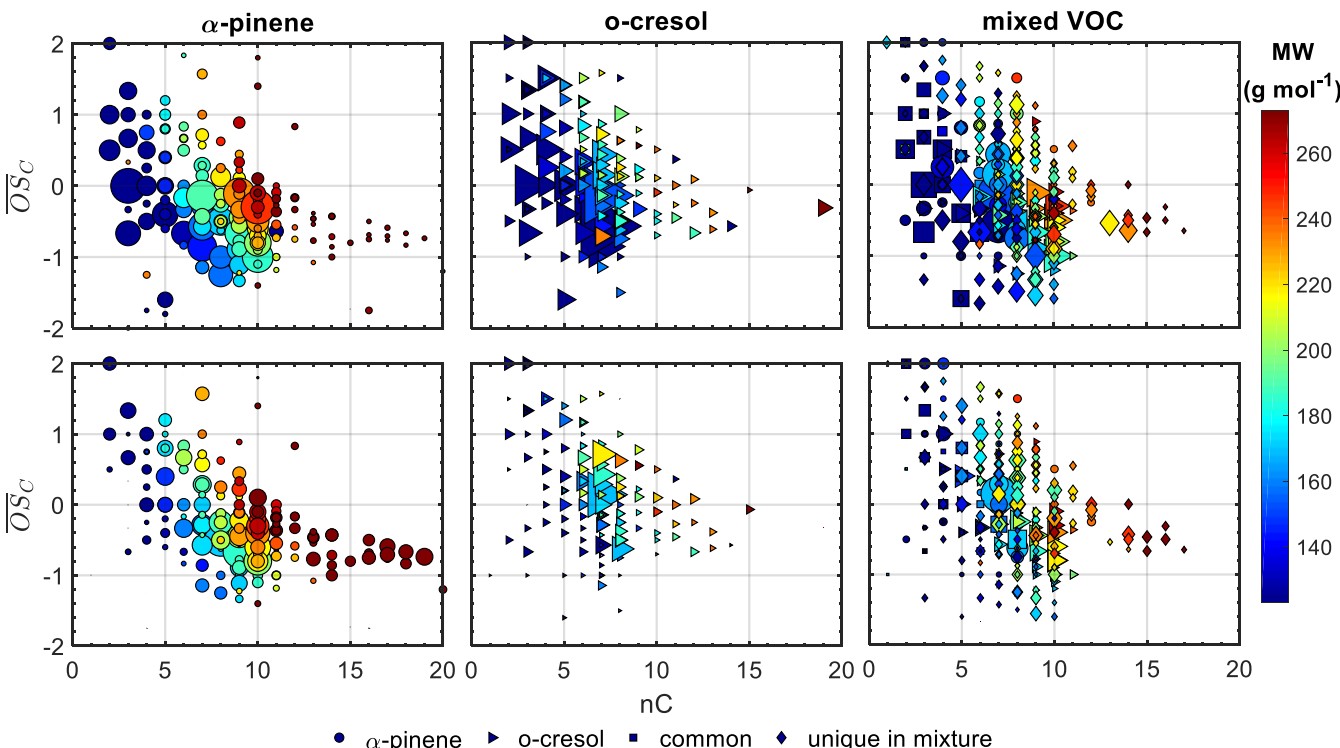

**Figure 1: Gas (top panels) and particle (bottom panels) average carbon oxidation state ($\overline{OSc}$) against carbon number (nC) for all the products identified in characteristic experiments for each system (Exp. No. 2, 6 and 10; Table 1). In all the experiments the products from each precursor are represented by different symbols, while in the mixed VOC experiment, the products that were common and unique to the mix are also shown. The symbols are sized based on the square root of their signal intensity and are coloured based on their MW.**

Figure 1 shows the average carbon oxidation state ($\overline{OSc}$) against the number of carbon atoms (nC) of for all the products identified in characteristic experiments for each system (Exp. No. 2, 6 and 10; Table 1). In the mixed VOC experiment, the products that were corresponding to either precursor, the common and those that were classified as unique to the mixture are indicated with different symbols. Note that the $\overline{OSc}$ was calculated as $\overline{OSc}=2\times$O:C – H:C, where H:C is the oxygen to hydrogen ratio, according to Kroll et al. (2011). Although Priestley

et al. (2021) has shown that there is some uncertainty in the calculations of $\overline{OSc}$ for the N-containing compounds, for simplicity this effect was neglected here.

In all systems and in both gas and particle phases, the observed increasing $\overline{OSc}$ with the decrease of the nC shows evidence of processes deriving from radical initiation, propagation and termination reactions (Hallquist et al., 2009). However, each individual precursor system exhibited distinct features. For example, in the $\alpha$-pinene single precursor system, most of the products identified by the FIGAERO-CIMS both in the gas and particle phase had the same or fewer carbon atoms than the parent molecule (81 and 73% of the total signal in each phase, respectively; for clarity see Fig. S1). More specifically, in the particle phase, the highest contribution was found for products with nC=10 having $\overline{OSc}$ between -1.1 and 0.1, while in the gas phase for products with nC=7 and $\overline{OSc}$ =-1.1−1. In contrast, in the $o$-cresol single precursor system, the vast majority of the identified products in either phase showed a narrow distribution centred around the carbon number of the precursor (ie., nC=7; Fig.S1). This can be associated with the large signal contribution of two individual products in each phase, assigned as $C_7H_8O_2$ (50%) and $C_7H_7NO_4$ (36-40%) in the gas and particle phase, respectively (Fig. S9). These products are likely result of H abstraction and subsequent OH addition and nitration, respectively, suggesting that they are early generation products. The high contributions of these early generation products can be associated with the fact that the VOC precursors were not fully consumed within our experimental timeframes (Fig. S10), likely due to the high VOC:OH. As a result, it may be expected that they continuously formed throughout the duration of the experiment. Alternatively, the strong signal contributions of these products might indicate that our technique is particularly sensitive to these products. In the particle phase of both single precursor systems, a greater fraction of the signal is associated with products having more carbon atoms than each precursor, compared to the respective gas phase contributions.

In both gas and particle phases in the mixed VOC system, the largest fraction of the total signal was found for products with nC=7 and $\overline{OSc}$ in the range of ~-0.9−1.3. In the gas phase, the remaining signal was distributed in the region of nC=4−10 and $\overline{OSc}$ = -1−2, with stronger contributions at nC=5 with $\overline{OSc}$ = -0.8−1.4 and at nC=10 with $\overline{OSc}$ = -1.1−0.5. Correspondingly, in the particle phase, the products with nC=8 with $\overline{OSc}$ = -1.2−1.3 and nC=10 with $\overline{OSc}$ =-1.1−0.5 contributed considerably to the total particle phase signal. Unlike the single precursor systems, the contributions of products with larger than 10 carbon atoms and $\overline{OSc}$ =-0.6−0.5 were not substantially different between the gas and particle phases (Fig. S1).

Most of the products that were common between all systems had low carbon numbers (nC<5) and varying $\overline{OSc}$ (-

0.6−2.5), accounted for a higher fraction of the total gas than the particle phase signal (18% vs. 8%; Fig. S1). In both phases, $o$-cresol products appeared to have higher contributions to the total gas/particle signal of the mixed VOC system than those deriving from the $\alpha$-pinene with the majority of the signal being dominated by products with 7 carbon atoms and $\overline{OSc}$ = -0.5−0.7, similarly to the single precursor system. The products that were classified as unique products of the mixture accounted for approximately the same fraction in both phases and were mostly

found in the space of nC=5-10 and $\overline{OSc}$ = -1.5−1.6, with appreciable fractions in the nC>10 and $\overline{OSc}$ = -0.6−0.5 range.

Figure 2 shows the average (± 1σ) O:C ratio of all the products identified in each system from the FIGAERO-CIMS in the particle phase, weighted to the contribution of each product to the total particle phase signal, along with O:C ratio derived from the HR-AMS. Both HR-AMS and FIGAERO-CIMS showed comparable results, with

the O:C being highest in the $o$-cresol system (0.68 vs 0.56, respectively), followed by the mixed VOC (0.47 vs 0.49, respectively) and the $\alpha$-pinene system (0.35 vs 0.45, respectively). Although N:C ratios might provide similarly useful information, their quantification with FIGAERO-CIMS and HR-AMS has proven challenging. Particularly, the N:C quantification in the FIGAERO-CIMS can be affected by the differential sensitivity of the instrument (Iyer et al., 2016), while for the HR-AMS, the extensive fragmentation, which is not readily

characterised for the N-containing fragments (Farmer et al., 2010), can affect the estimated N:C.

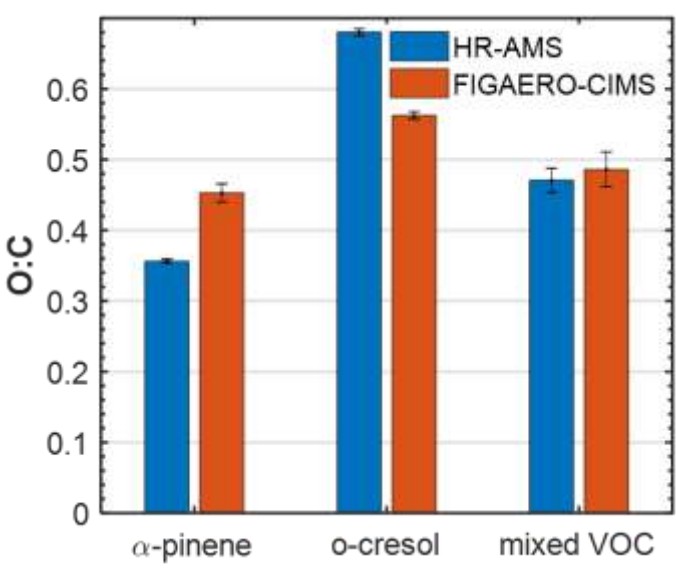

**Figure 2: Average (± 1σ) oxygen to carbon (O:C) ratio for all the products identified in the particle phase from the FIGAERO-CIMS and the HR-AMS for each system.**

## 3.2. SOA particle volatility

### 3.2.1. Insights on the SOA particle volatility from the FIGAERO-CIMS

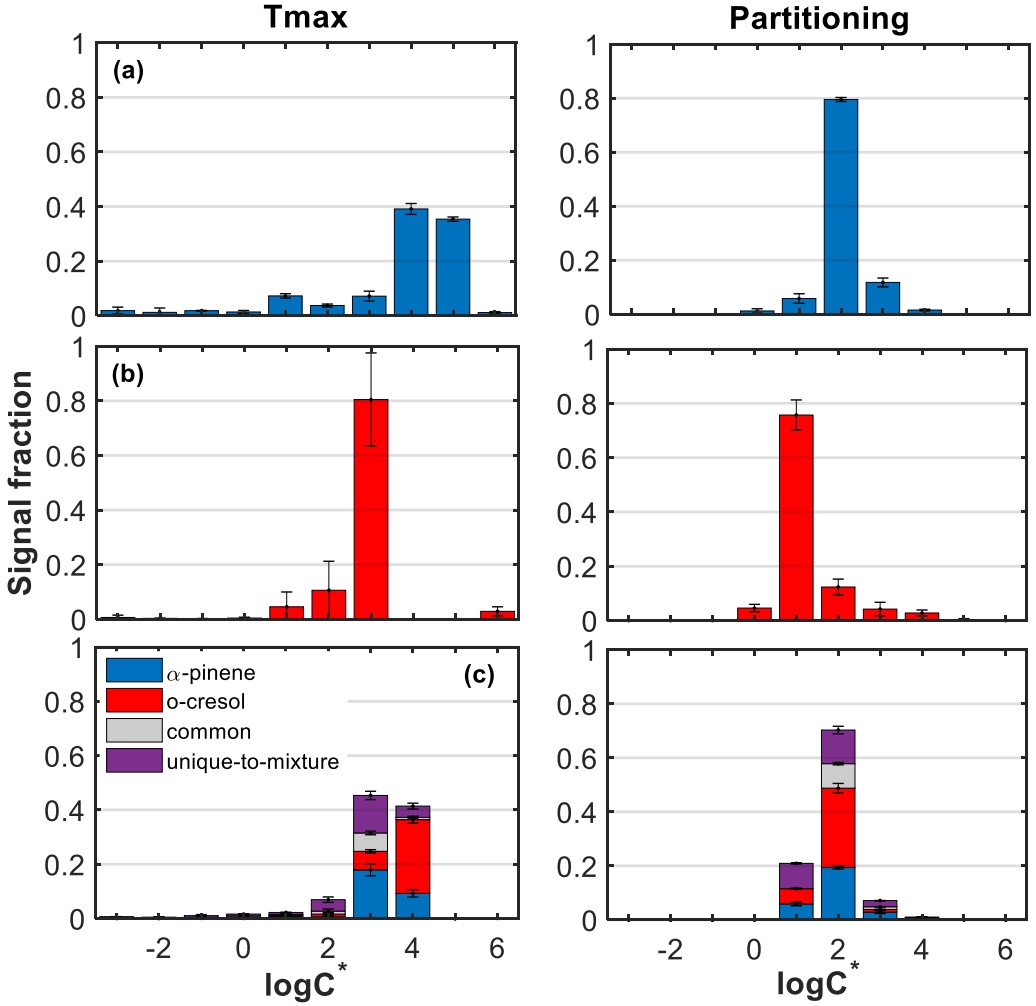

**Figure 3: Average (±1σ; n=2) volatility distributions as a function of the FIGAERO-CIMS total particle phase signal in the (a) α-pinene, (b) o-cresol and (c) mixed VOC systems using the calibrated $T_{max}$ (left panels)**
**and partitioning approaches (right panels). The bars in the mixed system are separated to show the**

**contributions from the products of either precursor, the common found in all systems and those that were found unique in the mixture.**

Figure 3 shows the retrieved average (±1σ; *n*=2) volatility distributions from the products identified in the
FIGAERO-CIMS in each system using the calibrated $T_{max}$ and the partitioning approaches. The distributions in the mixed VOC system (Fig. 3c) show the contributions from the products of either precursor, the common products and the products unique to the mixture in each volatility bin. The material with $-3 \leq \log_{10}C^* \leq 0.5$ can be characterised as low volatility (LVOC), though it should be noted that that the $\log_{10}C^*=-3$ bin here contains also compounds with lower volatility such as extremely LVOC (ELVOC). The material with $\log_{10}C^*$ between -0.5 and 2.5 as semi-
volatile (SVOC) and that with $\log_{10}C^*>2.5$ as intermediate volatile (IVOC) (Donahue et al., 2012).

The SOA particle volatility using the calibrated $T_{max}$ approach showed that the vast majority of the particle phase FIGAERO-CIMS signal was attributed to products having volatility in the IVOC range whereas using the partitioning approach, the largest fraction of the signal was attributed to the SVOC range. Characteristically, using the calibrated $T_{max}$ approach the split between LVOC, SVOC and IVOC was 5, 13, and 82% for the *α*-pinene
system, 2, 2 and 96% for the *o*-cresol system and 2, 14 and 84% for the mixed VOC system. In contrast, using the partitioning approach, the respective fractions for the LVOC, SVOC and IVOC was 0, 88 and 12% in the *α*-pinene system, 0, 95 and 5% for the *o*-cresol system and 0, 90 and 10% for the mixed VOC system.

In more detail, the volatility distribution in the *α*-pinene system from the calibrated $T_{max}$ approach showed strong contributions of products having $\log_{10}C^*=4-5$, sharply decreasing to products with lower volatility ($\log_{10}C^*<4$).
The volatility retrieved from the partitioning approach in that system showed that the largest fraction of the signal (~80%) had $\log_{10}C^*$ between 2 and 3. In contrast, the largest fraction of the particle phase signal in the *o*-cresol system appeared to be centred in the $\log_{10}C^*=2$ using the calibrated $T_{max}$ approach (~80%) and in the $\log_{10}C^*=1$ using the partitioning approach (~76%). The volatility distribution derived from either approach appeared to be particularly narrow that can be associated with high particle phase signal contributions of the $C_7H_7NO_4$ in that
system (Fig. S9).

The volatility distribution of the mixed VOC system using the calibrated $T_{max}$ approach appeared to be largely composed of products having $\log_{10}C^*=3-4$, with slightly higher contributions in the $\log_{10}C^*=3$ bin. The SOA volatility in the mixed system derived from the gas-to-particle ratio of the species had strong signal contributions

of compounds with $\log_{10}C^*=2$, followed by those with $\log_{10}C^*=1$. The products from the photo-oxidation of $o$-cresol appeared to account for the largest fraction of the particle phase signal in the $\log_{10}C^*=4$ and 2, in the $\log_{10}C^*$ space retrieved using the calibrated $T_{max}$ and partitioning approaches, respectively. Correspondingly, those derived from the $\alpha$-pinene had higher contributions in the $\log_{10}C^*=3$ and 1, respectively, while the unique products of the mixture contributed substantially in the $\log_{10}C^*=2-3$ and $1-2$, respectively. The products that were found in all systems (i.e., common) showed their highest signal contributions in the $\log_{10}C^*=3$ and in the $\log_{10}C^*=2$ when their $\log_{10}C^*$ was derived from the calibrated $T_{max}$ and partitioning approaches, respectively.

### 3.2.2. Insights on the SOA particle volatility from the TD-AMS

The estimated SOA particle volatility distributions ($\pm$ retrieval error) from the TD measurements for three characteristic experiments (Exp. No. 2, 6 and 10; Table 1) are presented in Figure 4, expressed in the VBS framework. The bin with $\log_{10}C^*=-3$ may also include the less volatile material (extremely low volatility organic compounds) and similarly, the bin with $\log_{10}C^*=2$ may also include more volatile material. The volatility distribution of the SOA formed in the $\alpha$-pinene experiment, indicates a higher contribution of species with $\log_{10}C^*\geq2$, compared to $o$-cresol and mixture experiments. Conversely, in the $o$-cresol and in the mixture experiments the bins with $\log_{10}C^*=\leq0$ exhibited relatively higher values compared to the single precursor $\alpha$-pinene system, suggesting a higher contribution of less volatile material. More specifically, the mass fraction of the LVOC and SVOC was 28 and 72% for the $\alpha$-pinene SOA, 39 and 61% for the $o$-cresol SOA and 37 and 63% for their mixture, respectively.

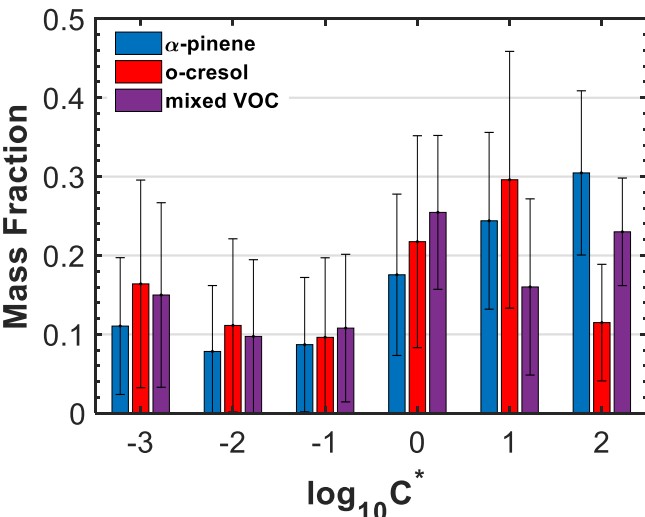


**Figure 4: SOA particle volatility distributions retrieved from the TD measurements for three characteristic experiments (Exp. No. 2, 6 and 10; Table 1). The error bars represent the uncertainty in the estimated values in each volatility bin.**

The volatility distribution inversion algorithm employed here (Karnezi et al., 2014) can also provide the enthalpy

of vapourisation and the mass accommodation coefficient for each of the experiments investigated. The enthalpy of vapourisation was found to be higher in the $\alpha$-pinene experiment (120 kJ mol$^{-1}$) and lower at the $o$-cresol (106 kJ mol$^{-1}$). This is qualitatively consistent with the larger molecular weight of the molecules in the $\alpha$-pinene SOA (Fig. 1). The mass accommodation coefficients were above 0.1 in both systems (0.36 for the $\alpha$-pinene and 0.23 for the $o$-cresol SOA; Fig. S11) suggesting only small resistances to mass transfer in the temperature range of the TD

(Riipinen et al., 2010;Saleh et al., 2011). The respective values for the mixed VOC experiment were found to be intermediate of those for the individual single-precursor systems (109 kJ mol$^{-1}$ and 0.27, respectively). Nonetheless, due to the corresponding uncertainties which were $\pm$ ~20% for the enthalpy of vapourisation and much higher for the mass accommodation coefficient, these differences may not reflect the reality.

The relationship between the O:C and the MFR can provide indirect additional information about the volatility of

the species in each system. Figure 5 shows the O:C enhancement ratio (i.e., O:C at the TD line divided by the corresponding O:C at the BP line) as a function of the particle MFR for a number of experiments in each system. The increasing O:C enhancement ratio as the MFR decreases in all systems is indicating that as the temperature in

the TD increases, the remaining SOA particles are less volatile and more oxygenated, consistent with previous observations in chamber experiments (e.g., Huffman et al., 2009). All systems appeared to behave similarly at low

TD temperatures (<40 °C), with the O:C enhancement ratio being between 1 and 1.04 and MFR>80%, implying the existence of products with similar degree of oxygenation and similarly high volatility. However, at higher TD temperatures (>40 °C or MFR<0.5), each system showed distinctive features. The SOA particles from the *o*-cresol system showed the highest O:C enhancement ratio, reaching up to 1.16, followed by those from the mixed VOC and the *α*-pinene system which exhibited similar maximum values (1.097 and 1.091, respectively). More

specifically, the slope between the highest O:C enhancement ratio and the MFR was twice as high in the *o*-cresol compared to *α*-pinene systems, whereas that of the mixed system was approximately half-way between. These results indicate that the less volatile SOA particles from the *o*-cresol system (MFR≤0.5) are likely more oxygenated compared to those of the *α*-pinene system. It should be noted however that in the *o*-cresol single system, the O:C enhancement ratio in this range showed high variability among the experiments, suggesting that it may not be

substantially different than that of the mixed VOC system.

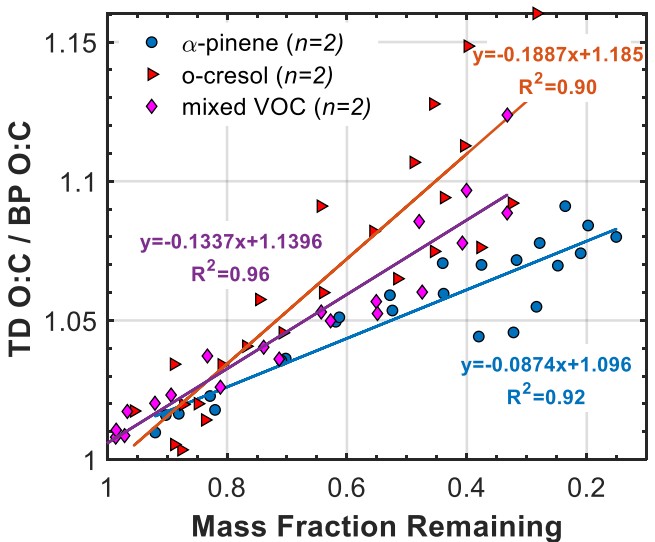

**Figure 5: O:C enchantment ratio (i.e., O:C at the TD line divided by the corresponding O:C at the BP line) as a function of the particle MFR in each system investigated.**


**4.  Discussion**

## 4.1. Comparison of the SOA particle volatility estimation methods

In this work, the SOA particle volatility distribution has been estimated using two independent instrumental techniques, the TD-AMS and the FIGAERO-CIMS. The TD-AMS technique can be used to derive volatility estimations for the bulk of the SOA particles, whilst the FIGAERO-CIMS is able to detect a subset of the total SOA particles at varying sensitivities (Lee et al., 2014;Lopez-Hilfiker et al., 2014). In the latter technique, the estimation of the particle volatility is possible through several different approaches (e.g., Stark et al., 2017). Here, based on the available information, we estimated the SOA particle volatility in the VBS framework using a series of explicit $T_{max}-V_p$ calibrations as well as based on the gas to particle ratio of the compounds identified.

Substantial discrepancies were found between the two methods used to derive the $log_{10}C^*$ of the compounds identified by the FIGAERO (Fig. 3), with more than one order of magnitude difference. The estimated SOA particle volatility from the calibrated $T_{max}$ method was significantly higher compared to that estimated using the partitioning approach. The particularly high volatility estimated with this approach can be attributed to the obtained $T_{max}-V_p$ calibrations and the $T_{max}$ distribution of the compounds in each system (Fig. S8 and S12). More specifically, the vast majority of the compounds identified by the FIGAERO exhibited $T_{max}<100°C$ which, based on our calibrations, resulted in $V_p> 10^{-3}$ Pa and thereby high estimated $log_{10}C^*$. The series of $T_{max}-V_p$ calibrations were conducted in this study using the syringe method. Recently it was shown that this has the potential to shift the $T_{max}$ of the compounds to substantially higher values than calibrations using atomised solutions, which may represent more closely the evaporation of the particles in the instrument (Ylisirniö et al., 2021). Consequently, the particularly high SOA particle volatility estimated with this method here might be attributed to the selected delivery of the calibrants on the FIGAERO. However, it should be further noted that an appreciable fraction of the total integrated particle signal lies at higher desorption temperatures (Fig. S3) suggesting that a considerable fraction of the ions is observed at the long "tails", yet with their maximum peak being at lower temperatures. There are several reasons that may lead to this tailing effect, such as the existence of multiple isomers and/or fragments from the thermal decomposition of oligomers within each HR peak that have lower volatilities and/or a change in the evaporation behaviour due to the changing composition and thereby the activity coefficient of the sample (Schobesberger et al., 2018). There have been attempts to resolve any potential isomers/decomposition products using multi-peak fitting methods (Lutz et al., 2019) or explain the whole thermograms using positive matrix factorisation (Buchholz et al., 2020); however there is some ambiguity in the implementation and interpretation of such approaches.

In contrast, the SOA particle volatility estimated from the partitioning approach was particularly narrow (Fig. 3). The volatility distributions estimated using this approach is limited by the detection limit of the measurable gas to particle ratios by the FIGAERO (and thereby the calculated $f_p$), where in this study, the measured $f_p$ (Fig. S13) in conjunction with the measured $C_{tot}$ (Table 1), can result in differences in the $\log_{10}C^*$ of 1−2 orders of magnitude in each system. Similarly narrow volatility distributions from the partitioning compared to other volatility

estimation approaches from the FIGAERO have been also observed previously in ambient aerosols and was attributed to signal to noise limitations that result in confined $f_p$ value ranges (Stark et al., 2017).

The SOA particle volatility distributions estimated from the FIGAERO-CIMS using either the calibrated $T_{max}$ or the partitioning methods were narrower and they exhibited considerably higher signal fractions of more volatile material than those estimated by the TD-AMS (Fig. 3 and 4). The SOA particle volatility retrieved from the

partitioning approach showed its highest signal contributions at the same $\log_{10}C^*$ bins as those observed by the TD-AMS technique, while those retrieved by the calibrated $T_{max}$ approach were more than 1 order of magnitude higher. The particularly high particle volatility obtained from the FIGAERO-CIMS using the calibrated $T_{max}$ approach appears most unrealistic, given that compounds with such high volatilities (i.e., $\log_{10}C^*>3$) should be predominantly in the gas phase and cannot explain their large fraction in the particle phase (Fig. S13) at our SOA

particle mass loadings. The similarity in the magnitude of the volatilities obtained from the partitioning approach with the independent TD-AMS technique might indicate that is more realistic. The SOA particle volatility retrieved by the TD-AMS technique has previously shown to have good agreement with the sum thermograms and is likely closer to reality (Stark et al., 2017). Similarly, here, converting the thermograms of each individual elemental formula to volatility distributions by expressing the desorption temperatures to $\log_{10}C^*$ based on our $T_{max}-V_p$

relation, we obtained comparable volatility distributions to those from the TD-AMS technique (Fig. S14). This suggests that sum thermograms obtained by the FIGAERO-CIMS might lead to more representative qualitative volatility estimations compared to the other methods adopted here. With such an approach however, the volatility of each individual assigned elemental formula is represented as a distribution instead of a single $\log_{10}C^*$ value and any potential artefacts from the thermal decomposition of oligomers are neglected.

The above analysis highlight that the SOA particle volatility estimation from the FIGAERO-CIMS is challenging and the results obtained might be limited based on the selected estimation approach along with any associated assumptions. Clearly, further work is needed in order to better understand the evaporation of the organics in the FIGAERO-CIMS and the interpretation of the obtained results. Such an investigation is beyond the scope of the

current study. However, both techniques along with the adopted volatility estimation methods, showed that the

SOA particles from the photo-oxidation of $\alpha$-pinene were composed by more volatile material compared to those derived from the photo-oxidation of $o$-cresol, while those derived from the photo-oxidation of the mixture appeared to be somewhere in-between. This can be further supported by the "lower level" data obtained by each technique that is the MFR in the TD (Fig. S4), the sum thermograms/$T_{max}$ distribution and the $f_p$ in the FIGAERO (Fig. S3, S12 and S13, respectively). Therefore, despite challenges in quantifying the $\log_{10}C^*$, the broad agreement in the

volatility trends between all the methods provide some confidence that the results presented capture the bulk SOA particle volatility changes between the systems.

Considering the above, for the remaining manuscript, the volatility distributions estimated using the partitioning approach from the FIGAERO will be used to link the chemical composition with the volatility.

**4.2.   Linking the gas and particle chemical composition to SOA volatility**

The gas and particle phase composition, as can be obtained by the FIGAERO-CIMS, can provide indications about the systems chemical behaviour and thereby the potential effect on SOA volatility. Oligomerisation is a process that increases the products' molar mass decreasing their volatility while tending to maintain their $\overline{OSc}$. Functionalisation also decreases a compound's volatility by the addition of oxygenated functional groups and thus

increasing their $\overline{OSc}$. Fragmentation (i.e., cleavage of carbon-carbon bond) will increase component volatility due to the decrease of the product's molar mass, though in some cases the fragmented products are subsequently functionalised (e.g., $O_2$ addition after the scission of a double bond) leading to a decrease of their vapour pressure, with the net effect on vapour pressure not immediately predictable. It should be noted that the selected VOC precursors in this study have fundamentally different structures; $\alpha$-pinene is a bicyclic alkene while $o$-cresol is an

aromatic molecule. In the $\alpha$-pinene, one C=C bond can be broken by $O_3$ without the loss of C, while the stability of the aromatic ring makes $o$-cresol practically unreactive towards $O_3$. Once however the aromaticity is broken, fragmentation reactions by $O_3$ will occur creating smaller molecules. Based on the chemical information obtained by the FIGAERO-CIMS and the volatility distribution estimations from both techniques employed, in this section, we will attempt to provide links between the systems' chemical composition and volatility.

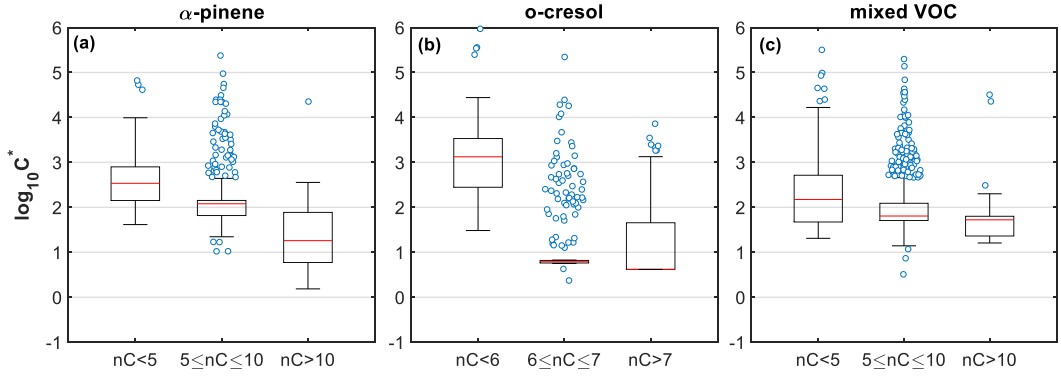

**Figure 6: Boxplots of the estimated volatility ($\log_{10}C^*$) of the products identified in the (a) *α*-pinene, (b) *o*-cresol and (c) mixed VOC experiments (exp. no. 2, 6 and 10, respectively), weighted to their contributions to the total particle phase signal and grouped based on their number of carbon atoms. Red lines represent the median values, each boxes' upper and lower limits represent the 75[th] and 25[th] percentile, respectively, while the errorbars represent the minimum and maximum values. The open circles show the individual compounds that were classified as outliers being outside the interquartile range times 1.5.**

### 4.2.1. Single precursor systems

The FIGAERO-CIMS results described above indicate that fragmentation is a major process occurring in all the systems due to the significant fraction of products having lower carbon numbers than their precursors (Fig. 1; Fig. S1). Of course, it should be noted that the interpretation of FIGAERO data is dependent on the components that are readily accessible using the chemical ionisation scheme. The iodide CIMS (I⁻-CIMS) has a wider range of sensitivity than acetate or $NO_3^-$ instruments although each detects a subset of those components that are present at varying sensitivities (Riva et al., 2019). The I⁻-CIMS is efficiently detecting compounds in the IVOC and SVOC range with modest carbon and oxygen numbers while $NO_3^-$-CIMS is more sensitive towards compounds in the LVOC and ELVOC range with higher carbon and oxygen numbers, such as HOM (Priestley et al., 2021). Although the development of the FIGAERO coupled to I⁻-CIMS enabled the detection of particle phase compounds in the LVOC/ELVOC range (Isaacman-VanWertz et al., 2017), here, we were not able to reliably identify substantial fractions of compounds with volatility in this range using either the calibrated $T_{max}$ or the partitioning approach, in contrast to the TD measurements (Fig. 3 and 4). As discussed in Section 4.1., the observation of lower volatility

species is likely related to selected method to retrieve the SOA particle volatility from the instrument and any associated assumptions and limitations. Nonetheless, here, the general agreement in the volatility trends between the FIGAERO-CIMS and the TD-AMS might indicate that the FIGAERO-CIMS can capture the changes in the SOA particle volatility across the different systems. Therefore the quantification of the $\log_{10}C^*$ might be limited from the approach presented here but may be able to provide representative trends between the systems.

Given the above, the higher fraction of compounds with lower carbon number than the precursors in both gas and particle phases in the α-pinene system suggests that fragmentation is more pronounced compared to the other systems (Fig. 1; Fig. S1), in agreement with previous findings (Eddingsaas et al., 2012b;Eddingsaas et al., 2012a) and thus potentially explaining the higher fraction of more volatile products with $\log_{10}C^*=2$ compared to the other systems (Fig. 3 and 4). Meanwhile, the appreciable contribution of molecules with high MW (>260 g mol$^{-1}$), nC (nC>10) and moderate $\overline{OSc}$ (-1−-0.5) and their stronger presence in the particle than in the gas phase (Fig.1 and S1; ~6.5 vs. 1%, respectively), shows that gaseous and/or condensed phase oligomerization/dimerisation reactions produced lower volatility products (Fig. 3 and 4). This can be further observed from the Figure 6a, where the weighted median volatility for the nC>10 compounds was $\log_{10}C^*=1.23$ compared to those with nC between 5 and 10 that was $\log_{10}C^*=2.08$. The higher median volatility ($\log_{10}C^*=2.35$) observed in the heavily fragmented products (nC<5; MW<180 g mol$^{-1}$) suggests that they had higher fractions in the gas compared to the particle phase. Their broad $\log_{10}C^*$ distribution and their relatively high $\overline{OSc}$ suggests that they might originate from various pathways, including thermal decomposition processes in the FIGAERO (Thornton et al., 2020). It should be noted however that these products did not contribute substantially to the total gas and particle phase signals (~7.97 and 2.07%, respectively) thus their effect on the overall SOA particle volatility from the FIGAERO-CIMS is expected to be equivalently low.

In o-cresol experiments, the dominance of products in both gas and particle phases having 7 carbon atoms suggest that the OH addition pathway that mainly leads to ring-retaining products, likely plays a major role in the o-cresol photo-oxidation, consistent with previous observations (Olariu et al., 2002;Schwantes et al., 2017). In the nC=7, the identified products with higher degree of functionalisation ($\overline{OSc}$>0) showed clearly higher contributions in the particle than in the gas phase (Fig. 1), implying that they are of relatively lower volatility. Further, the increased O:C and O:C enhancement ratio observed in this system (Fig. 2 and 3), in conjunction with recent findings suggesting that the oxidation of aromatic compounds proceeds rapidly through multiple generations of oxidation (Garmash et al., 2020;Wang et al., 2020), supports the observed higher contributions of lower volatility material

($\log_{10}C*=1$) in that system. It should be noted that the particularly narrow volatility distribution of the nC=6−7 in this system and the large fraction of outliers observed (Fig. 6b), can be related to the dominance of the $C_7H_7NO_4$ (Fig. S9). At the same time, the observed presence of molecules with higher nC than the precursor (nC>7), having varying $\overline{OSc}$ ( $\overline{OSc}$ =-0.8−0.8; Fig. 1) and low volatilities (median $\log_{10}C*=0.62$), shows evidence of oligomerisation/dimerisation reactions. The considerable presence of products with lower than six carbon atoms,

indicates that the oxygen addition pathway, leading to bicyclic intermediates that subsequently decompose to smaller multifunctional compounds might be responsible (Schwantes et al., 2017). This pathway can have complex effects in both directions for the resultant SOA volatility while the potential thermal decomposition of products in the FIGAERO, can possibly explain the large range of volatilities observed (Fig. 6b).

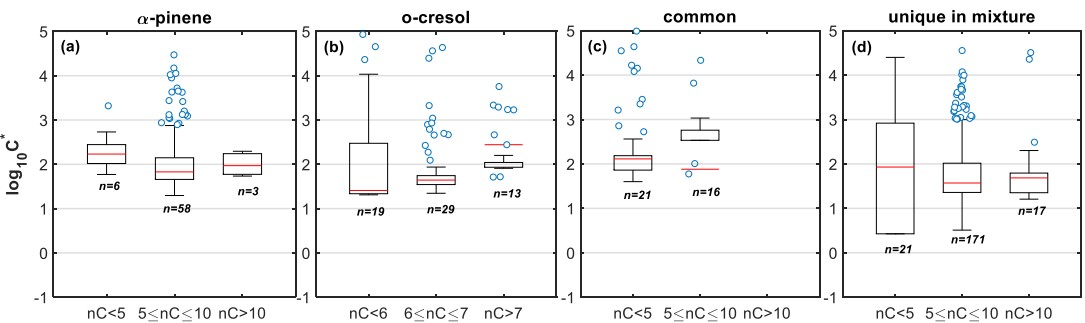

**Figure 7: Boxplots of the estimated volatility ($\log_{10}C*$) of the products identified in the mixture and were classified as products of (a) $\alpha$-pinene, (b) $o$-cresol, (c) common and (d) unique to the mixture, weighted to their contributions to the total particle phase signal and grouped based on their number of carbon atoms. Red lines represent the median values, each boxes' upper and lower limits represent the 75th and 25th percentile, respectively, while the errorbars represent the minimum and maximum values. The open circles**

**show the individual compounds that were classified as outliers being outside the interquartile range times 1.5.**

### 4.2.2. Mixed VOC system

Exploring the links between the SOA composition and volatility in a mixed system may provide evidence about

the potential of molecular interactions to alter the SOA properties. The products identified in the mixed system

here may comprise i) products derived from the oxidation of either precursor, ii) cross-products formed in the particle or gas phase (with sufficiently low volatility to partition to the particle phase) and iii) individual precursor products resulting from the alteration of the oxidation mechanism of each precursor due to the interactions. It should be noted that our formula separation technique (See Section 2.4.1; Fig. S7) is unable to identify any new individual precursor products from (iii) that are not listed in the MCM (see 2.4.) and would classify them as unique products of the mixture.

The products deriving from $\alpha$-pinene and $o$-cresol account for the largest fraction of the gas and particle phase signal of the mixture at their respective nC (nC=10 and 7; Fig. 1 and S1). The oxidation products of $\alpha$-pinene with modest number of carbon atoms (nC=5-10) found in the mixture appear to have similar volatility compared to the products in the same range carbon atoms in the single precursor experiments (median $\log_{10}C^*$=~1.83 vs. 2.08; Fig 6a and 7a). On the other hand, the o-cresol's products with nC=6-7 in the mixture appeared to have higher volatility compared to those at the same range in the single precursor experiments (median $\log_{10}C^*$=1.64 vs 0.82, respectively; Fig 6b and 7b). A potential explanation for this discrepancy is the higher SOA mass loadings and thereby the total absorptive mass, in the mixture experiments compared to the single $o$-cresol (Table 1) that could have favoured the partitioning of the more volatile species. However, the difference in the total mass between the two experiments after the 5$^{th}$ hour was relatively small (~20-30%; Fig. S15) to observe differences in the volatilities expressed in logarithmic bins. Alternatively, this difference could have been caused due to sampling/measurement artefacts and/or the formation of isomers in the mixture with different volatility. Indeed, in this experiment, the $C_7H_7NO_4$ exhibited much lower $f_p$ compared to the $o$-cresol experiment (~0.67 ±0.06 vs. 0.85 ±0.04, respectively), suggesting that the latter might be a potential cause for this discrepancy.

The products of the mixture that were classified as common are molecules that were found in all systems (i.e., single precursor and mixture) with the same assigned elemental formula. These products appeared to account for the largest fraction of gas and particle phase signal in the nC<5 range. These extensive fragmentation and/or thermal decomposition products appeared to have similar volatilities across all carbon numbers ($\log_{10}C^*$~2; Fig. 7c), and their relatively low contribution to the total particle phase signal (~5%) suggest that they had equally low effect in the observed volatilities.

Interestingly, the formation of products from each individual precursor with higher number of carbon atoms (nC>7 and nC>10 for $o$-cresol and $\alpha$-pinene, respectively) appeared to be substantially supressed both in the gas and particle phase of the mixture (Fig. 1, S1). Considering that these products were found to have lower volatilities

(median $\log_{10}C^*$<2; Fig. 6c), this suppression effect might partly explain the lower fraction of lower volatility products found in the mixture (i.e., $\log_{10}C^*$<1; Fig. 4). McFiggans et al. (2019) previously suggested that the potential suppression in the formation of $\alpha$-pinene-derived higher MW compounds when it was mixed with isoprene could increase the overall volatility and partly explain the observed reduced yields in that system. Here, our observations in a different system show evidence that the high MW product scavenging has the potential to increase

the overall volatility in mixed precursor systems.

At the same time, a wealth of products unique to the mixture were identified, having their highest contributions in the nC=5−10 and at varying $\overline{OSc}$ (-1.5−2; Fig. 1 and S1). The formation of these products, as well as those with nC>10, could be both attributed to processes described above for each individual VOC and/or due to the interactions between the resultant products (such as $RO_2$+R'$O_2$ termination reactions, with R and R' from $o$-cresol and $\alpha$-pinene

respectively). Their similar presence in the gas and particle phase (Fig. 1) further supports their measured moderate volatility (median $\log_{10}C^*$=1.57 and ~1.68 for the nC=5−10 and nC>10, respectively; Fig. 7d). Evidently, based on the above, the unique products of the mixture have lower volatility compared to the products of either precursor within the mixture, across all carbon numbers (Fig. 7). Neglecting the compounds with low number of carbon atoms (nC<5) that are contributing minimally to the total particle phase signal (~5%) and the common products

that our technique is unable to resolve, the lower volatility of the unique-to-the-mixture compared to the individual precursor's products in the mixture can be associated with their higher O:C, $\overline{OSc}$ and MW (Fig. S16). This suggests that the potential molecular interactions results in larger, more functionalised products that have volatilities at the lower end of the SVOC range, decreasing the overall volatility of the mixture.

Previously, Emanuelsson et al. (2013) observed an increase in the volume fraction remaining (and thus a reduction

on the volatility) in SOA formed from the mixing various anthropogenic and biogenic precursors reacting sequentially. Here, the volatility of the SOA particles in the mixed VOC system, both from the TD and the FIGAERO-CIMS, was found to be in-between the volatilities obtained in the $o$-cresol and $\alpha$-pinene single precursor systems (Fig. 3 and 4). Based on our observations, this behaviour can be linked with the suppression in the formation of higher MW and lower volatility products that increases the volatility of the mixture compared to the

single precursor experiments and simultaneously, the formation of unique products with moderate/low volatilities that decrease the volatility of the mixture.

It should be noted here, that in order to form adequate amounts of SOA mass required for offline analysis (discussed in separate subsequent manuscripts), particularly high VOC concentrations (and consequently $NO_x$) had to be used

(Table 1). Previous studies have showed that the ambient OH reactivity (estimated by measurements of OH lifetime) was between ~10 and 116 s$^{-1}$ (Whalley et al., 2016), while in our experiments was ~404 s$^{-1}$ (product of VOC concentration with reaction rate towards OH). In the ambient atmosphere, the OH reactivity might be affected by the presence of smaller compounds (such as CO, CH$_4$), in turn affecting the fate of RO$_2$ and the RO$_2$:HO$_2$. In our setup, the particularly high concentration used is expected to promote the RO$_2$-RO$_2$ cross reactions and supress the RO$_2$+HO$_2$ and RO$_2$+NO reactions (Peng and Jimenez, 2020;Schervish and Donahue, 2020), which are considered as dominant radical termination pathways in the atmosphere. Consequently, the increased presence of unique-to-the-mixture compounds due to the interactions might partly attributed to the selected experimental conditions. Nonetheless, our scope here is not to fully mimic the ambient atmosphere, rather than explore the potential of the molecular interactions in mixtures to alter the SOA particle volatility.

### 4.3. Interpretation of FIGAERO-CIMS data

As has been shown here and elsewhere (e.g., Lopez-Hilfiker et al., 2016b;D'Ambro et al., 2017), linkages between the chemical composition and the particle volatility are possible using the FIGAERO-CIMS, while the individual product's volatility can be inferred qualitatively (Wang and Hildebrandt Ruiz, 2018) or quantitatively within an uncertainty (e.g., Bannan et al., 2019). Here we show for the first time that the technique can be used to distinguish the SOA products deriving from the oxidation of multiple precursors in environmental chambers, while being able to assess the potential role of product interactions to alter the SOA particle volatility. However, some limitations may impact the results derived from the technique, such as the thermal desorption of the SOA collected in filters and the differential instrument sensitivity toward classes of compounds (Huang et al., 2018;Stark et al., 2017;Wang and Hildebrandt Ruiz, 2018;Lopez-Hilfiker et al., 2014).

We observed that the dominant $o$-cresol derived product(s) with elemental formula C$_7$H$_7$NO$_4$, exhibited considerably different maximum desorption temperatures between the single precursor and the mixture systems (74.3±3 and 59.5±2 °C). This resulted to a different estimation of its volatility using the calibrated T$_{max}$ approach between the two systems (log$_{10}$$C$*=3.1±0.2 and 4.0±0.1, respectively). The difference in the desorption profiles of the products in the FIGAERO can be impacted by the matrix and/or saturation effects (Wang and Hildebrandt Ruiz, 2018;Ylisirniö et al., 2021;Schobesberger et al., 2018). In these studies, a positive relation of the filter loading to the maximum desorption temperature has been previously reported. However, in these systems, the total particle mass and consequently the loading of the filter in the FIGAERO, was considerably higher in the mixture than in

the single precursor systems (Fig. S15) and consequently cannot explain these differences. The observed shift towards lower $T_{max}$ with increasing the filter loading could be possibly attributed to the formation of isomers in the mixture that had higher volatility. This can be possibly supported by the partitioning calculations where the calculated $f_p$ was lower in the mixture than the single precursor experiments (0.68±0.06 and 0.82±0.09, respectively) that correspondingly led to higher volatility estimates ($\log_{10}C^*=1.7\pm0.1$ and 1.0±0.3, respectively). Alternatively, the change in the SOA particle viscosity due to the mixing could have possibly affected the evaporation kinetics in the FIGAERO (Buchholz et al., 2019;Schobesberger et al., 2018). Supporting that, disregarding the uncertainty of our method, the mass accommodation coefficient estimated in the $o$-cresol was lower compared to that in the mixed VOC system (Fig. S11), suggesting, if anything, stronger mass transfer limitations (Schobesberger et al., 2018). Further, Ylisirniö et al. (2021) observed differences in the $T_{max}$ vs. $V_p$ calibrations between carboxylic acids and polyethylene glycols with the former that are less viscous, to desorb at lower temperatures. Therefore, it might be worthwhile to experimentally investigate the effect of the aerosol phase state/viscosity to the desorption profiles of the products in the FIGAERO.

Recently, Hammes et al. (2019) showed that the sum of the 10 highest product signals (and any potential isomers therein) accounted for ~50% of the total acetate FIGAERO-CIMS signal in limonene photo-oxidation experiments. Here, in all the systems examined we also find that >45% of the total signal of the FIGAERO-CIMS can be attributed to the top 10 elemental formulas identified (Fig. S9). Particularly, in the $o$-cresol experiments ~50% and ~40% of the total signal in the gas and particle phase respectively can be attributed to a single elemental formula ($C_7H_8O_2$ and $C_7H_7NO_4$, respectively; Fig. S9). These results may indicate either that the aerosol composition is dominated by those few products or that the technique is overly sensitive to these compounds or a combination of both explanations.

The sum thermograms (defined as the sum of the individual product's thermograms) or, as in this study, the calibrated $T_{max}$ and the partitioning approaches, have been widely used to infer the volatility of the aerosols (e.g., Lopez-Hilfiker et al., 2015;Huang et al., 2018). However, a large fraction of the total gas or particle phase signal is due to as few as 10 products and any potential isomers therein (out of hundreds or thousands typically found in the mass spectra) which in turn will largely define the shape of the volatility distributions. Figure 3 shows that the volatility distribution of the $o$-cresol experiment is narrow, partly owing to the dominant product(s) accounting for ~30-40% of the total particle phase signal. Figure 1 and S1 suggests that a large fraction of the total particle phase signal of the FIGAERO-CIMS in the mixed VOC system is ostensibly related to $o$-cresol products (33%) with the

signal from two elemental formulas dominating this fraction ($C_7H_7NO_4$ =26% and $C_8H_{11}NO_3$=3%; Fig. S9). Therefore, caution should be exercised when attributing the contributions of individual precursor products to the mixed precursor products as the expression of FIGAERO-CIMS results in terms of the signal fraction can be misleading.

Previous experimental and theoretical studies showed that the sensitivity of iodide CIMS towards nitrogen containing compounds with hydroxyl functional groups could be significant (Iyer et al., 2016;Lee et al., 2014). Considering that *o*-cresol oxidation largely yields OH- containing products (Schwantes et al., 2017), in conjunction with the observed particularly high contributions of N-containing products to the total signal in all *o*-cresol-containing experiments (Fig. S17), it is likely that differential sensitivity plays a role in this apparent dominance.

Interestingly however, each system exhibited comparable number fraction of products containing C, H, O and N atoms (Fig. S17b), suggesting either that the nitrogen-containing oxidation products of *o*-cresol actually contribute substantially to the SOA mass and/or that there is a significant differential sensitivity of the instrument towards molecules containing C, H, O and N. Because of the lack of information regarding the iodide CIMS sensitivity towards aromatic and nitrogen-containing compounds, this cannot be directly investigated here but should be an area for future research.

The SOA particle yield from the *o*-cresol experiments was found to be modest (14±3%), while that from the *α*-pinene considerably higher (27±4%; Fig. S18), suggesting that the observed domination of the *o*-cresol products in the mixed systems is unlikely. Considering the relative agreement in the volatility derived from the FIGAERO-CIMS and the TD and the analysis presented above (Sections 4.1-4.2), we can argue that the SOA particles in the mixture, are composed by the products of the higher yield precursor (i.e., *α*-pinene) with considerable contributions of the *o*-cresol derived products, as well as those deriving from the mixing of the two precursors.

## 5. Conclusions

Many climate/air quality prediction models express SOA volatility based on single-precursor experiments and neglecting the potential interactions of the molecules deriving from various precursors that are emitted in the atmosphere and lead to the SOA formation (Yahya et al., 2017;Tsimpidi et al., 2018). However, an increasing number of studies show that the results obtained under using single VOC precursor systems might not represent accurately the ambient atmosphere in which hundreds of VOC are oxidised simultaneously (McFiggans et al.,

2019;Shilling et al., 2019). Here, we comprehensively investigate the SOA composition and volatility upon mixing of $\alpha$-pinene and $o$-cresol under photo-oxidation conditions, using two independent techniques. Our experimental setup might not fully represent the real atmosphere owing the lack of smaller organic molecules (such as CO or CH$_4$) that may influence the radical termination pathways (Schervish and Donahue, 2020), however provides the grounds to investigate the potential of the molecular interactions to alter the SOA composition and properties in

mixtures.

The two volatility estimation techniques showed substantial discrepancies in the obtained $\log_{10}C^*$ values, highlighting the complexity in deriving the volatility distribution and the need for further studies to investigate the thermal evaporation of the organics. However they all showed similar trends; the SOA particle volatility from the photo-oxidation of $\alpha$-pinene had stronger contributions of higher volatility material compared to that from the $o$-

cresol, while SOA particle volatility of their mixture appeared to be in-between. This was further supported by the qualitative data that are used for the estimation of the volatility that is the MFR in the TD and the sum thermograms and the partitioning coefficient distribution in the FIGAERO-CIMS. The chemical information obtained enabled us to provide links between the chemical composition and volatility and showed that the extensive fragmentation occurred in the $\alpha$-pinene system probably resulted in the observed higher volatility SOA. On the other hand, the

majority of the particle phase products from the $o$-cresol had 6-7 carbon atoms, higher O:C and $\overline{OSc}$, potentially explaining their lower volatility. Importantly, our results first show that upon mixing $\alpha$-pinene and $o$-cresol there were two opposite effects; first the suppression in the formation of higher MW and lower volatility compounds derived from each individual precursor that increases the volatility of the mixture and second, the formation of unique products in the mixture that had lower volatility, higher $\overline{OSc}$, O:C and MW that decreased the volatility of

the mixture. The trade-off between these two effects could be determining the overall SOA particle volatility in conditions where multiple VOC precursors are reacting simultaneously, where in this case, was found to be in-between those measured in the single precursor experiments.

There is some ambiguity in interpreting the FIGAERO-CIMS data owing to the potential shift of the thermograms due to the different filter loadings and likely, the evaporation behaviour of the compounds in the FIGAERO, as

well as the a priori selectivity of the instrument towards certain products, defined by the instrument setup (Lee et al., 2014;Riva et al., 2019). The relative agreement however in the obtained volatility trends between our independent techniques might indicate that the FIGAERO-CIMS, at least qualitatively, could capture the SOA

volatility changes in mixed precursor systems and identify corresponding changes in the SOA chemical composition.

The results presented here call for further studies to explore the mechanisms that are driving these observations. Furthermore, studies conducted on various chemical regimes and using various oxidants might improve our understanding about the importance of molecular interactions in mixtures. The formation of new products in mixtures as well as the differential contribution of each precursor's product to the particle phase implies a range of potential changes on the SOA properties and thereby their subsequent interactions of the SOA with water vapour

and other particles and gases (Champion et al., 2019;Riipinen et al., 2015;Pankow et al., 2015). Further, the potential effect of mixing anthropogenic and biogenic products might significantly affect the predicted SOA radiative forcing (Gordon et al., 2016;Kelly et al., 2018).

**Data availability**

All the data used in this work can be accessed on the open dataset of the EUROCHAMP programme

(https://data.eurochamp.org/data-access/chamber-experiments/).

**Acknowledgements**

The Manchester Aerosol Chamber received funding from the European Union's Horizon 2020 research and innovation programme under grant agreement no. 730997, which supports the EUROCHAMP2020 research programme. AV acknowledges the Presidents Doctoral Scholarship from the University of Manchester and the

support from the Natural Environment Research Council (NERC) EAO Doctoral Training Partnership. MRA acknowledges funding support from the Natural Environment Research Council (NERC) through the UK National Centre for Atmospheric Science (NCAS). Instrumentational support was funded through the NERC Atmospheric Measurement and Observational Facility (AMOF).

**Competing interests**

The authors declare that they have no conflict of interest.

## Author contributions

GM, MRA, AV, YW and YS conceived the study. AV, YW, YS and MD conducted the experiments. SNP provided the volatility retrieval algorithm. TJB provided on-site help deploying the FIGAERO-CIMS and discussing the data analysis procedure. AV conducted the data analysis and wrote the manuscript with inputs from all co-authors.

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
