# Peer review of "Exploring the composition and volatility of secondary organic aerosols in mixed anthropogenic and biogenic precursor systems"

_Atmospheric Chemistry and Physics, 2021_

## Referee Comment (RC1)

**Review of Voliotis et al. 2021 "Exploring the composition and volatility of secondary organic aerosols in mixed anthropogenic and biogenic precursor systems"**

**Summary**

The authors investigated the composition and particle volatility of secondary organic aerosol systems from a-pinene (biogenic volatile organic compound, bVOC), o-cresol (anthropogenic VOC, aVOC), and a mixture of the two. As the two precursors are structurally very different (bicyclic monoterpene vs aromatic compound), the resulting SOA systems exhibit distinct chemical composition profiles, SOA mass yields, and particle volatility. In the mixed SOA system, a unique group of compounds appears in the "moderate" to "less" volatility range. This shows that, already in a binary mixture, complex interactions between the different oxidation processes can occur and these will impact the volatility distribution of the resulting SOA.

This work falls into the scope of ACP as it furthers our understanding of the interaction of different precursor groups (here aromatic and terpenoid compounds) and how this interactions impact SOA particle formation and properties. I recommend this work for publication after a few mayor concerns and many minor issues have been addressed. In several places, the manuscript is unnecessarily difficult to understand due to some poorly phrased sentences. I recommend that the native English speakers in the author team should have a more careful look at the overall language of the manuscript when submitting the revised verion.

**Major Comments**

1) The study suggests that the interaction of intermediate oxidation products (among them $RO_2$) is very important when more than one precursor is present. The chosen experiment conditions have a realistic OH concentration level (order of $10^6$ cm$^{-3}$) The VOC/NOx ratio is also in an atmospherically relevant range. But the VOC and NOx concentrations are orders of magnitude higher than could be expected in most regions with strong bVOC emissions. The fate of $RO_2$ (e.g. which termination products are formed or how many autooxidation steps occur) strongly depends on the balance between the consecutive reactions with $NO_2$, $HO_2$, and $RO_2$. The experiment conditions may favour the $NO_2$ and $RO_2$ paths over auto-oxidation and $HO_2$ reactions. How well will such experiments represent the actual atmosphere?

2) I disagree with how the authors decided to treat the elevated signal levels at the start of some thermograms ("instrument background", described in lines 254ff). From the example in Fig S1b, this looks like a carry-over from the preceding gas phase measurement (exponential-like decrease after switching). The gas phase zeros also show a delay in the decrease/increase of the signal indicating that this is a "sticky" compound. Another option is that semi/intermediate volatility gaseous compounds were adsorbed on the filter and quickly evaporate when the flow is switched to pure $N_2$.

This will occur already at room temperature. The driver is switching from a high concentration gas flow (= sampling) to the pure $N_2$ for desorption.

However, for the given example, the value decreases to close to 0 before the actual desorption peak starts at ~36min. I.e. the carry-over/artefact does not impact the region of interest. Note how the signal level before and after the desorption peak are the same and close to 0. The authors used the average value from 30 – 60 seconds as the "instrument background" and subtracted it from the whole ion thermograms. I indicated the value for the given example below (derived "by eye"). Subtracting the value of the red line will remove more than half of the signal from this peak. This will lead to systematic bias in the data. "Sticky" ions with high enough concentrations in the gas phase will be underestimated using this procedure. To avoid this bias, the authors could either remove the first X seconds from the integration for every ion (or equivalently set a minimum temperature from which to start the integration). Or they could fit an exponential (or other suitable) function to the 15-60 sec data and then use that as the gas-phase carry-over background value.

[Figure]

**Figure 1:** Copy of Figure S1b with added estimated "instrument background" level as red line.

3) I find the presentation of the FIGAERO-CIMS composition results in Figure 1 only useful for part of the discussion. The logarithmic scale is good to show the small, but important contributions from C>10 compounds (or C>7 for o-cresol) which would be probably invisible on a linear scale. But the eye is misguided by the seemingly large bars. E.g., the unique in mixture compounds seem to be predominantly present at C>10. But the majority of these compounds really falls into the C4-C8 range where the contribution is dwarfed by the large a-pinene compound bars. Also, this Figure reduces the composition information to the carbon number. While this is an important property to investigate, the degree of oxidation (OSc) as a measure of functionality (and somewhat Mw) is equally important - especially when discussing volatility.

I therefore recommend that the authors add a different depiction of their mass spectra results to the manuscript. One option is to plot the OSc value for each ion (calculated from the sum formula) vs the carbon number and then use marker size to indicate signal intensity (using a logarithmic spacing). The colour of the marker can be used to indicate the compound category. Examples for this type of figure can be found as Figure 5 in Faiola et al. (2019), Figure 8 in Ylisirniö et al.(2020), or Figure 1 in Isaacman-VanWertz et al. (2017). This type of figure can be helpful when investigating why the fraction of C>10 is not significantly different between the gas and particle phase for the mixture. It would directly show if the degree of oxidation is different for the compounds grouped by their carbon number (see specific comment about line 308ff). It will also be helpful for the discussion in section 4.1.

I am aware that there is a degree of uncertainty for the calculation of OSc for N containing compounds (Priestley et al., 2021). However, for plotting purposes this may be neglected.

[Figure]

**Figure 5.** Ozonolysis oxidation products measured in the chamber during the last 30 min of the experiment plotted on a 2-D axis of oxidation state (OSc) and carbon number for the (a) healthy plant emissions and (b) aphid-stressed emissions. Each diamond marker denotes one peak identified in the acetate-CIMS, and the size of the marker indicates intensity of the signal. Filled circles indicate the location of dominant SOA precursor species in the ozonolysis experiments with (solid orange circles) for sesquiterpenes and (solid blue circles) for monoterpenes.

**Figure 2:** Example for OSC vs C number plot from (Faiola et al., 2019)

4) Determining the particle phase volatility is an integral part of this study. Using volatility categories to simplify the complex FIGAERO-CIMS data is a valid approach. But unfortunately, the authors do not explain how the volatility categories were derived for the FIGAERO-CIMS measurements. They define three categories with rather hand-wavy terms (more, moderately, and less volatile) instead of using the established categories of semi-, low-, and extremely low volatile. There may be reasons to use these broader categories (however, no reasons are given). But it is surprising that the authors do not even mention a $T_{max}$-> $p_{sat}$/C* calibration using compounds with known $p_{sat}$ values here when one of the co-authors published about the necessity of a robust calibration (Bannan et al., 2019) and three more from the author list also participated in that publication. Also, the authors are aware of a more comprehensive study on FIGAERO-CIMS $T_{max}$ -> $p_{sat}$ calibration methods (Ylisirniö et al., 2021) as they cite this study in a different context. When they introduce their volatility categories, they instead refer to a non-relevant paper (Saha and Grieshop, 2016) using a thermodenuder method (the desorption temperature categories cannot be directly compared to FIGAERO desorption temperature) and a study (D'Ambro et al., 2019) which has two example ions which exhibit peaks a t 55 °C and 90 °C (and uses these to explain how there can be compounds with different volatility appearing at the same sum formula). Lopez-Hilfiker et al. (2014) does contain a $T_{max}$ -> $p_{sat}$ calibration curve, however model calculations (Schobesberger et al., 2018) and experiment by Ylisirniö et al. (2021) have shown that such calibration lines are instrument specific and cannot be easily transferred.

The authors have to clearly explain

    a. why they chose these desorption temperatures as category borders

    b. how these relate to the established S/L/ELVOC classification system c. what calibration was used for the FIGAERO-CIMS

    d. If they did not perform the necessary $T_{max}$->$p_{sat}$ calibrations for this data set, they need to mention this and clearly state what other method to relate desorption temperature and volatility they use.

5) Following up on the previous comment. It is commendable that two independent methods were applied to derive particle volatility. But the authors have not clearly explained how the volatility categories from FIGAERO-CIMS measurements relate to the VBS distribution. It is important to stress that compounds detected at a desorption temperature of 50 °C in the FIGAERO-CIMS do not necessarily have the same volatility as the fraction that evaporates during 30 sec in the thermodenuder at 50 °C.

6) Section 4.1 contains the main part of the interpretation of the observed data. The authors try to link the observed composition and particle volatility to the SOA formation processes. But unfortunately, this whole section is very hard to follow for multiple reasons:

    a. It is not clear when the authors are talking about processes/measurements of the gas phase are solely about the particle phase. Does "FIGAERO-CIMS measurements" imply only particle phase data (e.g. in line 474f)?

    b. They use fraction of contribution for different compound groups defined by carbon number. But this quantity is not easy to visualise from Figure 1 as it is in log scale. It may help to create some additional graphs/table in the supplement to help the reader follow this discussion. I understood that the authors are looking at three groups: oligomers with C > precursor (precursor =10 or 7), monomers with moderate fragmentation leading to precursor-x< C< precursor, significant fragmentation C< precursor -x. x is 5 for a-pinene and the mixture.

    c. The fraction of contribution of compound groups are then linked to the fractional contributions in the volatility distribution. But carbon number is not the only factor for volatility. That Mw may be the more direct measure that correlates with volatility. It should be easy to verify this by looking at the sum thermogram for each compound group (e.g. summing up all C>10 compound thermograms)?

    d. It is hard to follow how the authors determine what is comparable to the subgroups of the "unique to the mixture" compounds. Should not the properties of the C>10 compounds from the single precursor experiments be compared the properties of the C>10 in the unique to mixture category when investigating if the interactions in the mixture lead to more or less volatile compounds in that carbon number range?

I recommend that the authors carefully evaluate what they really want to express in this section and clarify their message.

**Specific comments**

1) Line 26: "…compounds with saturation concentration less or equal than 0.01ug m$^{-3}$…" This isolated value of 0.01 ug m$^{-3}$ is not very reader friendly. It is better to give the volatility class that this refers to (so LVOC ?).

2) Line 60ff: "A number of studies…" This sentence is generally difficult to understand. It is also an oversimplification. "Functionality" is usually associated with a larger number of functional groups (e.g. -OH or -C=O). Adding more functional groups to a molecule typically will reduce the vapour pressure as intermolecular forces become stronger and the molecular weight (Mw) increases. But a larger carbon backbone may have an even stronger impact on the volatility of a compound, e.g. when comparing monomer and dimer compounds which have same relative functionality (functional groups per carbon backbone). Li et al. (2016) show that volatility had a strong correlation with Mw for a wide range of functionalities.

3) Line 102: "aromatic oxidation" This term can be misunderstood as the oxidation being of aromatic character. Better use the term "oxidation of aromatic compounds"

4) Line 128: "electronic capture device grade nitrogen" ECD is not a specific enough purity identifier. A quick google search gave 3 different purity grades (between N4.8 and N6.0) depending on the supplier. Replace this by something specific.

5) Line 132: "…during the final fill cycle…" the description of the chamber operation does not specify any filling cycles. It rather suggests that there is a continuous flushing of the chamber before and after the experiments ("cleaning cycle")

6) Line 150ff: How much O$_3$ was formed during the experiments? This may have relevance as non-aromatic products from o-cresol may react with O$_3$.

7) Line 162: "…established by stabilising the chamber with clean air…" What is meant by "stabilising"?

8) Line 170ff: While I agree with using the "high O$_3$" cleaning procedure in general, the authors should not be so general in their statement here. The aromatic compounds (like C$_7$H$_8$O$_2$, the dominant product from o-cresol) will not react with O$_3$ even at this high concentration. So, this will not help with "oxidising any remaining gas-phase organic species".

9) Line 195ff: Was the assumption of minimal changes in the aerosol composition for this time period verified with the available instrumentation (AMS, FIGAERO-CIMS)?

10) Line 195ff: Which instrument was used for the MFR calculations? AMS total mass? AMS organic mass? SMPS total mass?

11) Line 205: Was the FIGAERO from Aerodyne or custom build? This should be stated as different instrument designs are currently in use.

12) Line 209: "high purity N$_2$" This is now the third purity grade for N$_2$ (after ECD and UHP). Where 3 different types of N$_2$ used?

13) Line 210 (and below): "0.635 cm OD" Is that a ¼ inch tube? While I strongly favour using metric units wherever possible, it feels odd to not use the precise description of the tubing.

14) Line 250f: "… were subtracted from the measured values in each cycle by linear interpolation." This is not clear. What was linearly interpolated? How does subtraction by interpolation work?

15) Line 250f: It is not clear if the authors are speaking about the gas phase part of the data or everything. Was the "instrument background" and the "chamber background" subtracted only for gas-phase data or also from the particle phase?

 16) Line 253ff: "…high initial particle phase signal was observed…" Did the authors scale the signals to the primary ions (I-species)?

17) Line 266-275: This whole paragraph is unnecessarily difficult to understand. See next three comments.

18) Line 268f: "…products with assigned formulae which were common among all systems and CIMS is unable to resolve (i.e., isomers)…" This means to me that the "common" category consists mainly of isomeric compounds, i.e., that there

 are even less true common compounds between a-pinene and o-cresol. Is that what the authors want to imply?

19) Line 270ff: "…we compared the resultant products categorised as unique to the mixture with the Master Chemical Mechanism." What is compared to the MCM? The compounds grouped into "common"? What are they compared to? MCM using a-pinene as input? o-cresol? A mixture?

20) Line 270ff: Why would a single precursor product expected from MCM calculations not show up in the single precursor

 experiments but only in the mixture experiment?

21) Line 277-288: This is another example where the authors do not clearly state what they did: Which data was used for the MFR and volatility model calculations? The mass concentration from AMS or SMPS? If AMS was used, what CE&RIE were used? How well did AMS and SMPS total mass agree?

22) Line 277-288: What is the uncertainty of the derived volatility distribution for this specific setup?

 23) Line 291: "Figure 1 shows the average distribution…." What was averaged? The authors have the reader guessing if the averaged multiple FIGAERO samples from one experiment. Or did they combine data from multiple experiments?

24) Line 294: "…unidentified fraction (se Section 2.4)" Section 2.4.1 states that more than 80% of the total fitted signal was assigned (Line 245). But there is no explanation about the unidentified fraction. Why is it unidentified? is there no reasonable formula possible? Is it outside of the trusted m/z range?

 25) Line 301f: The two main products for o-cresol oxidation are $C_7H_8O_2$ and $C_7H_7NO_4$. These are formally a H abstraction and OH addition or H abstraction and O&NO$_2$ addition. This seems to suggest that very little oxidation occurs for this precursor system. How does this compare to the expectation stated in Lines 102ff. Could this behaviour be linked to high VOC and NOx concentration in relation to OH (see mayor comment 1)?

26) Line 308f: The contribution of C>10 compounds is not significantly different between the gas and particle phase of the

 mixture. But are these really the same compounds? Are the compounds in the particle phase more oxidised? What is the volatility of the particle phase compounds (derived from Tmax of their ion thermograms)? Are they expected to be semi-volatile and thus show contribution in both gas and particle phase?

27) Line 311ff: Are $C_7H_8O_2$ and $C_7H_7NO_4$ still the dominant species for the o-cresol group? I.e. what fraction of the o-cresol group is attributed to these two compounds. Is this similar to what was seen for the single precursor experiment.

28)   Line 323: I do not understand this sentence. I guess this is supposed to be an interpretation of O:C ratio?

29)   Line 326 ff: The O:C(AMS) is smaller than the FIGAERO value for a-pinene SOA but larger for o-cresol SOA. For mixed SOA the effect seems to be cancelled. Could this change in trend be caused by the AMS using a parameterisation optimised for ambient data?

30)   Line 328 – 331: Again, an example for unclear language. The main challenges for FIGAERO-CIMS are the sensitivity effects, while the AMS mostly struggles with the fragmentation (i.e. detecting organo-nitrates mostly as $NO^+$ or $NO_2^+$). But the sentence structure implies that both problems apply to both instruments. This does not impact the overall message, but this sloppy style may be misleading for a novice to AMS and FIGAERO-CIMS while annoying the expert.

31)   Line 339ff: What is meant by "…provides an indirect measure of the volatility as a function of the FIGAERO-CIMS total particle phase signal"? Is this supposed to describe the meaning of the sum thermogram (sum of all ion signals as a function of desorption temperature)? "Volatility as a function of the total particle phase signal" means something else to me.

32)   Line 342: "average sum thermogram for each system" Again, what was averaged? All FIGAERO samples from one experiment? All samples from all experiments? How many samples are part of the averaging? (see also comment 34)

33)   Line 342: For the average values used before ±1σ was used. Why is ±2σ used here?

34)   Line 345: "characteristic experiment" why were these experiments chosen as characteristic? The authors should clearly state (best in the methods part):

a.   what were the averages they used (averaged over one experiment or multiple experiments?)

b.   how many FIGAERO samples where averaged?

c.   why certain experiments were chosen as representative for a SOA system.

d.   mark the "characteristic experiments in Table 1.

35)   Line 360 "modest volatilities" modest is the wrong word here. "moderate" is probably more fitting.

36)   Line 373: Unidentified ions was not clearly introduced. Was it just not possible to assign a sum formula? Are these the same masses as the unidentified in the a-pinene and o-cresol case?

37)   Figure 5: How different were the total aerosol mass concentrations ("$c_{OA}$") in the three experiment types? How much will these different loadings impact the observed volatility distribution of the SOA particles?

38)   Line 406: What does "large size of the molecules" refer to? High Mw? Higher carbon number?

39)   Line 407: what does "close to unity" mean in this context. The values were close to 1? The values from the two different systems were almost the same?

40)   Lines 413-427: What are the uncertainties of the O:C enhancement ratio? It looks like that it is possible to pick one out of the two experiments for each category and get almost overlapping slopes. I.e. the highest points from a-pinene and the lowest points from o-cresol almost overlap.

41)   Line 414: "O:C enchantment" = enhancement?

42) Line 415: The term "BP line" was not introduced. Is that the bypass line for the thermodenuder? By now I am a bit tired of having to guess what the authors are trying to say.

43) Line 425: "…whereas that of the mixed system was approximately between." What doe you mean by approximately between? The values for the mixed system a definitively between the values for the single precursor systems? Is this supposed to be "approximately half-way between"?

44) Line 426f: "…the less volatile SOA particles from the o-cresol system are considerably more oxygenated compared to the average than those of the α-pinene system." What is compared to what here? What is the "average"?

45) Line 436: The variable "maximum desorption temperature" was not properly introduced anywhere. This term can be misleading to a non- FIGAERO expert. As described in other comments, the authors should clearly introduce the relationship of $T_{max}$ (desorption temperature of the maximum of the ion/sum thermogram) and volatility.

46) Line 440: "This broad consistency…" If the authors want to express that the two volatility distributions roughly agree, then "broad" is not the right word here.

47) Line 447-467: Do the general statements of the authors about fragmentation include the initial oxidation step? Or are they looking at the fate of $RO/RO_2$ radicals?

48) Line 447-467: For the general fragmentation vs functionalisation discussion, the authors should point out the fundamental difference in the precursor structures. a-pinene is a bicycle molecule. I.e. two C-C bonds can be broken without actually loosing a C. o-cresol is a C6 aromatic compound. I.e. the aromatic ring will be very stabile against oxidation (e.g. no ozonolysis reaction. But once the aromaticity is broken, fragmentation reactions will create two very small molecules.

49) Line 460ff: The statement about the sensitivity of $I^-$ CIMS with regard to volatility classes is misleading in this context. It is correct that in "gas-phase mode" CIMS mostly detects I- and SVOCs. But the particle phase mode of FIGAEO-$I^-$-CIMS detects SVOC to ELVOC compounds equivalent to $C^*$ $10^4 – 10^{-6}$ ug $m^3$ or even lower. (Isaacman-VanWertz et al., 2017). 60% of the particle phase signal appears at desorption temperatures >90 °C. Even with all the differences between different FIGAERO instruments, that is in the LVOC range.

50) Line 476f: The conclusion that dimers are not relevant because the observed fraction of C>10 compounds is low is not correct. Many dimers are thermally labile and will be detected as thermal decomposition products with sum formulas similar to the corresponding monomer composition. A clear example is shown by D'Ambro et al. (2019) for $C_5H_{12}O_4$ which shows two distinct peaks (at 55°C and 90°C). More examples can be found in a recent overview on FIGAERO-

CIMS (Thornton et al., 2020). The authors need to adjust this statement to reflect the importance of thermal decomposition for oligomers.

51) Line 480: "…with 7 or more carbon numbers". this should be either "with 7 or more carbon atoms" or "with a carbon number of 7 or more"

52) Line 492: Fig S2 is not the right figure reference. Is this refereeing to Figure 1?

53)  Line 498ff: The authors compare the volatility of unique to mixture products to the rest of compounds in the mixed SOA particles with the same carbon numbers and attribute the observed differences to the change in O:C. But is the determining factor really the increase in oxidation or is it the increase in Mw?

54)  Line 504f: "The vast majority of those products were found to be common between all experiments…" This is confusing. This is supposedly about the "unique to mixture" compounds. How can they be found in all experiments? Is it not by
definition that these compounds only occur in the mixture experiments?

55)  Line 501-514: The most important indication if these C<5 compounds are likely from thermal decomposition should not be their O:C value but rather the thermogram data (Figure 7). Are compounds with a volatility equivalent to an average $T_{max} > 100\ °C$ expected to have a significant fraction in the gas phase?

56)  Line 504ff: It is not clear what the comparison of O:C is aiming at. The unique to mixture C<5 compounds have higher
O:C than the other C<5 compounds? Or the particle phase compounds compared with the gas phase compounds?

57)  Line 537ff: "…to assess the potential role of product interactions in altering the SOA particle volatility" This sentence suggests that conducting FIGAERO-CIMS measurements with a precursor mixture would be enough to achieve this. As I understand the identification method described in this manuscript, the key to assigning the compounds to the different categories (common/unique to mixture etc.) is the combination of chamber experiments with single precursors and the
mixture. The authors need to clarify what they mean in this sentence.

58)  Lines 543-558: I disagree with the interpretation of the decreasing $T_{max}$ values with increasing filter loading. To my knowledge and according to the modelling framework presented by Schobesberger et al. (2018) artefacts related to increases in filter loading will always lead to a shift to higher $T_{max}$ values. Previous studies have shown that the largest $T_{max}$ shifts when filter loadings where <200 ng to <2.5ug and with loading >~2.5 ug, $T_{max}$ changes were minor (Huang
et al., 2018; Wang and Hildebrandt Ruiz, 2018).

The change in the ion thermogram of $C_{10}H_{16}O_4$ (Figure S5) looks more like a change in volatility distribution for that specific sum formula. I.e., there are two or more isomers with different volatility and the more volatile one is not there for the lower mass loading sample. If this is the case, this is not an artefact but a real feature of the data. Without the information about the evolution of the mass concentration in the chamber, there are three possible explanations:

a.  The lower mass loading on the filter corresponds to a lower particle mass concentration in the chamber. Hence, the partitioning is affected and the volatile isomer with $T_{max}$~40 °C does not partition into the particle phase

            b.  There is ongoing chemistry in the chamber and the mass concentration reflects the evolution. At the time when the low loadings were sampled the more volatile isomer did not exist in sufficient quantities. Either
     because it had not formed yet or because it had reacted away.

            c.  If the lower mass loadings are "later" in the experiments, the more volatile isomer may react in the particle phase to form something else, possibly a dimer. Such a dimer will either show up at the dimer formula, at the same formula at higher desorption temperature or at a different sum formula if the thermal decomposition is more complex.

To verify this explanation, the authors should check their data. Does the $T_{max}$ decrease occur only for $T_{max}$ values in the SVOC range?

    59) Line 569: "…as few as 10 products" This suggests that 1 sum formula is just 1 compound is not strictly true. The authors should formulate this more carefully.

**References**

Bannan, T. J., Le Breton, M., Priestley, M., Worrall, S. D., Bacak, A., Marsden, N. A., Mehra, A., Hammes, J., Hallquist, M., Alfarra, M. R., Krieger, U. K., Reid, J. P., Jayne, J., Robinson, W., McFiggans, G., Coe, H., Percival, C. J. and Topping, D.: A method for extracting calibrated volatility information from the FIGAERO-HR-ToF-CIMS and its experimental application, Atmos. Meas. Tech., 12(3), 1429–1439, doi:10.5194/amt-12-1429-2019, 2019.

D'Ambro, E. L., Schobesberger, S., Gaston, C. J., Lopez-Hilfiker, F. D., Lee, B. H., Liu, J., Zelenyuk, A., Bell, D., Cappa, C. D., Helgestad, T., Li, Z., Guenther, A., Wang, J., Wise, M., Caylor, R., Surratt, J. D., Riedel, T., Hyttinen, N., Salo, V.-T., Hasan, G., Kurtén, T., Shilling, J. E. and Thornton, J. A.: Chamber-based insights into the factors controlling IEPOX SOA yield, composition, and volatility, Atmos. Chem. Phys. Discuss., (August), 1–20, doi:10.5194/acp-2019-271, 2019.

Faiola, C. L., Pullinen, I., Buchholz, A., Khalaj, F., Ylisirniö, A., Kari, E., Miettinen, P., Holopainen, J. K., Kivimäenpää, M., 295 Schobesberger, S., Yli-Juuti, T. and Virtanen, A.: Secondary Organic Aerosol Formation from Healthy and Aphid-Stressed Scots Pine Emissions, ACS Earth Sp. Chem., doi:10.1021/acsearthspacechem.9b00118, 2019.

Huang, W., Saathoff, H., Pajunoja, A., Shen, X., Naumann, K. H., Wagner, R., Virtanen, A., Leisner, T. and Mohr, C.: α-Pinene secondary organic aerosol at low temperature: Chemical composition and implications for particle viscosity, Atmos. Chem. Phys., 18(4), 2883–2898, doi:10.5194/acp-18-2883-2018, 2018.

Isaacman-VanWertz, G., Massoli, P., O'Brien, R. E., Nowak, J. B., Canagaratna, M. R., Jayne, J. T., Worsnop, D. R., Su, L., Knopf, D. A., Misztal, P. K., Arata, C., Goldstein, A. H. and Kroll, J. H.: Using advanced mass spectrometry techniques to fully characterize atmospheric organic carbon: Current capabilities and remaining gaps, Faraday Discuss., 200, 579–598, doi:10.1039/c7fd00021a, 2017.

Li, Y., Pöschl, U. and Shiraiwa, M.: Molecular corridors and parameterizations of volatility in the chemical evolution of 305 organic aerosols, Atmos. Chem. Phys., 16(5), 3327–3344, doi:10.5194/acp-16-3327-2016, 2016.

Lopez-Hilfiker, F. D., Mohr, C., Ehn, M., Rubach, F., Kleist, E., Wildt, J., Mentel, T. F., Lutz, A., Hallquist, M., Worsnop, D. and Thornton, J. A.: A novel method for online analysis of gas and particle composition: description and evaluation of a Filter Inlet for Gases and AEROsols (FIGAERO), Atmos. Meas. Tech., 7, 983–1001, doi:10.5194/amt-7-983-2014, 2014.

Priestley, M., Bannan, T. J., Le Breton, M., Worrall, S. D., Kang, S., Pullinen, I., Schmitt, S., Tillmann, R., Kleist, E., Zhao, 310 D., Wildt, J., Garmash, O., Mehra, A., Bacak, A., Shallcross, D. E., Kiendler-Scharr, A., Hallquist, Å. M., Ehn, M., Coe, H.,

Percival, C. J., Hallquist, M., Mentel, T. F. and McFiggans, G.: Chemical characterisation of benzene oxidation products under high- And low-NOx conditions using chemical ionisation mass spectrometry, Atmos. Chem. Phys., 21(5), 3473–3490, doi:10.5194/acp-21-3473-2021, 2021.

Saha, P. K. and Grieshop, A. P.: Exploring Divergent Volatility Properties from Yield and Thermodenuder Measurements of
Secondary Organic Aerosol from α-Pinene Ozonolysis, Environ. Sci. Technol., 50(11), 5740–5749, doi:10.1021/acs.est.6b00303, 2016.

Schobesberger, S., D'Ambro, E. L., Lopez-Hilfiker, F. D., Mohr, C. and Thornton, J. A.: A model framework to retrieve thermodynamic and kinetic properties of organic aerosol from composition-resolved thermal desorption measurements, Atmos. Chem. Phys., 18(20), 14757–14785, doi:10.5194/acp-18-14757-2018, 2018.

Thornton, J. A., Mohr, C., Schobesberger, S., D'Ambro, E. L., Lee, B. H. and Lopez-Hilfiker, F. D.: Evaluating Organic Aerosol Sources and Evolution with a Combined Molecular Composition and Volatility Framework Using the Filter Inlet for Gases and Aerosols (FIGAERO), Acc. Chem. Res., 53(8), 1415–1426, doi:10.1021/acs.accounts.0c00259, 2020.

Wang, D. S. and Hildebrandt Ruiz, L.: Chlorine-initiated oxidation of alkanes under high-NO conditions: Insights into secondary organic aerosol composition and volatility using a FIGAERO-CIMS, Atmos. Chem. Phys., 18(21), 15535–15553,
doi:10.5194/acp-18-15535-2018, 2018.

Ylisirniö, A., Buchholz, A., Mohr, C., Li, Z., Barreira, L., Lambe, A., Faiola, C., Kari, E., Yli-Juuti, T., Nizkorodov, S. A., Worsnop, D. R., Virtanen, A. and Schobesberger, S.: Composition and volatility of secondary organic aerosol (SOA) formed from oxidation of real tree emissions compared to simplified volatile organic compound (VOC) systems, Atmos. Chem. Phys., 20(9), 5629–5644, doi:10.5194/acp-20-5629-2020, 2020.

Ylisirniö, A., Barreira, L. M. F., Pullinen, I., Buchholz, A., Jayne, J., Krechmer, J. E., Worsnop, D. R., Virtanen, A. and Schobesberger, S.: On the calibration of FIGAERO-ToF-CIMS: Importance and impact of calibrant delivery for the particle-phase calibration, Atmos. Meas. Tech., 14(1), 355–367, doi:10.5194/amt-14-355-2021, 2021.

---

## Author Response (AR1)

**Responses to reviewer 1**

**Review of Voliotis et al. 2021 "Exploring the composition and volatility of secondary organic aerosols in mixed anthropogenic and biogenic precursor systems"**

**Summary**

The authors investigated the composition and particle volatility of secondary organic aerosol systems from a-pinene (biogenic volatile organic compound, bVOC), o-cresol (anthropogenic VOC, aVOC), and a mixture of the two. As the two precursors are structurally very different (bicyclic monoterpene vs aromatic compound), the resulting SOA systems exhibit distinct chemical composition profiles, SOA mass yields, and particle volatility. In the mixed SOA system, a unique group of compounds appears in the "moderate" to "less" volatility range. This shows that, already in a binary mixture, complex interactions between the different oxidation processes can occur and these will impact the volatility distribution of the resulting SOA.

This work falls into the scope of ACP as it furthers our understanding of the interaction of different precursor groups (here aromatic and terpenoid compounds) and how this interactions impact SOA particle formation and properties. I recommend this work for publication after a few mayor concerns and many minor issues have been addressed. In several places, the manuscript is unnecessarily difficult to understand due to some poorly phrased sentences. I recommend that the native English speakers in the author team should have a more careful look at the overall language of the manuscript when submitting the revised verion.

The authors would like to thank the reviewer for his/her time and effort to provide constructive comments/suggestions to our work. Your contribution is much appreciated.

Please find below itemised replies to each one of the comments/suggestions.

**Responses to major comments:**

1) The study suggests that the interaction of intermediate oxidation products (among them RO2) is very important when more than one precursor is present. The chosen experiment conditions have a realistic OH concentration level (order of $10^6$ cm-3) The VOC/NOx ratio is also in an atmospherically relevant range. But the VOC and NOx concentrations are orders of

magnitude higher than could be expected in most regions with strong bVOC emissions. The fate of RO2 (e.g. which termination products are formed or how many autooxidation steps occur) strongly depends on the balance between the consecutive reactions with NO2, HO2, and RO2. The experiment conditions may favour the NO2 and RO2 paths over auto-oxidation and HO2 reactions. How well will such experiments represent the actual atmosphere?

As also stated in the original manuscript (L82-94), this study is a part of a larger project investigating the secondary organic aerosol (SOA) formation in mixed systems employing a suite of offline and online instrumentation. In order to form adequate amounts of SOA mass required for offline analysis, particularly high VOC concentrations (and consequently $NO_x$) had to be used. As an indication of the relevance of the magnitude of oxidant reactivity in our system, previous studies have showed ambient OH reactivity (estimated by measurements of OH lifetime) to be between ~10 and 116 $s^{-1}$ (Whalley et al., 2016), while in our experiments are ~404 $s^{-1}$ (product of VOC concentration with reaction rate towards OH). In the ambient atmosphere, the OH reactivity will include that of smaller compounds (such as CO, $CH_4$), in turn affecting the speciation and fate of $RO_2$ and the $RO_2:HO_2$. In our setup, the particularly high concentration used is expected to promote the $RO_2$-$RO_2$ cross reactions and supress the $RO_2+HO_2$ and $RO_2+NO$ reactions (Peng and Jimenez, 2020), which are considered as dominant radical termination pathways in the atmosphere. Consequently, the increased presence of "unique to the mixture" compounds might be attributable to the selected experimental conditions to an unknown degree. Nonetheless, our scope here is not to mimic the ambient atmosphere, rather than explore the potential of the molecular interactions in mixtures to alter the SOA particle volatility. Clearly, it would have been desirable to work in more atmospherically relevant chemical regime, however due to analytical limitations certain compromises were necessary. The above discussion has been included in the discussion (L672-683) and conclusion (L767-771) sections in the revised manuscript.

2) I disagree with how the authors decided to treat the elevated signal levels at the start of some thermograms ("instrument background", described in lines 254ff). From the example in Fig S1b, this looks like a carry-over from the preceding gas phase measurement (exponential-like decrease after switching). The gas phase zeros also show a delay in the decrease/increase of the signal indicating that this is a "sticky" compound. Another option is that semi/intermediate volatility gaseous compounds were adsorbed on the filter and quickly evaporate when the flow is switched to pure N2. This will occur already at room temperature. The driver is switching from a high concentration gas flow (= sampling) to the pure N2 for desorption. However, for the

given example, the value decreases to close to 0 before the actual desorption peak starts at ~36min. I.e. the carry-over/artefact does not impact the region of interest. Note how the signal level before and after the desorption peak are the same and close to 0. The authors used the average value from 30 – 60 seconds as the "instrument background" and subtracted it from the whole ion thermograms. I indicated the value for the given example below (derived "by eye"). Subtracting the value of the red line will remove more than half of the signal from this peak. This will lead to systematic bias in the data. "Sticky" ions with high enough concentrations in the gas phase will be underestimated using this procedure. To avoid this bias, the authors could either remove the first X seconds from the integration for every ion (or equivalently set a minimum temperature from which to start the integration). Or they could fit an exponential (or other suitable) function to the 15-60 sec data and then use that as the gas-phase carry-over background value.

We agree with the reviewer that the selected method to account for instrument contamination from the gas phase sampling might be less than ideal. We also acknowledge that the example thermogram shown in the SI of the original manuscript (Fig. S1b) was not adequate to support our rationale, as it does not represent the full range of different thermograms observed. In the Figure 1 below (now Fig. S5 in the revised manuscript), more representative examples of few ions with different desorption profiles are shown as a function of the FIGAERO-CIMS raw signal. As can be seen, the products observed in our experiments exhibited substantially different desorption profiles. Characteristically, ion #1 signal shows roughly a flat line until the desorption temperature reaches ~50 ºC where it starts decreasing, while ion #4 signal is decreasing sharply until ~75 ºC where it's decreasing trend changes. On the other hand, ions #2, 3 and 5 show a somewhat increased initial signal, exhibiting expected thermogram shapes. This shows that accounting for the instrument contamination from the gas phase is not straightforward. Whilst actual mass might be present in the thermograms of the ions #1 and #4, the large carry-over signal from the gas phase makes their observation challenging. Owing to these trends, disregarding the first x seconds from the desorption cycle will not take into account the slow carry-over and will result in erroneous integrations. Similarly, although the exponential fitting might work in some cases (e.g. ion #4), in other cases, such as the #3 and #5 (and for clarity, ions like those in Figure 2 below), the exponential fitting in the first x seconds will result to an exponentially increasing "instrument background" line. This will therefore result to the complete removal of some of the products after the subtraction (e.g. example 3 in Fig. 2) or it will lead to

an underestimation of the contributions of others (e.g. example 1, 2 and 4 in Fig. 2). Consequently, any of the reviewers' suggested methods will still result in a bias.

The vast majority of the thermograms observed have similar shapes as those of the ions #2, 3 and 5 or those shown in the Fig. 2 below, however the presence of ions with different profiles, despite they represent a small fraction of the ions observed, shows that the background treatment is not straightforward. We believe that our selected method provides a reasonable correction for all the types of thermograms observed within our experimental design. In addition, after looking on the example given on Figure S1b in the SI of the original manuscript, we further decided to shift our "instrument background" window to 60-90[th] second of the desorption cycle, where the temperature is still at ambient levels (Fig. S2 of original manuscript), that could lower the bias in compounds where the carry-over gas phase signal is declining sharply over the first minute of the cycle.

Clearly, working with high concentrations in chambers might induce certain biases in the particle phase of the FIGAERO-CIMS from the "sticky" compounds. Additional future work will be needed to develop robust methods for preventing such contaminations from the gas phase.

[Figure]

**Figure 1:** Examples of desorption profiles of various ions identified. The vast majority of the products identified were having similar profiles as the ions #2, 3 and 5, however some ions exhibited different shapes, as the ions #1 and 4. The dashed black box shows an approximate of the 60-90[th] second of the desorption cycle that was selected as "instrument background" region.

[Figure]

**Figure 2:** Examples of measured thermograms (blue lines), resulted exponential fitting over the 30ᵗʰ-60ᵗʰ second of the desorption cycle (red lines) to establish the "instrument background" signal and corrected for background signal (i.e., measured-fitted; yellow lines).

1) I find the presentation of the FIGAERO-CIMS composition results in Figure 1 only useful for part of the discussion. The logarithmic scale is good to show the small, but important contributions from C>10 compounds (or C>7 for o-cresol) which would be probably invisible on a linear scale. But the eye is misguided by the seemingly large bars. E.g., the unique in mixture compounds seem to be predominantly present at C>10. But the majority of these compounds really falls into the C4-C8 range where the contribution is dwarfed by the large a-pinene compound bars. Also, this Figure reduces the composition information to the carbon number. While this is an important property to investigate, the degree of oxidation (OSc) as a measure of functionality (and somewhat Mw) is equally important - especially when discussing volatility. I therefore recommend that the authors add a different depiction of their mass spectra results to the manuscript. One option is to plot the OSc value for each ion (calculated from the sum formula) vs the carbon number and then use marker

size to indicate signal intensity (using a logarithmic spacing). The colour of the marker can be used to indicate the compound category. Examples for this type of figure can be found as Figure 5 in Faiola et al. (2019), Figure 8 in Ylisirniö et al.(2020), or Figure 1 in Isaacman-VanWertz et al. (2017). This type of figure can be helpful when investigating why the fraction of C>10 is not significantly different between the gas and particle phase for the mixture. It would directly show if the degree of oxidation is different for the compounds grouped by their carbon number (see specific comment about line 308ff). It will also be helpful for the discussion in section 4.1. I am aware that there is a degree of uncertainty for the calculation of OSc for N containing compounds (Priestley et al., 2021). However, for plotting purposes this may be neglected.

The Figure 1 plots have been adjusted to show the OSc vs. nC while the nC plots with logarithmic scale have been moved to the supplementary material. All the discussion in sections 3 and 4 have been adjusted accordingly.

2) Determining the particle phase volatility is an integral part of this study. Using volatility categories to simplify the complex FIGAERO-CIMS data is a valid approach. But unfortunately, the authors do not explain how the volatility categories were derived for the FIGAERO-CIMS measurements. They define three categories with rather hand-wavy terms (more, moderately, and less volatile) instead of using the established categories of semi-, low-, and extremely low volatile. There may be reasons to use these broader categories (however, no reasons are given). But it is surprising that the authors do not even mention a Tmax-> psat/C* calibration using compounds with known psat values here when one of the co-authors published about the necessity of a robust calibration (Bannan et al., 2019) and three more from the author list also participated in that publication. Also, the authors are aware of a more comprehensive study on FIGAERO-CIMS Tmax -> psat calibration methods (Ylisirniö et al., 2021) as they cite this study in a different context. When they introduce their volatility categories, they instead refer to a non-relevant paper (Saha and Grieshop, 2016) using a thermodenuder method (the desorption temperature categories cannot be directly compared to FIGAERO desorption temperature) and a study (D'Ambro et al., 2019) which has two example ions which exhibit peaks a t 55 °C and 90 °C (and uses these to explain how there can be compounds with different volatility appearing at the same sum formula). Lopez-Hilfiker et al. (2014) does contain a Tmax -> psat calibration curve, however model calculations (Schobesberger et al., 2018) and experiment by Ylisirniö et al. (2021) have shown that such calibration lines are instrument specific and cannot be easily transferred.

The authors have to clearly explain
a. why they chose these desorption temperatures as category borders

b. how these relate to the established S/L/ELVOC classification system

c. what calibration was used for the FIGAERO-CIMS

d. If they did not perform the necessary Tmax->psat calibrations for this data set, they need to mention this and clearly state what other method to relate desorption temperature and volatility they use.

On the specific instrument setup used in this study, only one series of Tmax->psat calibrations were available using the syringe method on a certain filter loading using polyethylene glycols as calibrants (see Fig. S8 in the revised SI). After the programme of study presented here, the instrument was rebuilt so it was impossible to conduct more calibrations. Owing to the recent works from Ylisirniö et al. (2021) highlighting the importance of multiple calibrations at different filter loadings and the impact of the calibrant delivery method to the resultant volatilities, the authors initially considered that the representation of the volatility from only a series of calibrations might be inadequate and potentially erroneous (e.g., discussion on the original manuscript on the Tmax shift over the filter loading). Nonetheless, after considering the reviewers' major comments (including those that follow below) we decided to apply these calibrations to our results and represent the volatilities from the FIGAERO-CIMS in the VBS framework. Furthermore, we also estimated the SOA particle volatility by performing partitioning calculations and the results between all the methods are compared in the Section 4.1 of the revised manuscript.

3) Following up on the previous comment. It is commendable that two independent methods were applied to derive particle volatility. But the authors have not clearly explained how the volatility categories from FIGAERO-CIMS measurements relate to the VBS distribution. It is important to stress that compounds detected at a desorption temperature of 50 °C in the FIGAERO-CIMS do not necessarily have the same volatility as the fraction that evaporates during 30 sec in the thermodenuder at 50 °C.

Clearly this last statement is correct and important and we had not meant to imply any such correspondence between components evaporating at the same reported temperature in the two different instruments. As stated in the response to major comment 4, the volatilities from the FIGAERO-CIMS are now expressed in the VBS framework and are compared with those from the TD (see Sections 3 and 4 in the revised manuscript).

6) Section 4.1 contains the main part of the interpretation of the observed data. The authors try to link the observed composition and particle volatility to the SOA formation processes. But unfortunately, this whole section is very hard to follow for multiple reasons:

a. It is not clear when the authors are talking about processes/measurements of the gas phase are solely about the particle phase. Does "FIGAERO-CIMS measurements" imply only particle phase data (e.g. in line 474f)?

b. They use fraction of contribution for different compound groups defined by carbon number. But this quantity is not easy to visualise from Figure 1 as it is in log scale. It may help to create some additional graphs/table in the supplement to help the reader follow this discussion. I understood that the authors are looking at three groups: oligomers with C > precursor (precursor =10 or 7), monomers with moderate fragmentation leading to precursor-x< C< precursor, significant fragmentation C< precursor -x. x is 5 for a-pinene and the mixture.

c. The fraction of contribution of compound groups are then linked to the fractional contributions in the volatility distribution. But carbon number is not the only factor for volatility. That Mw may be the more direct measure that correlates with volatility. It should be easy to verify this by looking at the sum thermogram for each compound group (e.g. summing up all C>10 compound thermograms)?

d. It is hard to follow how the authors determine what is comparable to the subgroups of the "unique to the mixture" compounds. Should not the properties of the C>10 compounds from the single precursor experiments be compared the properties of the C>10 in the unique to mixture category when investigating if the interactions in the mixture lead to more or less volatile compounds in that carbon number range?

I recommend that the authors carefully evaluate what they really want to express in this section and clarify their message.

In the revised manuscript, this particular section (now Section 4.2) has been completely revised. We now explicitly state whether the total signal is referring to gas or particle phase (6.a.), Fig. 1 has been changed while additional figures have been added to explain the volatilities per nC group numbers (6b.). Furthermore, in addition to the carbon number and the O:C we now investigate the MW and OSc of all groups in order to provide a more thorough investigation (6c). The single precursor experiments are also compared in more detail in the revised manuscript (6d).

**Responses to minor comments:**

1) Line 26: "…compounds with saturation concentration less or equal than 0.01ug m-3…" This isolated value of 0.01 ug m-3 is not very reader friendly. It is better to give the volatility class that this refers to (so LVOC ?).

We agree that isolated values might not be always very reader friendly in the context of the abstract. In the revised manuscript, the abstract has been changed to reflect the necessary changes in response to the reviewers' comments and this sentence no longer exists. Please find the revised abstract copied below:

"*Secondary organic aerosol (SOA) formation from mixtures of volatile precursors may be influenced by the molecular interactions of the components of the mixture. Here, we report measurements of the volatility distribution of SOA formed from the photo-oxidation of o-cresol, α-pinene and their mixtures, representative anthropogenic and biogenic precursors, in an atmospheric simulation chamber. The combination of two independent thermal techniques (thermal denuder; TD, and the Filter Inlet for Gases and Aerosols coupled to a high resolution time of flight chemical ionisation mass spectrometer; FIGAERO-CIMS) to measure the particle volatility, along with detailed gas and particle phase composition measurements provides links between the chemical composition of the mixture and the resultant SOA particle volatility. The SOA particle volatility obtained by the two independent techniques showed substantial discrepancies. The particle volatility obtained by the TD was wider, spanning across the LVOC and SVOC range, while the respective derived from the FIGAERO-CIMS using two different methods (i.e., calibrated Tmax and partitioning calculations) showed to be substantially higher (mainly in the SVOC and IVOC, respectively) and narrow. Although the quantification of the SOA particle volatility was challenging, both techniques and methods showed similar trends, with the volatility of the SOA formed from the photo-oxidation of α-pinene being higher that measured in the o-cresol system, while the volatility of the SOA particles of the mixture was between those measured at the single precursor systems. This behaviour could be explained by two opposite effects, the scavenging of the larger molecules with lower volatility produced in the single precursor experiments that led to an increase of the average volatility and the formation of unique-to-the-mixture products that had higher O:C, MW, OSc and consequently, lower volatility compared to those derived from the individual precursors. We further discuss the potential limitations of FIGAERO-CIMS to report quantitative volatilities and their implications to the reported results and we show that the particle volatility changes can be qualitatively assessed, while caution should be held when linking the chemical composition to the particle volatility. These results present the first detailed observations of SOA particle volatility and composition in mixed anthropogenic and biogenic systems and provide an analytical context that can be used to explore particle volatility in chamber experiments.*"

2) Line 60ff: "A number of studies…" This sentence is generally difficult to understand. It is also an oversimplification. "Functionality" is usually associated with a larger number of functional groups (e.g. -OH or -C=O). Adding more functional groups to a

molecule typically will reduce the vapour pressure as intermolecular forces become stronger and the molecular weight (Mw) increases. But a larger carbon backbone may have an even stronger impact on the volatility of a compound, e.g. when comparing monomer and dimer compounds which have same relative functionality (functional groups per carbon backbone). Li et al. (2016) show that volatility had a strong correlation with Mw for a wide range of functionalities.

The sentence has been revised to:

*"The SOA volatility is strongly dependent on the molecular weight of the compound (Li et al., 2016) and the functional groups present (and to a lesser extent, the activity coefficients) (McFiggans et al., 2010;Barley et al., 2009;Topping et al., 2011)."*

3) Line 102: "aromatic oxidation" This term can be misunderstood as the oxidation being of aromatic character. Better use the term "oxidation of aromatic compounds"

The correction has been made according to your suggestion.

4) Line 128: "electronic capture device grade nitrogen" ECD is not a specific enough purity identifier. A quick google search gave 3 different purity grades (between N4.8 and N6.0) depending on the supplier. Replace this by something specific.

The nitrogen used was ECD grade N4.8 (purity 99.998%). This information has been included in the revised manuscript (L134).

5) Line 132: "…during the final fill cycle…" the description of the chamber operation does not specify any filling cycles. It rather suggests that there is a continuous flushing of the chamber before and after the experiments ("cleaning cycle")

We see that this statement might have caused some confusion. After the flushing of the chamber with clean air ("pre-experiment"), MAC was stabilised with clean air without reactants for about an hour ("chamber background"; renamed as "chamber stabilisation" in the revised manuscript). In order to ensure rapid and effective mixing of the reactants, MAC was then partly flushed to an approximate 1/3 of its volume and all the reactants were subsequently added to the chamber while filling with clean air to its maximum capacity (i.e., final fill cycle). When all the reactants were introduced and the chamber was on its maximum volume, the "dark unreactive" phase was commenced. We have clarified this in the revised manuscript (L175-180) to read as:

*"iii) The "dark unreactive" phase commenced after the addition of the VOC precursor(s), NO2 and seed aerosol to the chamber in the dark. In order to ensure rapid and effective mixing, MAC was partly flushed to an approximate 1/3 of its volume and all the reactants were subsequently added to the chamber while filling with clean air to its maximum capacity (i.e., final fill cycle). During this period (also ~ 1 h), the initial conditions of the chamber were established."*

6) Line 150ff: How much O3 was formed during the experiments? This may have relevance as non-aromatic products from o-cresol may react with O3.

Typical O3 concentrations formed are shown for specific experiments on Fig. 3 below. As also stated in the section 2.2 of the original manuscript, the secondary O3 formed is expected to affect the iso-reactivity as it will react with all non-aromatic compounds leading to O3- derived oxidation products.

[Figure]

**Figure 3:** Typical O3 concentrations formed in experiments in each of the systems.

7) Line 162: "…established by stabilising the chamber with clean air…" What is meant by "stabilising"?

Subsequently after the "pre-experiment" phase MAC was left with clean air in the absence of light and reactants, so the deployed instrumentation establish baselines. The sentence has been revised (L173-174) to:

*"ii) The "chamber stabilisation" was established subsequently by leaving the chamber with clean air without reactants. This phase lasted for ~1 h and the established baselines were thereafter subtracted from the measurements."*

8) Line 170ff: While I agree with using the "high O3" cleaning procedure in general, the authors should not be so general in their statement here. The aromatic compounds (like C7H8O2, the dominant product from o-cresol) will not react with O3 even at this high concentration. So, this will not help with "oxidising any remaining gas-phase organic species".

We agree with the reviewer that this might sound as a generalisation. It is true that aromatic species are not expected to react with $O_3$ at the given concentrations and timescales. The sentence has been corrected to include only the $O_3$- reacting species (L182-184) as:

*"v)A "post-experiment" cleaning procedure normally comprised the high flowrate flushing of the chamber with clean air for ~1.5 h, subsequently filling with high concentration of O3 ($\geq$1 ppm) and soaking overnight to oxidise any remaining O3-reacting gas-phase organic species."*

It should be further noted that an intensive cleaning procedure was conducted periodically (at least once a week) within the experimental campaign. In this procedure, the chamber was filled with high concentrations of O3 ($\geq$1 ppm) and water vapour (RH$\geq$70%) and the chamber was illuminated for at least 4 h , after the quartz filters have been removed in front of the arc lamps to maximise the jO(1D) and thereby the OH levels. The chamber was then flushed for 1.5h with high flow rate clean air (3 $m^3$ $min^{-1}$).This procedure has been also included in the revised manuscript (L184-190), as:

*"It should be further noted that within the experimental campaign an intensive cleaning procedure was conducted periodically (at least once per week). In this procedure, the chamber was filled with high concentrations of O3 ($\geq$1 ppm) and water vapour (RH$\geq$70%) and the chamber was illuminated for at least 4 h, after the quartz filters have been removed in front of the arc lamps to maximise the jO(1D) and thereby the OH levels. The chamber was then flushed for 1.5h with high flow rate clean air (3 $m^3$ $min^{-1}$). All the results given below correspond to the "experiment" phase only".*

9) Line 195ff: Was the assumption of minimal changes in the aerosol composition for this time period verified with the available instrumentation (AMS, FIGAERO-CIMS)?

Figure 4 below shows the trajectories of the f43 and f44 signals of the AMS and Fig. 5 shows the number of carbon atom distribution as a function of the particle phase signal of all the products identified by the FIGAERO-CIMS during the last two cycles, corresponding to particles collected between the 4th and 5.5th hour of each experiment.

Apparently, the compositional information obtained from both instruments was sufficiently stable to surmise minimal changes in the composition.

[Figure]

**Figure 4:** f44 and f43 signals obtained by the HR-AMS over the timeframe of the experiments.

[Figure]

**Figure 5:** Number of carbon atoms of all the products identified in the last two FIGAERO-CIMS cycles in the particle phase in characteristic experiments in each system.

10) Line 195ff: Which instrument was used for the MFR calculations? AMS total mass? AMS organic mass? SMPS total mass?

We thank the reviewer for pointing this out. In this study, the mass fraction remaining was calculated using the total SOA mass (i.e., organic) measured by the HR-AMS. This has been clarified in the revised manuscript (L233-236)

*"The resulting mass fraction remaining (MFR) in each temperature step was calculated as the ratio of the organics mass (measured by the HR-AMS; hereafter SOA mass) passing through the TD to the average bypass concentration obtained before and after each step."*

11) Line 205: Was the FIGAERO from Aerodyne or custom build? This should be stated as different instrument designs are currently in use.

The FIGAERO used in this study was made from Aerodyne Research Inc. This has been clarified in the revised manuscript (L241-242) as:

*"Near-real time gas and particle composition was measured using the Filter Inlet for Gas and Aerosols (FIGAERO; Aerodyne Research Inc.) (Lopez-Hilfiker et al., 2014) coupled to an Iodine high-resolution time of flight chemical ionisation mass spectrometer (Lee et al., 2014) (hereafter FIGAERO-CIMS)"*

11) Line 209: "high purity N2" This is now the third purity grade for N2 (after ECD and UHP). Where 3 different types of N2 used?

We thank the reviewer for pointing out this typo; it should state "ultra-high purity" instead of "high purity". Two types of N2 were used; a) ECD grade (99.998%) used as carrier gas to carry the VOC and NOx to the chamber and b) UHP (99.999%) used in the FIGAERO-CIMS operation. The correction has been made in the revised manuscript (L245).

*"After a period of time, the deposited aerosol is thermally desorbed using ultra high purity (UHP) N2 as a carrier gas".*

12) Line 210 (and below): "0.635 cm OD" Is that a ¼ inch tube? While I strongly favour using metric units wherever possible, it feels odd to not use the precise description of the tubing.

The 0.635 cm OD corresponds to ¼-inch tube. The intention was to keep the units in the metric system; we understand however that some readers might not be familiar with the corresponding

measures. The correction has been made in the corresponding parts of the revised manuscript (L246-250) as:

*"In our setting, particles were sampled from the middle of the chamber through a 2 m long, ¼-inch stainless steel tube at 1 sL min-1 and deposited on a PTFE filter (Zefluor, 2.0 µm pore size). The flow over the filter is continually monitored by an MKS mass flow meter (MFM) after the sample has passed over the filter so ensure a known volume is sampled in each cycle. The gas phase was sampled through a separate line, using a 2 m long, ¼-inch PTFE tubing (Sulyok et al., 2002;Mittal et al., 2013) at 2 sL min-1 that results in a sample residence time of <2s."*

13) Line 250f: "… were subtracted from the measured values in each cycle by linear interpolation." This is not clear. What was linearly interpolated? How does subtraction by interpolation work?

We understand that this sentence might be confusing to the readers and the "linearly interpolated" was not the right expression here. The gas phase "instrument background" was periodically established in each gas phase sampling cycle (see Fig. S1 in the original manuscript or Fig. S5 in the revised manuscript) by flushing the instrument with UHP $N_2$. The signal over each background cycle was averaged and was subtracted from the nearest in time measurements. This has been clarified in the revised manuscript (L287-288) as:

*"The gas-phase data obtained during "instrument background" mode (see Section 2.3 and Fig. S5b) were averaged in background cycle and subtracted from the nearest in time measurements."*

14) Line 250f: It is not clear if the authors are speaking about the gas phase part of the data or everything. Was the "instrument background" and the "chamber background" subtracted only for gas-phase data or also from the particle phase?

The procedure followed was similar for each phase (i.e., gas and particle) and was comprising two steps: first the "instrument background" subtraction, and second, the chamber background subtraction. The "instrument background", defined for the gas phase as the periodical flushing of the instrument with UHP N2 while sampling and, for the particle phase, as the 60-90[th] second of each desorption cycle, were subtracted from the measurements to account for any potential instrument contamination. The chamber background, defined for the gas phase as the measurements conducted during the "chamber background" phase (now "chamber stabilisation" phase; i.e., chamber filled only with clear air in the absence of light) were subtracted from the measurements to account for any potential background gas phase species

existing in the chamber prior the addition of any of the reactants. For the particle phase, the chamber background was defined as the desorption cycle measured during the "dark uncreative" phase (i.e., chamber filled with seed, $NO_x$ and VOC in the absence of light), was subtracted from the measurements to account for any potential traces of particle phase species as well as any potential influence of the seed aerosol. We have attempted to make this more clear in the revised version of the manuscript (see section 2.4.1 of the revised manuscript) as:

*"The gas-phase data obtained during "instrument background" mode (see Section 2.3 and Fig. S5b) were averaged in background cycle and subtracted from the nearest in time measurements. To account for any potential chamber contamination, the gas phase sampling data obtained during the "chamber stabilisation" phase were also subtracted from the data obtained during the "experiment" phase (see Section 2.2 and Fig. S5). For the particle-phase data, a similar procedure was followed. In certain cases, a high initial particle phase signal was observed as it was carried out from the gas phase sampling, with a desorption profile not being evident (see Fig. S5c; examples #1 and 4). The reason for that is probably the instrument being contaminated by the high levels of "sticky" compounds measured during the gas phase sampling. In other cases, the high initial particle phase signal was depleted within several seconds and a desorption profile was evident (see Fig. S5c; examples #2, 3 and 5). To correct for these interferences a corresponding particle-phase "instrument background" period was assumed. We selected the 60-90th second of each desorption cycle as a particle-phase "instrument background" period (Fig. S5c) because (i) the desorption temperature was still at ambient levels (<22oC; Fig. S6b), implying that the thermal desorption process had not yet started and (ii) this period avoided the "noisy" signal that appeared during the first 10-15 s of each desorption cycle produced by the IMR pressure change due to the actuator shifting position (Fig. S6a). The average signal of the particle-phase "instrument background" period was then subtracted from the raw desorption data and the resultant thermograms (signal vs. desorption temperature) were integrated using the trapezoid rule. Subsequently, the integrated area of each peak obtained during the "dark unreactive" phase (see Section 2.2; Fig. S5a) was subtracted from the data to remove any seed and/or filter effect. All the subsequent analysis presented in this work is using the background-corrected data."*

16) Line 253ff: "…high initial particle phase signal was observed…" Did the authors scale the signals to the primary ions (I- species)?

The analysis presented here was not scaled to the reagent ions, therefore the initial high particle phase signal could not be attributed to this. As shown in our response on major comment #2, the high initial particle phase signal was carried over from the gas-phase sampling.

17) Line 266-275: This whole paragraph is unnecessarily difficult to understand. See next three comments.

The whole paragraph was re-written so the product classification procedure is more clear, while a flow chart was added in the revised SI (Fig. S7). For convenience, the revised text and supplementary figure are copied below.

*"To identify the products that are uniquely formed in the mixed VOC systems, as well as those that uniquely correspond to either precursor in the mixed system, we compared the products identified in single precursor experiments to those of the mixture as follows. First, the compounds with the same assigned formulae that were found in all systems (i.e., both single precursor and mixture) were classified as "common". Subsequently, the remaining products that were found in each single precursor system and in the mixed system were classified as unique products of either precursor in the mixture. All the remaining products, that were not found in either single precursor system nor were classified as "common", were classified as "initially unique to the mixture". Finally, in order to take into account any potential products which were not identified in single experiments but still uniquely correspond to either of the two precursors and were potentially miscounted as "unique to the mixture" products, we compared the products categorised as "unique to the mixture" with the Master Chemical Mechanism (MCM v3.2;Saunders et al., 2003;Jenkin et al., 2003). More specifically, the products initially classified as "unique to the mixture" were compared with the product lists of α-pinene and o-cresol found in MCM and any products that were found to uniquely match any of the records or were found common between all lists were subsequently moved to the corresponding individual precursor's or common lists. The above procedure is shown schematically in Fig. S7."*

[Figure]

*Figure S7: Schematic of the formula separation procedure to classify the products of the mixture as products deriving from either precursor (i.e., α-pinene/o-cresol), common (i.e., products with the same assigned formulae found in all systems) and unique products of the mixture.*"

18)     Line 268f: "…products with assigned formulae which were common among all systems and CIMS is unable to resolve (i.e., isomers)…" This means to me that the "common" category consists mainly of isomeric compounds, i.e., that there are even less true common compounds between a-pinene and o-cresol. Is that what the authors want to imply?

Common category includes the products that were having the same assigned elemental formulae and were observed in all systems. Therefore, these products could not be assigned as products of o-cresol, α-pinene or unique in mixture. These products could be isomers, thermal decomposition products or oxidation products with common elemental formula. In the revised manuscript this is clarified as:

*"First, the compounds with the same assigned formulae that were found in all systems (i.e., both single precursor and mixture) were classified as "common".*

19)     Line 270ff: "…we compared the resultant products categorised as unique to the mixture with the Master Chemical Mechanism." What is compared to the MCM? The compounds

grouped into "common"? What are they compared to? MCM using a-pinene as input? o-cresol? A mixture?

Only the "initially unique to the mixture" compounds were compared with both the a-pinene and o-cresol MCM lists. This was an additional step to investigate whether the compounds that were initially assigned as "unique to the mixture" were corresponding to any of the known a-pinene or o-cresol products that were not assigned/observed in the single precursor experiments. In the revised manuscript this sections reads as:

*"Finally, in order to take into account any potential products which were not identified in single experiments but still uniquely correspond to either of the two precursors and were potentially miscounted as "unique to the mixture" products, we compared the products categorised as "unique to the mixture" with the Master Chemical Mechanism (MCM v3.2;Saunders et al., 2003;Jenkin et al., 2003). More specifically, the initially classified as "unique to the mixture" products were compared with the product lists of α-pinene and o-cresol found in MCM and any products that were found to uniquely match any of the records or were found common between all lists were subsequently moved to the corresponding individual precursors' or common lists."*

20)     Line 270ff: Why would a single precursor product expected from MCM calculations not show up in the single precursor   experiments but only in the mixture experiment?

The formula assignment in the FIGAERO-CIMS is ambiguous. Our intention here is to ensure, to our best capability, that any product classified as "unique to the mixture" actually corresponds to unique products formed in the mixture and they are not products that might have assigned differently by the user between the different experiments/systems. For example, if we take a hypothetical scenario of HR peak fitting in the 299 m/z, in the α-pinene single precursor experiment there are two possible suggested formulas, the $C_9H_{16}O_3$ and the $C_8H_{12}O_4$, with fitting errors 2.88 and 1.75 ppm, respectively. In this case, the user may select to fit the ion with the lowest error, i.e., the $C_8H_{12}O_4$. In the mixed VOC experiment, at the same m/z, the same two ions appear but the $C_9H_{16}O_3$ has lower fitting error than the $C_8H_{12}O_4$, so the user assigns the $C_9H_{16}O_3$. Both products can be found in the MCM as α-pinene products. Consequently, in this hypothetical scenario, both products are likely α-pinene-derived products however, their different assignment between the experiments might lead to their classification as "unique in the mixture" experiments. Therefore, in order to account for such differences and/or potential assignment errors, we introduced an additional step and compared the initially "unique to the mixture" products with the MCM. The procedure can be seen in our responses to the comments above as well as the Fig. S7 in the revised SI.

21)     Line 277-288: This is another example where the authors do not clearly state what they did: Which data was used for the MFR and volatility model calculations? The mass concentration from AMS or SMPS? If AMS was used, what CE&RIE were used? How well did AMS and SMPS total mass agree?

We understand that the readers might benefit from a more detailed description of our procedure. As mentioned in the response of the minor comment #10, and in the revised manuscript (L233-236). The MFR was calculated using the SOA mass (i.e., organics) from AMS.

The applied RIE was obtained based on the ion balance of ammonium nitrate and ammonium sulfate, obtained by calibrations using monodisperse ammonium nitrate and sulfate particles (350 nm) and for $NH_4^+$ and $SO_4^{2-}$ were $3.57 \pm 0.02$ and $1.28 \pm 0.01$, respectively.

A real-time collection efficiency (CE) was applied to the data of each experiment by comparing the HR-AMS total mass with the total mass from the SMPS multiplied by an effective density, based on the organic/inorganic ratio from the HR-AMS, assuming densities of 1.77 and 1.4 g $cm^{-3}$ for the inorganic and organic fraction, respectively. The CE for ammonium sulfate, determined in the "dark unreactive" period was found to be ~0.3-0.5, in line with previous findings (Alfarra et al., 2004), and the more SOA condensed the CE increased to reach almost unity (i.e., ~0.9-1.1), also in line with previous of observations (Matthew et al., 2008).

The above procedure has been included in the revised manuscript (L213-223).

22)     Line 277-288: What is the uncertainty of the derived volatility distribution for this specific setup?

The error bars shown in Fig. 5 of the original manuscript represent the model uncertainty. This has been clarified in the revised caption of that figure (now Fig. 4) in the revised manuscript.

23) Line 291: "Figure 1 shows the average distribution…." What was averaged? The authors have the reader guessing if the averaged multiple FIGAERO samples from one experiment. Or did they combine data from multiple experiments?

We see the confusion of the reviewer regarding our averaging procedure. Briefly, all the available FIGAERO-CIMS samples collected at the last cycle (t>5h) for all the experiments in each system were averaged. Subsequently, due to the low variability amongst the repeat experiments, certain experiments were randomly selected as characteristic, while figures with the average values are provided in the revised SI to support these statements. This has been clarified in the revised manuscript (L191-198) as:

*"As shown in Table 1, a number of repeat experiments were conducted for each system in order to increase confidence in the validity of our results as well as to overcome technical difficulties due to instrument failures over certain experiments (see Table S1). The repeat experiments showed good agreement, as it is depicted in the maximum SOA masses (Table 1), as well as from the relatively low standard deviations of the average values within each system for a number of parameters such as the carbon and oxygen number distribution, the sum thermograms and the mass fraction remaining (see Fig. S1-4). In the sections below, three experiments were randomly chosen as characteristic; Experiments 2, 6 and 10 (Table 1) to represent the single α-pinene, single o-cresol and mixed VOC, respectively."*

24) Line 294: "…unidentified fraction (se Section 2.4)" Section 2.4.1 states that more than 80% of the total fitted signal was assigned (Line 245). But there is no explanation about the unidentified fraction. Why is it unidentified? is there no reasonable formula possible? Is it outside of the trusted m/z range?

As stated in the section 2.4.1, L244-245 of the original manuscript, we initially sorted all the unit mass resolution (UMR) peaks based on their contribution to the total signal and we began the HR peak identification in descending order, until ≥80% of the total signal has been assigned. At the point of ≥80% of the total signal has been assigned, each individual UMR peak contributed something in the order of <0.3% of the total signal and each individual HR peak had no reasonable possible formulas within our trusted error range (i.e., <6 ppm). For the benefit of the readers, we have added this information in the methods section (sec 2.4.1) of the revised manuscript (L282-284) as:

*"The remaining unidentified individual UMR peaks contributed minorly to total signal (≤0.3%) and no possible formulas were available within a reasonable fitting error (i.e., < 6 ppm), while 30-50% of it had low signal to noise ratio (S/N≤2)."*

Furthermore, we have investigated the unidentified fraction and it was found that the majority of the unidentified peaks was barely above, if not below, the detection limit of our technique (i.e., signal to noise ratio ≤2; Figure 6 below). This corresponds to ~6-11% of the total particle phase signal or ~30-50% of the total particle phase signal of the unidentified fraction. Clearly, this unreliable fraction should have been removed from our analysis in the first place. Considering further that we are unable to link the volatility of this fraction with the chemical composition information, which is a main aim of this work, we decided to exclude the results relating to the unidentified fraction from the revised paper. This is a common practice followed in the literature (e.g., Mohr et al., 2019;Le Breton et al., 2019;Lutz et al., 2019;Mehra et al., 2020). We are now stating this in the revised manuscript (L284) as:

*"In this work, only the species with assigned formulas are considered."*

[Figure]

**Figure 6:** Signal to noise ratio (S/N) for all the identified (left panels) and unidentified (right panels) fractions, separated to show peaks with low or high S/N (i.e., S/N≤2 or S/N>2, respectively), for an example experiment in each system (α-pinene, top panels; o-cresol, middle panels; mixed VOC, bottom panels).

25) Line 301f: The two main products for o-cresol oxidation are C7H8O2 and C7H7NO4. These are formally a H abstraction and OH addition or H abstraction and O&NO2 addition. This seems to suggest that very little oxidation occurs for this precursor system. How does this compare to the expectation stated in Lines 102ff. Could this behaviour be linked to high VOC and NOx concentration in relation to OH (see mayor comment 1)?

In all of our experiments, the parent VOC was not fully consumed within the experimental our timeframes (see Figure 7 below and now Fig. S10 in the revised SI). As a consequence, the

high contribution of the early generation products might be expected throughout the duration of the experiments. The high contributions of these products might suggest either that indeed, little oxidation occurs in the system and/or that there is a very high sensitivity of the CIMS towards these products. The high VOC-NO$_x$/OH might partly be the reason for the high contributions of those products in the former case, whereas the latter is challenging to assess. Nonetheless, the strong presence of C7H7NO4 as well as the appreciable fractions of later generation products, such as the C7H7NO5 and C7H7O4 in the particle phase (see Fig. S6 in the original manuscript or Fig. S9 in the revised manuscript), suggests that there is not much oxidation needed before the compounds enter the particle phase in this system. In order to include this discussion, a few sentences have been added to the revised manuscript (L393-401) to include this that read as:

*"These products are likely result of H abstraction and subsequent OH and ONO2 addition, respectively, suggesting that they are early generation products. The high contributions of these early generation products can be associated with the fact that the VOC precursors were not fully consumed within our experimental timeframes (Fig. S10), likely due to the high VOC:OH. As a result, it may be expected that they are continuously formed throughout the duration of the experiment. Alternatively, the strong signal contributions of these products might indicate that our technique is particularly sensitive to these products. In the particle phase of both single precursor systems, a greater fraction of the signal is associated with products having more carbon atoms than each precursor, compared to the respective gas phase contributions."*

[Figure]

**Figure 7: Average (±1σ) normalised VOC decay in all the single and mixture experiments.**

26) Line 308f: The contribution of C>10 compounds is not significantly different between the gas and particle phase of the mixture. But are these really the same compounds? Are the compounds in the particle phase more oxidised? What is the volatility of the particle phase compounds (derived from Tmax of their ion thermograms)? Are they expected to be semivolatile and thus show contribution in both gas and particle phase?

In the revised figure 1 it can be seen that these compounds have higher signal in the particle than in the gas phase and they have the same elemental formula (as depicted by their nC vs OSc). The volatility of these products based on the partitioning approach is in the SVOC region (see Fig. 6 in the revised manuscript), so they are expected to appear in both phases. Section 4 in the revised manuscript provides a thorough discussion on the volatility and its links to the chemical composition of the identified products.

27) Line 311ff: Are C7H8O2 and C7H7NO4 still the dominant species for the o-cresol group? I.e. what fraction of the o-cresol group is attributed to these two compounds. Is this similar to what was seen for the single precursor experiment.

As can be seen from the Fig. S6 in the original SI (or Fig. S9 in the revised SI), in the mixed VOC system the C7H7NO4 is the dominant species of the o-cresol group in the particle phase. In the gas phase however, C7H7NO3 and C7H8O2 were the most dominant species. Therefore, it seems that in the gas phase of the mixed system the contribution of the C7H8O2 is lower than the C7H7NO3 where the opposite effect is observed in the single precursor system. However in the particle phase, the C7H7NO4 is consistently the dominant species.

28)     323: I do not understand this sentence. I guess this is supposed to be an interpretation of O:C ratio?

We agree that this sentence was confusing. In the revised manuscript, we have omitted this sentence.

29)     Line 326 ff: The O:C(AMS) is smaller than the FIGAERO value for a-pinene SOA but larger for o-cresol SOA. For mixed SOA the effect seems to be cancelled. Could this change in trend be caused by the AMS using a parameterisation optimised for ambient data?

Considering that the setup of HR-AMS was the same in all experiments, we cannot see this being the likely reason for this discrepancy. On the other hand, the detection efficiency of the CIMS towards compounds with different characteristics can vary substantially, while it can only measure a subset of the total SOA mass. It should be further stressed that the O:C values from the CIMS are displayed as weighted averages based on the signal contribution of each product in the particle phase. This suggests that compounds with high signal contributions will affect the weighted average values. For example, in the o-cresol system, the C7H7NO4 that

accounts for 40% of the total particle phase signal is determinant for the weighted average value. In the mixed VOC system the product(s) with the same elemental formula account for 26% of the total particle phase signal, thereby its effect is less pronounced. Therefore, the O:C values derived from the CIMS is strongly related to the products measured and their signal contributions that may be also affected by the efficiency they are detected.

30)    Line 328 – 331: Again, an example for unclear language. The main challenges for FIGAERO-CIMS are the sensitivity effects, while the AMS mostly struggles with the fragmentation (i.e. detecting organo-nitrates mostly as NO+ or NO2+). But the sentence structure implies that both problems apply to both instruments. This does not impact the overall message, but this sloppy style may be misleading for a novice to AMS and FIGAERO-CIMS while annoying the expert.

We understand that this might have been unclear in the original sentence. The sentence has been revised to read as:

*"Particularly, the N:C quantification in the FIGAERO-CIMS can be affected by the differential sensitivity of the instrument (Iyer et al., 2016), while for the HR-AMS, the extensive fragmentation which is not readily characterised for the N-containing fragments (Farmer et al., 2010) can affect the estimated N:C."*

31) Line 339ff: What is meant by "…provides an indirect measure of the volatility as a function of the FIGAERO-CIMS total particle phase signal"? Is this supposed to describe the meaning of the sum thermogram (sum of all ion signals as a function of desorption temperature)? "Volatility as a function of the total particle phase signal" means something else to me.

We see the confusion of the reviewer in this sentence. The results and discussion have been changed according to the major comments and this sentence has been deleted in the revised manuscript as the volatility from the FIGAERO-CIMS is now expressed in the VBS framework.

32) Line 342: "average sum thermogram for each system" Again, what was averaged? All FIGAERO samples from one experiment? All samples from all experiments? How many samples are part of the averaging? (see also comment 34) 33) Line 342: For the average values used before ±1σ was used. Why is ±2σ used here?

We thank the reviewer for pointing out this typo, it should be ±1σ. Our averaging procedure was explained in the response to the minor comment #23. Briefly all the available samples collected at t>5h in each repeat experiment were averaged. Further, this figure was not

representing correctly the variations observed in the o-cresol experiment and we thank the reviewer for pointing this out. The correct figure showing the standard deviations as errorbars is now included in the revised SI (see Fig. S3 in the revised SI).

34) Line 345: "characteristic experiment" why were these experiments chosen as characteristic? The authors should clearly state (best in the methods part):

a.      what were the averages they used (averaged over one experiment or multiple experiments?)

b.      how many FIGAERO samples where averaged?

c.      why certain experiments were chosen as representative for a SOA system.

d. mark the "characteristic experiments in Table 1.

As also stated in the response to the comment #23, certain experiments were randomly chosen as characteristic, considering that the repeated experiments exhibited very low variability in the chemical composition and volatility. A number of supplementary figures have been added in the revised SI to back up such statements (see Fig. S1-4 in the revised SI). It should be noted that the characteristic experiments were marked in the original manuscript and were referred to all figure captions or text by their experiment number. All the available TD-AMS samples and the FIGAERO samples collected at t>5h in each repeated experiment were averaged. The number of samples averaged varied for each parameter as certain instrument failures, as for example, the failure of the FIGAERO inlet on experiments #3-5, prevented the collection of particle phase data in these experiments. In order to clarify this, we have added a table in the revised SI with the instrument availability in each experiment (see Table S1 in the revised SI) and a relevant text in the methods section(L191-198)  that reads as:

*"As shown in Table 1, a number of repeat experiments were conducted for each system in order to increase confidence in the validity of our results as well as to overcome technical difficulties due to instrument failures over certain experiments (see Table S1). The repeat experiments showed good agreement, as it is depicted in the maximum SOA masses (Table 1), as well as from the relatively low standard deviations of the average values within each system for a number of parameters such as the carbon and oxygen number distribution, the sum thermograms and the mass fraction remaining (see Fig. S1-4). In the sections below, three experiments were randomly chosen as characteristic; Experiments 2, 6 and 10 (Table 1) to represent the single α-pinene, single o-cresol and mixed VOC, respectively."*

35)      Line 360 "modest volatilities" modest is the wrong word here. "moderate" is probably more fitting.

We thank the reviewer for pointing out this typo. The correction has been made in the revised manuscript.

36)     Line 373: Unidentified ions was not clearly introduced. Was it just not possible to assign a sum formula? Are these the same masses as the unidentified in the a-pinene and o-cresol case?

In our response to the reviewers' comment #24, we have introduced the unidentified fraction as well as the reasoning for excluding it from the revised manuscript.

37)     Figure 5: How different were the total aerosol mass concentrations ("cOA") in the three experiment types? How much will these different loadings impact the observed volatility distribution of the SOA particles?

The total aerosol concentrations (seed + SOA) are now shown in Fig. S15 in the revised SI and for convenience are shown in the figure 8 below. As can be seen, at the end of each experiment (ie, t>5h), the difference between the o-cresol and mixed VOC system is ~20-30%, while the total mass in the a-pinene systems is higher by a factor of ~2 compared to the other systems. Considering that the gas to particle partitioning can be affected by the available absorptive mass (in our case total mass; cOA), the different cOA may be expected to affect the volatility distributions. However considering that a change in the volatility is proportional to change in the cOA (Pankow, 1994), it will take an order of magnitude difference in the cOA to change the volatility by 1 order of magnitude. Therefore, given that the SOA particle volatilities are expressed in logarithmic bins in the VBS framework, it is not expected the differences in the cOA, which are much less than an order of magnitude in difference, to substantially affect the observed volatility distributions. This is being discussed in the Section 4.2 (L690-694), that reads as:

 *"A potential explanation for this discrepancy is the higher SOA mass loadings and thereby the total absorptive mass, in the mixture experiments compared to the single o-cresol (Table 1) that could have favoured the partitioning of the more volatile species. However, the difference in the total mass between the two experiments after the 5th hour was relatively small (~20-30% μg m-3; Fig. S15) to observe differences in the volatilities expressed in logarithmic bins."*

[Figure]

**Figure 8:** Total aerosol mass (seed+ SOA) in the experiments in all systems (std. deviations are shown as shaded areas)

38)     Line 406: What does "large size of the molecules" refer to? High Mw? Higher carbon number?

The larger size refers to the molecular weight of the products. This has been clarified in the revised manuscript (L637-638), where the revised Fig. 1 also supports these statements.

39)     Line 407: what does "close to unity" mean in this context. The values were close to 1? The values from the two different systems were almost the same?

We agree that this statement was confusing. Mass accommodation values above 0.1 suggest modest influence of resistances to mass transfer in the temperature range of the TD. We rephrased this sentence in the revised manuscript while we added few references to support our statement (L494-496) that read as:

*"The mass accommodation coefficients were above 0.1 in both systems (0.36 for the α-pinene and 0.23 for the o-cresol  SOA;  Fig. S11) suggesting only small resistances to mass transfer in the temperature range of the TD (Riipinen et al., 2010;Saleh et al., 2011)."*

40)     Lines 413-427: What are the uncertainties of the O:C enhancement ratio? It looks like that it is possible to pick one out of the two experiments for each category and get almost overlapping slopes. I.e. the highest points from a-pinene and the lowest points from o-cresol almost overlap.

Indeed, the O:C enhancement ratio, particualrly in the o-cresol system showed high variability between the experiments. Additional discussion has been added to the revised manuscript (L514-516) to soften such arguments that reads as:

 *"It should be noted however that in the o-cresol single system, the O:C enhancement ratio in this range showed high variability among the experiments, suggesting that it may not be substantially different than that of the mixed VOC system."*

41)     Line 414: "O:C enchantment" = enhancement?

We thank the reviewer for pointing out this typo. The correction has been made in the revised manuscript.

42)    415: The term "BP line" was not introduced. Is that the bypass line for the thermodenuder? By now I am a bit tired of having to guess what the authors are trying to say.

The abbreviation BP for bypass, has been clarified in the revised manuscript as:

*"In our configuration, the TD was operating at temperatures ranging from 30 to 90 $^oC$ in 12 steps and the HR-AMS and SMPS sampling alternated between the bypass (BP) and TD every 4 and 6 min, respectively, using a 3-way electronic valve."*

43) Line 425: "…whereas that of the mixed system was approximately between." What doe you mean by approximately between? The values for the mixed system a definitively between the values for the single precursor systems? Is this supposed to be "approximately half-way between"?

The word "half-way" has been added to sentence in the revised manuscript.

44) Line 426f: "…the less volatile SOA particles from the o-cresol system are considerably more oxygenated compared to the average than those of the α-pinene system." What is compared to what here? What is the "average"?

We see the confusion that this statement might have caused to the reviewer. The O:C enhancement ratio in the MFR<0.5 region was higher compared to that in the α-pinene experiments that resulted in a higher O:C enhancement slope. This suggests that the less volatile SOA particles from the oxidation of o-cresol are likely more oxygenated compared to those derived from the oxidation of a-pinene. The sentence has been revised (L512-514) to read as:

*"These results indicate that the less volatile SOA particles from the o-cresol system (MFR≤0.5) are likely more oxygenated compared to those of the α-pinene system."*

45) Line 436: The variable "maximum desorption temperature" was not properly introduced anywhere. This term can be misleading to a non- FIGAERO expert. As described in other comments, the authors should clearly introduce the relationship of Tmax (desorption temperature of the maximum of the ion/sum thermogram) and volatility.

We agree that this needs to be clarified in the revised manuscript. In the methods section (L339-340), we have added a sentence that reads as:

*"Briefly, in the former approach, the compound's maximum desorption temperature (Tmax) in the FIGERO-CIMS can be related to their enthalpy of vapourisation (Lopez-Hilfiker et al., 2014)"*

46) Line 440: "This broad consistency…" If the authors want to express that the two volatility distributions roughly agree, then "broad" is not the right word here.

The phrase "broad consistency" has been changed to "broad agreement" according to reviewers' suggestion (L585-586).

47) Line 447-467: Do the general statements of the authors about fragmentation include the initial oxidation step? Or are they looking at the fate of RO/RO2 radicals?

In this section we attempted to be as general as possible, in order to feed the subsequent discussion. In the original manuscript (L451), we refer to fragmentation as the cleavage of the carbon-carbon bond, regardless whether this is an initial oxidation step or a radical process. In the revised manuscript this sentence reads as (L596-599):

*"Fragmentation (i.e., cleavage of carbon-carbon bond) will increase component volatility due to the decrease of the product's molar mass, though in some cases the fragmented products are subsequently functionalised (e.g., O2 addition after the scission of a double bond) leading to a decrease of their vapour pressure, with the net effect on vapour pressure not immediately predictable."*

48)     Line 447-467: For the general fragmentation vs functionalisation discussion, the authors should point out the fundamental difference in the precursor structures. a-pinene is a bicycle molecule. I.e. two C-C bonds can be broken without actually loosing a C. o-cresol is a C6 aromatic compound. I.e. the aromatic ring will be very stabile against oxidation (e.g. no ozonolysis reaction. But once the aromaticity is broken, fragmentation reactions will create two very small molecules.

We agree with the reviewer that such information might benefit the readers. Additional discussion has been added (L599-603) that reads as:

*"It should be noted that the selected VOC precursors in this study have fundamentally different structures; α-pinene is a bicyclic molecule while o-cresol is an aromatic molecule. This suggests that in the α-pinene two C-C bonds can be broken by O3 without the loss of C, while the stability of the aromatic ring makes o-cresol practically unreactive towards O3. Once however the aromaticity is broken, fragmentation reactions by O3 will occur creating smaller molecules."*

49)     Line 460ff: The statement about the sensitivity of I- CIMS with regard to volatility classes is misleading in this context. It is correct that in "gas-phase mode" CIMS mostly detects I- and SVOCs. But the particle phase mode of FIGAEO-I-CIMS detects SVOC to ELVOC compounds equivalent to $C^*$ 104 – 10-6 ug m3 or even lower. (Isaacman-VanWertz et al., 2017). 60% of the particle phase signal appears at desorption temperatures >90 °C. Even with all the differences between different FIGAERO instruments, that is in the LVOC range.

We agree with the reviewer that this statement might be misleading. Although the development of the FIGAERO has enabled the detection of less volatile compounds, here, we were not able to observe substantial fractions of compounds with volatility in that range with either the calibrated Tmax or the partitioning approach, opposed to the TD measurements (see Fig. 3 and 4 in the revised manuscript). As rightfully pointed out by the reviewer in the major comments #4 and 5, the representation of the desorption temperature as proxy to the SOA particle volatility in, more or less, arbitrary desorption temperature bins could not capture the volatility in an established classification system, such as the VBS. In our response to these comments, we used Tmax-Psat calibrations as well as partitioning calculations to express the particle volatility in the VBS framework.

Below, in Figure 9, the Tmax distributions are given for both the identified and unidentified fractions, after removing the unreliable fraction (i.e., peaks with S/N≤2; see response to minor comment #24), in example experiments in each system. As can be seen, in all systems, the largest fraction of the signal corresponds to ions with Tmax<90°C and, based on our Tmax calibrations, this leads to volatility distributions dominated by IVOC. Similarly, the partitioning coefficients measured along with the high total concentrations led to volatility distributions dominated by SVOC. The limitations of these approaches are discussed in Section 4.1 of the revised manuscript.

Considering the above, we appreciate that our statement in the original manuscript for the ability of the FIGEARO-CIMS to detect low or extremely low volatility components could be perceived as a generalisation. However, in our experimental setup and analysis methods, it appears that a very low fraction of the ions that are in our "trusted" measurement range have volatilities within, or below, the LVOC fraction. In the revised manuscript, we have included the above discussion that now reads as:

*"Although the development of the FIGAERO coupled to I--CIMS enabled the detection of particle phase compounds in the LVOC/ELVOC range (Isaacman-VanWertz et al., 2017), here, we were not able to reliably identify substantial fractions of compounds with volatility in this range using either the calibrated Tmax or the partitioning approach, in contrast to the TD measurements (Fig. 3 and 4). As discussed in Section 4.1., the observation of lower volatility species is likely related to selected method to retrieve the SOA particle volatility from the instrument and any associated assumptions and limitations. Nonetheless, here, the general agreement in the volatility trends between the FIGAERO-CIMS and the TD-AMS might indicate that the FIGAERO-CIMS can capture the changes in the SOA particle volatility across the different systems. Therefore the quantification of the $log_{10}C^*$ might be limited from the approach presented here but may be able to provide representative trends between the systems."*

[Figure]

**Figure 9**: Distribution of Tmax, for example experiments in (a) a-pinene, (b) o-cresol and (c) mixed VOC systems. In either case, the bars are separated to show the identified and unidentified fractions.

50)     Line 476f: The conclusion that dimers are not relevant because the observed fraction of C>10 compounds is low is not correct. Many dimers are thermally labile and will be detected as thermal decomposition products with sum formulas similar to the corresponding monomer composition. A clear example is shown by D'Ambro et al. (2019) for C5H12O4 which shows two distinct peaks (at 55°C and 90°C). More examples can be found in a recent overview on FIGAERO-CIMS (Thornton et al., 2020). The authors need to adjust this statement to reflect the importance of thermal decomposition for oligomers.

We agree with the reviewer that this statement might be over-simplified. Indeed, thermal decomposition is a likely explanation for the low fraction of higher carbon number products (nC>10) and may not necessarily suggest that dimers/oligomers are unimportant. In the revised manuscript, the possibility of thermal decomposition have been included in many parts of the discussion section.

51)     Line 480: "…with 7 or more carbon numbers". this should be either "with 7 or more carbon atoms" or "with a carbon number of 7 or more"

The correction has been made in the revised manuscript.

52)     Line 492: Fig S2 is not the right figure reference. Is this refereeing to Figure 1?

We thank the reviewer for pointing this out; indeed it should write Fig. 1 instead of Fig. S2. In the revised manuscript this section has been changed along with the related supporting figures.

53) 498ff: The authors compare the volatility of unique to mixture products to the rest of compounds in the mixed SOA particles with the same carbon numbers and attribute the observed differences to the change in O:C. But is the determining factor really the increase in oxidation or is it the increase in Mw?

We thank the reviewer of this suggestion. Investigation of the OSc and MW changes has been included in the revised manuscript (see Section 4.2 in the revised manuscript). Indeed, the unique to the mixture products have higher MW, OSc and O:C compared to the products deriving from either precursor. Consequently the O:C is not the only driver of the lower volatility of these products.

54)     Line 504f: "The vast majority of those products were found to be common between all experiments…" This is confusing. This is supposedly about the "unique to mixture" compounds. How can they be found in all experiments? Is it not definition that these compounds only occur in the mixture experiments?

We understand that this sentence was confusing. Indeed, the unique to the mixture products are the those that were only observed in the mixed system and not in either of the single precursor system. The discussion has been changed to be more clear (see Section 4.2.2. of the revised manuscript), while additional figures have been created to back up our statements.

55)     Line 501-514: The most important indication if these C<5 compounds are likely from thermal decomposition should not be their O:C value but rather the thermogram data (Figure 7). Are compounds with a volatility equivalent to an average Tmax > 100 °C expected to have a significant fraction in the gas phase?

We agree with the reviewer. This is clearly depicted in the Fig. 6 and 7 in the revised manuscript where the volatility of these compounds were found to span across a wide range (i.e., $\log_{10}C^*=0.5-4$), suggesting that they might originate from multiple sources, including the thermal decomposition of oligomers. This has been also included in the discussion in the revised section 4 that reads as:

*"Their broad log10C* distribution and their relatively high OSc suggests that they might originate from various pathways, including thermal decomposition processes in the FIGAERO (Thornton et al., 2020)"*

56)     Line 504ff: It is not clear what the comparison of O:C is aiming at. The unique to mixture C<5 compounds have higher O:C than the other C<5 compounds? Or the particle phase compounds compared with the gas phase compounds?

We understand that this sentence was confusing. The intention of this sentence was to show that the products with nC<5 had relatively high O:C that could explain their high desopriton temperature and thereby low volatility. Due to the necessary changes in this section in response to the reviewers' major comments, this sentence no longer exists and the maximum desorption temperature boxplots have been replaced with volatility plots in the VBS framework.

57)     Line 537ff: "…to assess the potential role of product interactions in altering the SOA particle volatility" This sentence suggests that conducting FIGAERO-CIMS measurements with a precursor mixture would be enough to achieve this. As I understand the identification method described in this manuscript, the key to assigning the compounds to the different categories (common/unique to mixture etc.) is the combination of chamber experiments with single precursors and the  mixture. The authors need to clarify what they mean in this sentence.

We agree with the reviewer that this sentence might be misleading. Indeed, combination of single precursor experiments with mixtures is essential to identify the potential of the molecular interactions to the SOA composition and properties. The discussion section has been revised according to the reviewers' comments.

58)     Lines 543-558: I disagree with the interpretation of the decreasing Tmax values with increasing filter loading. To my knowledge and according to the modelling framework presented by Schobesberger et al. (2018) artefacts related to increases in filter loading will always lead to a shift to higher Tmax values. Previous studies have shown that the largest Tmax shifts when filter loadings where <200 ng to <2.5ug and with loading >~2.5 ug, Tmax changes were minor (Huang 265 et al., 2018; Wang and Hildebrandt Ruiz, 2018). The change in the ion thermogram of C10H16O4 (Figure S5) looks more like a change in volatility distribution for that specific sum formula. I.e., there are two or more isomers with different volatility and the more volatile one is not there for the lower mass loading sample. If this is the case, this is not an artefact but a real feature of the data. Without the information about the evolution of the mass concentration in the chamber, there are three possible explanations:

a. The lower mass loading on the filter corresponds to a lower particle mass concentration in the chamber. Hence, the partitioning is affected and the volatile isomer with Tmax~40 °C does not partition into the particle phase

b.     There is ongoing chemistry in the chamber and the mass concentration reflects the evolution. At the time when the low loadings were sampled the more volatile isomer did not exist in sufficient quantities. Either because it had not formed yet or because it had reacted away.

c.     If the lower mass loadings are "later" in the experiments, the more volatile isomer may react in the particle phase to form something else, possibly a dimer. Such a dimer will either show up at the dimer formula, at the same formula at higher desorption temperature or at a different sum formula if the thermal decomposition is more complex.

We agree with the reviewer that our interpretation might be over-simplified. Indeed, the difference in the absorptive mass might have affected the partitioning of an isomer(s) and that could have caused a difference in the Tmax.

According to referees' #2 suggestions we have shorten the length of the section 4.2 (now Section 4.3 in the revised manuscript) as much as possible, as it was deviating the manuscript from its original aims and the thermograms of the $C_{10}H_{16}O_4$ at different mass loadings are no longer shown in the revised manuscript. However, the potential explanations provided by the reviewer are used to explain a similar and more meaningful example that is used throughout the manuscript, the $C_7H_7NO_4$, which was found to have much lower Tmax in the mixed VOC system compared to o-cresol (~60 vs. ~74 °C, respectively), where the total mass (and thereby the filter loading) in the former was higher than the latter.

This discussion in the revised Section 4.3 is now focused in the volatilities expressed in the VBS framework.

59) Line 569: "…as few as 10 products" This suggests that 1 sum formula is just 1 compound is not strictly true. The authors should formulate this more carefully

It is true that each of these 10 elemental formulas may contain a number of isomers, so technically they may be more than 10 products. In the revised manuscript we have revised this sentence to:

 "*However, a large fraction of the total gas or particle phase signal is due to as few as 10 products and any potential isomers therein (out of hundreds or thousands typically found in the mass spectra) which in turn will largely define the shape of the volatility distributions.*"

* * *
**Review: Exploring the composition and volatility of secondary organic aerosols in mixed anthropogenic and biogenic precursor systems**

This manuscript presents chemical and volatility information of SOA formed in a batch reactor from a-pinene, o-cresol and an OH iso-reactive mixture of the two. The chemical composition of individual compounds and bulk chemical properties formed in these systems is contrasted and described using FIGAERO-ToF-CIMS and volatilities are qualitatively described using the FIGAERO-ToF-CIMS and quantified using TD-AMS. This work builds on recent studies highlighting mixed VOC systems cannot be treated as the sum of their single component systems. Mixed SOA products are observed that have an intermediate volatility between the two values of the single component mixtures. This study provides valuable discussion on the differences between SOA produced in single component and mixed systems and expands upon the unknown outcomes of mixed precursor systems. As such, this manuscript should be considered for publication in ACP after addressing the following comments.

The authors would like to thank the reviewer for his/her time and effort to provide constructive comments/suggestions to our work. Your contribution is much appreciated.

Please find below itemised replies to each one of the comments/suggestions.

**General Comments**

The intent of the manuscript is clear and well contextualised and the descriptions of the experimental conditions are comprehensive. The complexity of the system and the information made available by these measurements means many descriptive details are included, which at times is not presented in the most understandable way. Regarding presentation there are three major points to address:

■Sections describing key characteristics of SOA from the different systems in terms of product signal, Mw, C number, O:C and C* (and surrogate temperature ranges) are difficult to follow and should be simplified (e.g. sections 3.2.1 and 4.1). One suggestion might be a graphical summary or table of key findings to use as a reference for the reader to follow. Additionally smaller sections in the larger blocks of text (e.g. 4.1) might also help the reader follow the discussion better.

We thank the reviewer for this suggestion. In the revised manuscript have expressed the results from the FIGAERO-CIMS in the VBS framework using a set of Tmax->Psat calibrations as well as using partitioning calculations. We have further separated the large blocks of text in the discussion Section to smaller parts so it is easier for the readers to follow.

The larger the discussion on sensitivity effects of the FIGAERO-ToF-CIMS on data interpretation (Section 4.2) the more the focus of the manuscript shifts from its original aims. While the discussion is useful and in part necessary, it might be best placed, at least partially, in the supplementary material.

We agree with the reviewer that this part may have been unnecessarily extensive, shifting the main focus of the paper. In the revised manuscript we have shorten the Section 4.2 (Section 4.3 in the revised manuscript).

Despite the detail of the results and discussion, the conclusion is very general and overly simplified. For example, there is no summary of any of the major chemical composition findings or mention of the VOCs used in the experiments. This section would be improved by summarising more of the key findings.

We agree with the reviewer that many of our major findings were lost in the conclusions. In the revised manuscript we have revised the whole conclusions section that now reads as:

[revised manuscript text omitted]

Additionally some sentences are missing words or are badly formulated which should be corrected with a read through.

We thank the reviewer for pointing this out. A thorough English language check was performed in the whole manuscript.

**Specific Comments**

1) Line 26: 37% and 39% are very close, is there an error estimate to say how different these figures really are?

Indeed, these figures were quite similar. The errorbars shown on Fig. 5 of the original manuscript (Fig. 4 in the revised manuscript) were representing the model uncertainty. This has been clarified in

the revised manuscript. The whole results and discussion Sections have been adjusted based our new approach to show the volatility distributions from the FIGAERO-CIMS to the VBS framework and the abstract has been changed accordingly.

2)Like 70: "ultra-low volatility" vs extremely low volatility

The correction has been made in the revised manuscript.

3)Line 90 (and 141): For context, put numbers on the SOA yields in main text (not just low, moderate, high).

Indicative yields have been added in both sentences in the revised manuscript (L93-96 and L146-147) as:

*"This choice of precursors and contributory reactivities enabled investigation of SOA formation with mixed systems containing a low, a moderate and a high yield precursor (i.e., isoprene; 0-4%; Carlton et al. (2009), o-cresol; 7-12%; Smith et al. (1998) and α-pinene; 20-30%; Eddingsaas et al. (2012a), respectively) with equal initial chances of reacting with the dominant oxidant and equal contributions of first-generation oxidation products."*

And

*"Here, we focus on the α-pinene and o-cresol system, as representative biogenic and anthropogenic VOC precursor sources (Hallquist et al., 2009;Schwantes et al., 2017), both with appreciable SOA yields (7-30%; e.g., Henry et al., 2008;Eddingsaas et al., 2012a)."*

4)Line 128: What is electronic capture device grade? Is there a purity associated with that?

We use N4.8 ECD grade nitrogen with purity 99.998%. This has been included in the revised manuscript (L134).

5)Line 147 – 152: It would be helpful to include an equation to demonstrate the iso-reactive conditions in the chamber (i.e. concentrations and reaction rates).

Indeed, an equation might be beneficial to the readers (eq. 1). This has been included in the revised manuscript (L166).

6)Line 175: Is there a difference between e.g. experiments 1-4 or are these repeats? Should reacted VOC be included as well (for those interested in yields)?

The experiments conducted under the same conditions are indeed repeats. This has been clarified in the revised manuscript as:

*"As shown in Table 1, a number of repeat experiments were conducted for each system in order to increase our confidence in the validity of our results as well as to overcome technical difficulties due to instrument failures over certain experiments (see Table S1)."*

Further, the VOC consumptions are given in the Fig. S10 in the revised SI.

7) Line 214: How were line losses minimised by using 2m long PTF tubing and a 2 sL min$^{-1}$ flow?

PTFE tubing is considered more appropriate for sampling gases compared to stainless steel tubing, while our sample flow rate results in a residence time of the sample of ~<2 s, which is lower than the recommended times by the US EPA (20 s; USEPA, 2011). Nonetheless, it is true that in our case, it is unknown whether the line losses were minimised. In the revised manuscript, we omitted this statement and the sentence is now reads as:

*"The gas phase was sampled through a separate line, using a 2 m long, ¼-inch PTFE tubing (Sulyok et al., 2002;Mittal et al., 2013) at 2 sL min-1 that results in a sample residence time of <~2s"*

8) Line 217: What is the significant of the "known concentration mix"?

Indeed, the known concertation mix is insignificant here. The sentence has been changed in the revised manuscript (L253-255) as:

*"The reagent ions were produced by passing a mix of CH3I and UHP N2 over a $^{210}$Po radioactive source and passed directly into the IMR."*

9) Line 217: Formatting $^{210}$Po

We thank the reviewer for pointing out this typo. The correction has been made in the revised manuscript.

10) Line 244: You only mass calibrate to 381 m/z but peak fit up to 550 m/z, how confident are you of the peak fitting accuracy beyond 381 if you have no higher mass calibrant?

We agree with the reviewer that any peak assignment above 381 m/z can be erroneous. Nonetheless, in all the experiments conducted the fraction of the signal in the >381 m/z region was <2% of the total particle phase signal and <1% of the total gas phase signal indicating that any assignment in that region would have negligible effect in the reported results expressed as signal fraction.

11) Line 246: Why Kendrick mass defect and not just mass defect?

We agree with the reviewer that it is no longer necessary to state "Kendrick's" mass defect, so in the revised manuscript the word "Kendrick's" was omitted from the sentence.

12) Line 248: What interferences occur during the soak period? Does this affect your integrations?

We agree that the word "interferences" might not be appropriate in this context. The soak period is designed to ensure that any remaining organic compounds would have been desorbed from the filter and that the filter is "clean" for the next cycle. In order to integrate the signals during the

desorption cycle a linear increase in the desorption temperature is required (e.g., Buchholz et al., 2020), therefore the measurements conducted during the soak period were not included here. We have altered the wording of the original sentence to read as:

*"Further, note that only the particle phase data collected during the temperature ramp were considered here as the integration of the data requires a linear increase in the desorption temperature (e.g., Buchholz et al., 2020)."*

13) Line 265: Typically particle phase backgrounds have been reported using a preconditioned filter (e.g. Bannan et al., 2019) or passing the sample through a filter to remove aerosol (e.g. Lutz et al., 2019). The method described here is unusual and I am uncertain it would capture correctly the instrument response. A background desorption cycle is needed.

We appreciate that our initial description might not been adequate in this section. Similarly to Bannan et al., 2019 and to Lutz et al., 2019, a background desorption cycle was performed in each experiment and subtracted from the measurements. Particularly, the desorption cycle obtained during the "dark unreactive" phase, where only VOC, NOx and seed in the absence of light was used a background correction cycle. We have clarified this in the revised manuscript (L303-305) as

*"Subsequently, the integrated area of each peak measured during the FIGAERO-CIMS desorption cycle during "dark unreactive" phase (see Section 2.2; Fig. S5a) was subtracted from the data to remove any seed and/or filter effect."*

14) Line 305: Is the word total correct? Sum of gas and particle?

We see the confusion that these statements might have caused. We have altered all of our statements in the revised manuscript to explicitly state whether the total signal corresponds to the gas or particle phases.

15) Line 328: Does the O:C calculated for figure 2 only account for CHO compounds? Are you more confident that the sensitivity of the CHO is not as variable as CHON?

The O:C in this figure was calculated for all the products identified (ie both the CHO and CHON) and was weighted to the signal of each compound to the particle phase. This has been clarified in the revised manuscript as:

*"Figure 2 shows the average (± 1σ) O:C ratio of all the products identified in each system from the FIGAERO-CIMS in the particle phase, weighted to the contribution of each product to the total particle phase signal, along with O:C ratio derived from the HR-AMS."*

16)Line 342: Did you try applying gas/particle partitioning to derive C* from the FIGAEROToF-CIMS? This section I find difficult to follow because there is so much information, particularly the description of figure 4. I find the inserted bar chart contains more interesting information than the three main pie charts. Would you better express your point if you had a relative signal fraction as well as the absolute signal fraction in the bar chart? Also the temperature ranges and their corresponding descriptions are mixed, and confused with general descriptions of volatility e.g. 90°C = "less volatile"

but later within this section terms like 'low volatility' are used which I don't think are meant to refer to these ranges. Perhaps using the temperature ranges or acronyms might better separate these categories from general descriptions as well as the rationale for their use.

We agree with the reviewer that the selected presentation of the results in the original manuscript might have been difficult to follow and the limits that were chosen to represent the more or less volatile fraction were, more or less, arbitrary. There are multiple ways to derive the C* from the FIGAERO-CIMS, such as the gas/particle partitioning, using explicit Tmax->Psat calibrations with compounds with known vapour pressures (e.g. Bannan et al., 2019) or by using empirical formulations (eg. Mohr et al., 2019). In the revised manuscript we have expressed the volatility distributions derived from the FIGAERO-CIMS in the VBS framework using Psat-Tmax calibrations as well as from partitioning calculations and we discuss the implications of the selected method to the results in the Section 4.1. In this way, we avoid the general descriptions of volatility and we use the well-established VBS framework to discuss our results. Particularly, the volatility distributions derived from the FIGAERO-CIMS in the mixed VOC system are separated to show the contributions from the products of either precursor, the common, as well as those that were unique to the mix (see Fig. 3 in the revised manuscript). Furthermore, additional figures (see Fig. 6 and 7 in the revised manuscript) were made and to illustrate the effect of each category of compounds to the volatility of the mixture.

17)Line 380: The unidentified fraction grows for the >90°C mixed system. Do you state what these might be e.g. inorganics, sampling artefacts, deprotonated organics? You should be more explicit in how you define the signals you have investigated i.e. I⁻ adducts and/or deprotonated and what atoms you consider (CHON etc.). It might be that the unknown signal is an unimportant grouping and better removed as you are only interested in the identifiable CHO and CHON I⁻ adducts.

We agree with the reviewer that the unidentified fraction could induce further complications to our analysis and deviate the manuscript of its main goals ie., to link the SOA composition and volatility.

After interrogating our unidentified fraction, it appears that a considerable fraction of it (~up to 50%) is composed by ions barely above or below our detection ability (i.e., signal to noise ratio ≤2) Consequently, also in line with your suggestion, we have decided to exclude the unidentified fraction from our analysis and focus the revised paper on the identified fraction.

18)Line 426: "…volatile SOA particles from the o-cresol system are considerably more oxygenated compared to the average than those of the a-pinene system". This statement is a good example of an interesting finding being lost in the discussion and conclusion.

We agree with the reviewer that in the original manuscript some of the interesting results were lost in the discussion and conclusion. These sections have been revised to emphasise the main results.

19)Line 426: Figure 6. Why are the gradient terms of the lines of best fit negative?

The gradients of these lines on figure 6 of the original manuscript (Fig. 5 of the revised manuscript) are negative as the x-axis is plotted in reverse order (MFR=1->0).

20)Line 432: As no volatilities are derived from the FIGAERO-ToF-CIMS, this might be a good section to explain why you are comparing C* derived from the TD with qualitative descriptions of volatility from FIGAERO-ToF-CIMS. Can you comment on any discrepancies between TD measurements up to 90°C vs FIGAERO-ToF-CIMS up to 200°C? Some qualification on their comparison would be useful for the reader to assess the differences in information these similar techniques provide.

We agree that this approach was not adequate and these temperature limits were more or less arbitrary. It is very difficult to relate the desorption of the compounds in the FIGAERO-CIMS with the evaporation of the SOA in ~30s in the TD. In the revised manuscript we have expressed the FIGAERO-CIMS results to the VBS framework so a related comparison can be achieved in an established classification system.

21)Line 474: "Their relatively low contribution to the total FIGAERO-CIMS signal might indicate that they will make little contribution to the mass of the C*≤1 µg m$^{-3}$...". Generally I would be careful equating uncalibrated CIMS signal directly to mass and focus more on the relative differences between the cases.

We agree that this comparison might sound misleading, given that our CIMS was uncalibrated. Due to the change of the results and discussion, this sentence no longer exists in the revised manuscript. Nonetheless, we have soften such arguments throughout the revised manuscript.

22)Line 478: OH addition or abstraction?

As has been observed previously (Schwantes et al., 2017;Olariu et al., 2002), the main oxidation pathway of o-cresol leading to ring-retaining products proceeds through the OH addition and H abstraction to form a dihydroxy toluene ($C_7H_8O_2$).

23)Line 489: What is the "formula separation technique" here referring to?

Our formula separation technique refers to our method to identify the compounds that are deriving of each precursor, are common or unique in the mixture (as stated in the Methods section). This has been clarified in the revised manuscript (L307-320) and a supporting flow chart has been added (Fig. S7 in the revised SI).

24)Line 499, 510: What is meant by the word "bulk" in these contexts?

In the sentence in the L499 of the original manuscript the bulk was defined as the sum of the "individual precursor's plus common products in the mixture" (shown in brackets).

25)Line 506: These descriptions of high O:C, high desorption temperatures and lower volatility are too vague.

The discussion has been changed in this section owning to the different analysis conducted expressing the FIGAERO-CIMS results in the VBS framework.

26) Line 519: This appears to be a key finding missed from the conclusion.

We agree with the reviewer that this might have been lost in the conclusions. The conclusions section has been revised to include this information.

27) Line 526: Discussion of signal contributions throughout this section is confusing e.g. it does not feature in figure 7; is it that important to mention here? If it is, can it be discussed separately or reformulated?

According to the reviewer's major comments, this section has been split into subsections (now Section 4.2.1 and 4.2.2), while additional figure have been created so it easier to follow.

28) Line 585: This section focuses on I- CIMS sensitivity differences to the presence of nitrated functional groups (as this has been shown to impact sensitivity by up to two orders of magnitude). The variation in sensitivity between CHO compounds can also be 2-3 orders of magnitude (Aljawhary et al., 2013), which could also affect the thermogram shapes in a similar way. Should CHO compounds be considered similarly to CHON in this calibration sensitivity test?

We agree that with the reviewer that the sensitivity of the CHO is equally important to the CHON. Given the overwhelmingly higher amount of particle phase signal observed from the $C_7H_7NO_4$ in the o-cresol system defining the shape of the volatility distribution, along with the very high contribution of CHON, our intention with this exercise was to assess the potential role of the sensitivity towards the CHON group.

According to the reviewer's major comments, this section has been shorten and this exercise has been omitted from the revised manuscript as it was deviating the manuscript from its original scope that is to provide links between the chemical composition and volatility in mixed precursor systems.

39) Line 630: These points are difficult to follow. What does "average" refer to?

Owing to the different representation of the results and discussion these statements have been altered in the revised manuscript (L808-814) as:

"The SOA particle yield from the o-cresol experiments was found to be modest (14±3%), while that from the α-pinene considerably higher (27±4%; Fig. S18), suggesting that the observed domination of the o-cresol products in the mixed systems is unlikely. Considering further the relative agreement between the FIGAERO-CIMS and the TD and the analysis presented above (Sections 4.1-4.2), we can argue that the SOA particles in the mixture, are composed by the products of the higher yield precursor (i.e., α-pinene) with considerable contributions of the o-cresol derived products, as well as those deriving from the mixing of the two precursors."

---

## Author Response (AR2)

**Responses to the Referee's #2 comments**

We thank the reviewer for his/her time and effort to provide comments on our revised manuscript.

**Corrections:**

1) **Line 85. A-pinene is not mentioned as the bVOC of study until later in the introduction and should be mentioned earlier.**

We agree that the choice of α-pinene and isoprene in addition to o-cresol in the programme of study might have not been clear in the sentence found in Line 85. We have rephrased this to read as:

"*In this work, we extend the system studied by McFiggans et al. (2019; i.e., α-pinene and isoprene) with the investigation of the interactions of an aVOC with the two bVOC. We selected ortho-cresol (hereafter o-cresol), a product of toluene oxidation and also a directly emitted aromatic species, as a representative aVOC.*"

2) **Line 377. "oxidation state" should be referred to as average carbon oxidation state with the overbar.**

The correction has been made and the sentence now reads as:

"*Figure 1 shows the average carbon oxidation state $\overline{OSc}$ against the number of carbon atoms (nC) of for all the products identified in characteristic experiments for each system (Exp. No. 2, 6 and 10; Table 1).*"

Furthermore, the OSc has been replaced with $\overline{OSc}$ throughout the manuscript.

3) **Line 394. "...and ONO2 addition". You mean NO + RO2 -> RONO2 rather than the addition of -ONO2. Nitration would be more accurate.**

The sentences has been modified as:

"*These products are likely result of H abstraction and subsequent OH addition and nitration, respectively, suggesting that they are early generation products.*"

4) **Section 3.2.1. Some references to the VBS bins are using bin edges (e.g. 3.5-5.5) which is confusing. Referring to bin middles should be fine.**

We agree with the reviewer that this could be confusing. We are now only referring to bin middles instead of edges throughout the manuscript.

5) **Point 22 of the 1st response regarding line 478 of the original manuscript, now line 650 of the revised manuscript. "OH addition via hydrogen abstraction" doesn't make sense, these are separate mechanisms. Olariu et al. (2002) state "At 298 K it has been estimated that approximately 9% and 7% of the overall reactions of OH with phenol and o-cresol, respectively, proceed via H-atom abstraction (Atkinson, 1989), the remainder being addition." The hydrogen abstraction pathway is minor (7%).**

Indeed, we agree that this sentence was incorrect. We have altered this sentence to read as:

*"In o-cresol experiments, the dominance of products in both gas and particle phases having 7 carbon atoms suggest that the OH addition pathway that mainly leads to ring-retaining products, likely plays a major role in the o-cresol photo-oxidation, consistent with previous observations (Olariu et al., 2002;Schwantes et al., 2017)."*

**There are a number of identified typos and formatting errors of which some are listed below. This is not an exhaustive list and another effort at error checking should be made.**

A thorough check has been conducted in the whole manuscript for formatting/linguistic errors.

6) **Line 220. cm-3 formatting**

The correction has been made.

7) **Line 636. "This" / "Thus"?**

"This" was replaced with "Thus" and the sentence now reads as:

*"Thus potentially explaining the higher fraction of more volatile products with log10C*=2 compared to the other systems (Fig. 3 and 4)."*

8) **Line 644. OSc without overbar or subscript (This occurs elsewhere)**

The correction has been made throughout the manuscript, in line with the reviewer's comment #2.

9) **Line 724. "resulting into larger" / results in larger**

"Resulting into larger" was replaced with "results in larger" and the sentence now reads as:

*"This suggests that the potential molecular interactions are results in larger, more functionalised products that have volatilities at the lower end of the SVOC range, decreasing the overall volatility of the mixture."*

**10) Line 728. "on SOA" / "in SOA"**

"On SOA" was replaced with "in SOA" and the sentence now reads as:

*"Previously, Emanuelsson et al. (2013) observed an increase in the volume fraction remaining (and thus a reduction on the volatility) in SOA formed from the mixing various anthropogenic and biogenic precursors reacting sequentially."*

**11) Line 737. missing word**

The sentence has been rephrased as:

*"Previous studies have shown that the ambient OH reactivity (estimated by measurements of OH lifetime) was between ~10 and 116 $s^{-1}$ (Whalley et al., 2016), while in our experiments, OH reactivity (calculated as the product of VOC concentration with reaction rate towards OH) was ~404 $s^{-1}$"*

**12) Line 828. missing word**

The word "two" has been added and the sentence now reads as:

*"The two volatility estimation techniques showed substantial discrepancies in the obtained log10C\* values, highlighting the complexity in the deriving the volatility distribution and the need for further studies to investigate the thermal evaporation of the organics."*

**13) Throughout. "were having" / "had" and similar e.g Line 767**

"were having" was replaced with "had" in that particular example (line 767), while further changes were made in the manuscript after a thorough check for similar grammar mistakes.